# The 3′ Pol II pausing at replication-dependent histone genes is regulated by Mediator through Cajal bodies' association with histone locus bodies

Hidefumi Suzuki[1,8], Ryota Abe[1,8], Miho Shimada[1], Tomonori Hirose [1], Hiroko Hirose[1], Keisuke Noguchi[1], Yoko Ike[1], Nanami Yasui[1], Kazuki Furugori[1], Yuki Yamaguchi [2], Atsushi Toyoda [3], Yutaka Suzuki[4], Tatsuro Yamamoto[5], Noriko Saitoh[5], Shigeo Sato[6], Chieri Tomomori-Sato [6], Ronald C. Conaway[6,7], Joan W. Conaway [6,7] & Hidehisa Takahashi[1✉]

Non-polyadenylated mRNAs of replication-dependent histones (RDHs) are synthesized by RNA polymerase II (Pol II) at histone locus bodies (HLBs). HLBs frequently associate with Cajal bodies (CBs), in which 3′-end processing factors for RDH genes are enriched; however, this association's role in transcription termination of RDH genes remains unclear. Here, we show that Pol II pauses immediately upstream of transcript end sites of RDH genes and Mediator plays a role in this Pol II pausing through CBs' association with HLBs. Disruption of the Mediator docking site for Little elongation complex (LEC)–Cap binding complex (CBC)–Negative elongation factor (NELF), components of CBs, interferes with CBs' association with HLBs and 3′ Pol II pausing, resulting in increased aberrant unprocessed RDH gene transcripts. Our findings suggest Mediator's involvement in CBs' association with HLBs to facilitate 3′ Pol II pausing and subsequent 3′-end processing of RDH genes by supplying 3′-end processing factors.

[1] Department of Molecular Biology, Yokohama City University Graduate School of Medical Science, 3-9 Fukuura, Kanazawa-ku, Yokohama, Kanagawa 236-0004, Japan. [2] School of Life Science and Technology, Tokyo Institute of Technology, 4259 Nagatsuta, Yokohama, Kanagawa 226-8501, Japan. [3] Comparative Genomics Laboratory, National Institute of Genetics, 1111 Yata, Mishima, Shizuoka 411-8540, Japan. [4] Laboratory of Systems Genomics, Department of Computational Biology and Medical Sciences, Graduate School of Frontier Sciences, The University of Tokyo, 5-1-5 Kashiwanoha, Kashiwa, Chiba 277-8562, Japan. [5] Division of Cancer Biology, The Cancer Institute of JFCR, 3-8-31 Ariake, Koto-ku, Tokyo 135-8550, Japan. [6] Stowers Institute for Medical Research, 1000E 50th Street, Kansas City, MO 64110, USA. [7] Department of Biochemistry & Molecular Biology, University of Kansas Medical Center, Kansas City, MO 66160, USA. [8] These authors contributed equally: Hidefumi Suzuki, Ryota Abe. ✉email: hide0213@yokohama-cu.ac.jp

Eukaryotic RNA polymerase II (Pol II) transcribes several types of RNA, including polyadenylated mRNAs, which produce proteins and long noncoding RNAs, and non-polyadenylated RNAs, including small nuclear RNAs (snRNAs), small nucleolar RNAs (snoRNAs), and replication-dependent histone (RDH) mRNAs. In higher eukaryotes, transcription of the genes by Pol II is regulated by at least three steps: transcription initiation, elongation, and termination. Recent studies have shown that, as well as the transcription initiation step, the elongation and termination steps are critical for gene expression[1–5].

Transcription termination is generally accompanied by the 3′-end processing of transcripts. Such processing of genes producing polyadenylated mRNA is achieved by multimeric cleavage and polyadenylation-specific factors (CPSFs), consisting of CPSF1 (CPSF160), CPSF2 (CPSF100), CPSF3 (CPSF73), CPSF4 (CPSF30), hFip1 and WDR33, and multimeric cleavage stimulation factors (CstFs), which produce poly(A) tail transcripts[6–8]. In contrast, the 3′-end processing of replication-dependent histone (RDH) and snRNA genes, which produce non-polyadenylated mRNAs, is achieved by different sets of 3′-end processing factors[2,9]. The 3′-ends of RDH mRNAs are generated by specific processing machinery that recognizes two elements present in pre-mRNAs of RDH genes: a conserved stem-loop region and a purine-rich histone downstream element (HDE)[5]. Cleavage occurs between the two elements and the resulting transcripts are not polyadenylated. The stem-loop and HDE are bound by stem-loop-binding protein (SLBP) and U7 small nuclear ribonucleo-protein (snRNP), respectively[5,6]. FLICE-Associated Huge Protein (FLASH) is a specific factor needed for 3′-end processing of RDH genes and binds to Histone pre-mRNA Cleavage Complex (HCC) composed of CPSF2, CPSF3, Symplekin and CstF64[5,10,11]. It has been shown that FLASH plays a critical role in the recruitment of U7 snRNP through interaction with U7 snRNA-associated Sm-like protein LSM11[10]. The HCC component CPSF3 functions as an endonuclease responsible for the 3′-end cleavage of RDH gene transcripts[12]. Several lines of evidence have indicated that Pol II is arrested or pauses downstream of transcript end sites (TESs) of RDH genes. In vitro transcription assay with RDH genes of *Drosophila melanogaster* revealed that Pol II is arrested 32 to 35 nucleotides downstream of TESs[13,14]. Genome-wide analysis including ChIP-seq of Pol II and Global Run-On sequencing (GRO-seq) using human cells showed that Pol II pauses downstream of TESs of RDH genes[15,16]. Intriguingly, recent reports have shown that protein phosphatase 1 (PP1) dephosphorylates Spt5, a component of DRB-sensitivity inducing factor (DSIF), and decelerates Pol II elongation downstream of TESs of protein-coding genes including RDH genes[17–20]. These reports propose a model that the deceleration of Pol II beyond TESs is required for subsequent 5′→3′ degradation of RNA associated with Pol II. In addition, knockdown of 3′-end processing factors or terminal differentiation of cells causes read-through past normal sites of termination to conserved polyadenylation signals (PASs) just downstream of HDEs, resulting in the synthesis of polyadenylated RDH mRNAs of a small subset of the RDH genes[21,22]. These findings raise the possibility that the polyadenylation of RDH transcripts may also contribute to preventing Pol II read-through into downstream of the RDH genes. Furthermore, recent evidence showing that polyadenylation of RDH transcripts contributes to the expression of RDH genes other than in S phase or in terminally differentiated cells suggests that regulated polyadenylation of RDH genes could contribute to the supply of histone proteins after DNA damage repair or other cellular functions[23,24]. In contrast, 3′-end processing of snRNA genes is dependent on a 3′-box element downstream of the mature 3′-ends of snRNAs[4,9]. Integrator binds to the 3′-box and mediates 3′-end processing of pre-snRNA near the 3′-box[25]. Recently, it has been shown that

Integrator recruits protein phosphatase 2A (PP2A) to dephosphorylate Spt5 and Pol II, and decelerates Pol II elongation to facilitate termination of the transcription of snRNA genes[26]. Consistent with the fact that conserved PAS sequences are not found downstream of most snRNA genes, termination defects do not always result in the polyadenylation of snRNA gene transcripts[27].

The transcription of RDH genes and snRNA genes is regulated at two nuclear bodies, histone locus bodies (HLBs), and Cajal bodies (CBs), respectively[28,29]. After the initial finding of HLBs as nuclear bodies localized at RDH gene clusters in *Drosophila melanogaster*[30], CBs and HLBs have been recognized as distinct nuclear bodies in a variety of species[31–34]. HLBs are thought to be the sites for the synthesis of RDH mRNA and its maturation[28,30]. Nuclear protein, coactivator of histone transcription (NPAT) is an HLB marker protein that plays a critical role in RDH gene transcription[35–38]. It has been shown that NPAT interacts with a DNA-binding transcription activator, POU class 2 homeobox 1 (POU2F1), and functions as a critical coactivator in RDH gene transcription through recruiting a variety of coactivators including the transformation/transactivation domain-associated protein (TRRAP)–Tip60 complex[39,40]. Notably, NPAT is phosphorylated by Cyclin E–CDK2 at the beginning of S phase, triggering the transcription of RDH genes[37,41]. Furthermore, HLBs are the sites for RDH gene transcription and 3′-end processing of RDH gene transcripts[42,43]. CBs, first described by Ramon y Cajal in the early 1900s, are well-characterized nuclear bodies[44]. Coilin is a marker protein for CBs and has been used in immunofluorescent analysis of them[45]. Coilin interacts with a number of the components of CBs including snRNPs, survival of motor neuron protein (SMN), and WD40 repeat-containing protein encoding RNA antisense to p53 (WRAP53), and noncoding RNAs including snRNAs and snoRNAs[46]. Coilin plays an important role in CB formation and small-nuclear ribonucleoprotein (snRNP) assembly[46]. CBs are involved in multiple cellular functions, including transcription activation, RNA processing, biogenesis of a variety of classes of RNPs, and production of the spliceosome components snRNPs[29].

Intriguingly, HLBs and CBs frequently and physically associate with each other, suggesting that there is a functional link between them[29,31,32,47]. The association of CBs and HLBs was shown to be increased at S phase in which RDH genes are transcribed[33,34,48,49]. In addition, the colocalization of CBs and HLBs is increased at the late stage of the oogenesis in both *Xenopus* and *Drosophila*[50]. HLB formation was not affected by the depletion of Coilin;[51,52] in contrast, depletion of HLB components NPAT or FLASH results in the loss of Coilin from the HLBs at RDH genes[53,54]. Thus, the molecular mechanism and biological importance of the colocalization of CBs and HLBs remain unknown. Notably, interactome analysis of Coilin identified a number of constituents of CBs including LSM10 and LSM11, which are components of U7 snRNP[45,55]. In addition, it has been shown that Negative elongation factor (NELF) and CstF64 colocalize with CBs and facilitate 3′-end processing of RDH genes[21,22]. It was also demonstrated that U2 snRNP, which is present at CBs, binds to pre-mRNAs of RDH genes to facilitate U7 snRNP-dependent 3′-end processing[56]. Moreover, genome-wide 4C-seq analysis revealed that CBs associate with HLBs containing RDH gene clusters and play critical roles in both the transcription of RDH genes and the genomic conformation of RDH gene clusters[57]. Thus, CBs have been shown to contain the 3′-end processing factors for RDH genes, including the components of NELF, U7, snRNP, and HCC[29,31,46,58]. Considering that (i) 3′-end processing factors for RDH genes are enriched in CBs and (ii) CBs are frequently and physically associated with HLBs[29,31,32,47–49], 3′-end processing factors for RDH genes may be directly supplied from CBs to HLBs.

Mediator is a megadalton regulatory complex that is essential for multiple processes of transcription regulation[59–62]. It consists of over 30 subunits, each of which binds to particular proteins and is thought to have a unique function in the transcription of specific genes[63]. One of the Mediator subunits, MED26, is specifically found in metazoans. We previously showed that the N-terminal domain (NTD) of MED26 plays a role in the recruitment of Super elongation complex (SEC) or Little elongation complex (LEC) to polyadenylated genes including *c-Myc* and *Hsp70*, or non-polyadenylated genes including snRNA and RDH genes, respectively[64,65]. Thus, MED26-containing Mediator uses two different elongation complexes, SEC and LEC, to regulate different classes of genes. Previously, we showed that MED26 plays an important role in the recruitment of LEC–Cap-binding complex (CBC)–NELF to RDH genes and then helps to recruit 3′-end processing factors, including HCC or Integrator to the RDH or snRNA genes, respectively, leading to appropriate 3′-end processing[22].

In this work, we found that Pol II paused immediately upstream of the TES of RDH and snoRNA genes; in contrast, it paused at multiple sites around TES in snRNA genes. Notably, Pol II paused within the stem-loop region of most RDH genes. We called such pausing of Pol II "TES proximal pausing" (TPP). Knockdown of NELF-E, one of the components of NELF, or CBP80 significantly abolished Pol II TPP at RDH genes and moderately reduced it at snRNA genes, but not at both cistronic and intronic snoRNA genes. As we previously found that MED26 plays a role in transcription termination of RDH and snRNA genes through direct interaction with LEC–CBC–NELF[22], we thought it possible that Mediator's interaction with LEC contributes to TPP and subsequent 3′-end processing of RDH and snRNA gene transcripts. To test this possibility, we mutated the MED26-binding site of EAF1, which plays a role as a site for the docking of LEC on Mediator. In this EAF1-point-mutant cell line, Mediator's interaction with LEC–CBC–NELF was specifically abolished, but that of SEC was not. Intriguingly, EAF1 mutation drastically interfered with Pol II TPP at RDH genes and mildly did so at snRNA genes, resulting in increases in the levels of aberrant unprocessed RDH and snRNA gene transcripts. In addition, we found that EAF1 mutation interfered with CBs' association with HLBs. High-resolution microscopic analysis revealed that, while the components of LEC were mainly colocalized at CBs, Mediator and LSM11 were localized in the region between CBs and HLBs. Combinatorial analyses including antibody-based in situ biotinylation of proteins with high-throughput DNA sequencing (in situ biotinylation-seq) and 4C-seq revealed that CBs' association with RDH gene loci and higher-order inter-chromosome interaction between two RDH gene loci were also decreased in EAF1-mutant cells, indicating that Mediator–LEC interaction is required for higher-order inter-chromosome conformation through CBs' association with RDH gene loci. On the basis of our findings, we propose a model for the role of Mediator in TPP of Pol II through CBs' association with HLBs to facilitate the 3′-end processing of RDH genes. In this model, MED26-containing Mediator plays a role in CBs' association with HLBs through interaction with LEC–CBC–NELF. CBs' association with HLBs leads to TPP at RDH genes and subsequent 3′-end processing. Thus, our results raise the possibility that TPP is a key checkpoint of the transition from transcription elongation to transcription termination for appropriate 3′-end processing of RDH genes and the production of non-polyadenylated transcripts.

## Results

### Identification of genes at which Pol II paused at TES proximal region. To address the mechanisms by which transcription

termination of non-polyadenylated genes is regulated, we performed precision nuclear run-on sequence (PRO-seq) analysis to detect the transcription-engaging Pol II at the genes at single-nucleotide resolution[66]. In this experiment, we performed a nuclear run-on reaction using permeabilized cells. In this reaction, transcription-engaging RNA polymerases are paused by the incorporation of biotinylated NTPs (Supplementary Fig. 1a). We purified the resulting biotinylated RNAs through avidin-biotin purification. The purified RNAs were reverse-transcribed to cDNA and subjected to the construction of a cDNA library for next-generation sequencing. Thus, PRO-seq enables detection of the 3′ end of the nascent transcripts very precisely at single-nucleotide resolution. We did not detect any amplified cDNAs in library construction in the absence of biotin-NTPs (Supplementary Fig. 1b), indicating that we purified only biotinylated RNAs and eliminated contaminated RNAs from the cells. We calculated the 3′ pausing index by dividing the PRO-seq peaks of the TES proximal region (TESr) by those of the gene body (GB) (Fig. 1a, b). We found that Pol II strikingly paused at the TESr of genes including RDH, snRNA and snoRNA (Fig. 1c–g and Supplementary Fig. 2a–f). Pol II paused immediately upstream of the TES of RDH and snoRNA genes (Fig. 1c–e, g, h and Supplementary Fig. 2a, b, e); in contrast, it paused at multiple sites upstream or downstream of the TES in snRNA genes (Fig. 1f, h and Supplementary Fig. 2c, d). As there are two types of snoRNA gene, one of which is independently transcribed and the other is present in introns of snoRNA host genes and processed by exonucleases from the intron of the gene's transcript[67], we investigated which types of snoRNA genes are included among the genes at which Pol II paused. We found that Pol II paused immediately upstream of the TES of both types of snoRNA gene (Supplementary Fig. 2g). Such striking pausing of RNA polymerase was also found at Pol I-driven rRNA genes (Fig. 1a, b and Supplementary Fig. 2f), indicating that Pol I paused around the TES of these genes. Notably, in most of the RDH genes, Pol II pauses within the stem-loop region (Fig. 1c–e and Supplementary Fig. 3), indicating that it pauses before the stem-loop structure is formed at RDH genes. We called such pausing around TES "TES proximal pausing (TPP)." Comparison of genome browser tracks showing the distribution of the 3′-end of reads obtained from PRO-seq and RNA-seq in RDH, snoRNA, snRNA, rRNA and protein-coding genes showed that the peaks of TPP observed in PRO-seq were not detected in RNA-seq, ruling out the possibility that the peaks of TPP are derived from contaminated RNAs (Supplementary Fig. 1c–h). In previous studies, ChIP-seq of Pol II and GRO-seq revealed that Pol II pauses downstream of RDH genes[15,16]. As PRO-seq enables precise detection of the 3′ end of nascent transcripts, which are produced by elongating or paused polymerases, but not stalled or arrested polymerases, we think that our results do not contradict the previous reports based on ChIP-seq and GRO-seq. Taking the findings together, our analysis revealed that Pol II pauses immediately upstream of the TES of RDH genes. Next, we calculated the 5′ pausing index by dividing the PRO-seq peaks of the TSS (transcription start site) proximal region (TSSr) by those of GB and detected the genes in which Pol II paused at the promoter proximal region, which is commonly called promoter proximal pausing (PPP) of Pol II. While these PPP genes included a subset of protein-coding genes such as *Hsp70*, *c-Myc*, and *Snail2* (Supplementary Fig. 2h–j), pronounced TPP was not detected among them (Fig. 1a, b and Supplementary Fig. 2h–j). As shown in Supplementary Fig. 2k, we also found TPP at RDH and snRNA genes using PRO-seq data in a previous report[68].

Previous studies showed that the 3′ end of histone mRNA is digested 2–4 nucleotides into a stem-loop by 3′ exonuclease and then uridylated by TUT7 in the process of cytoplasmic

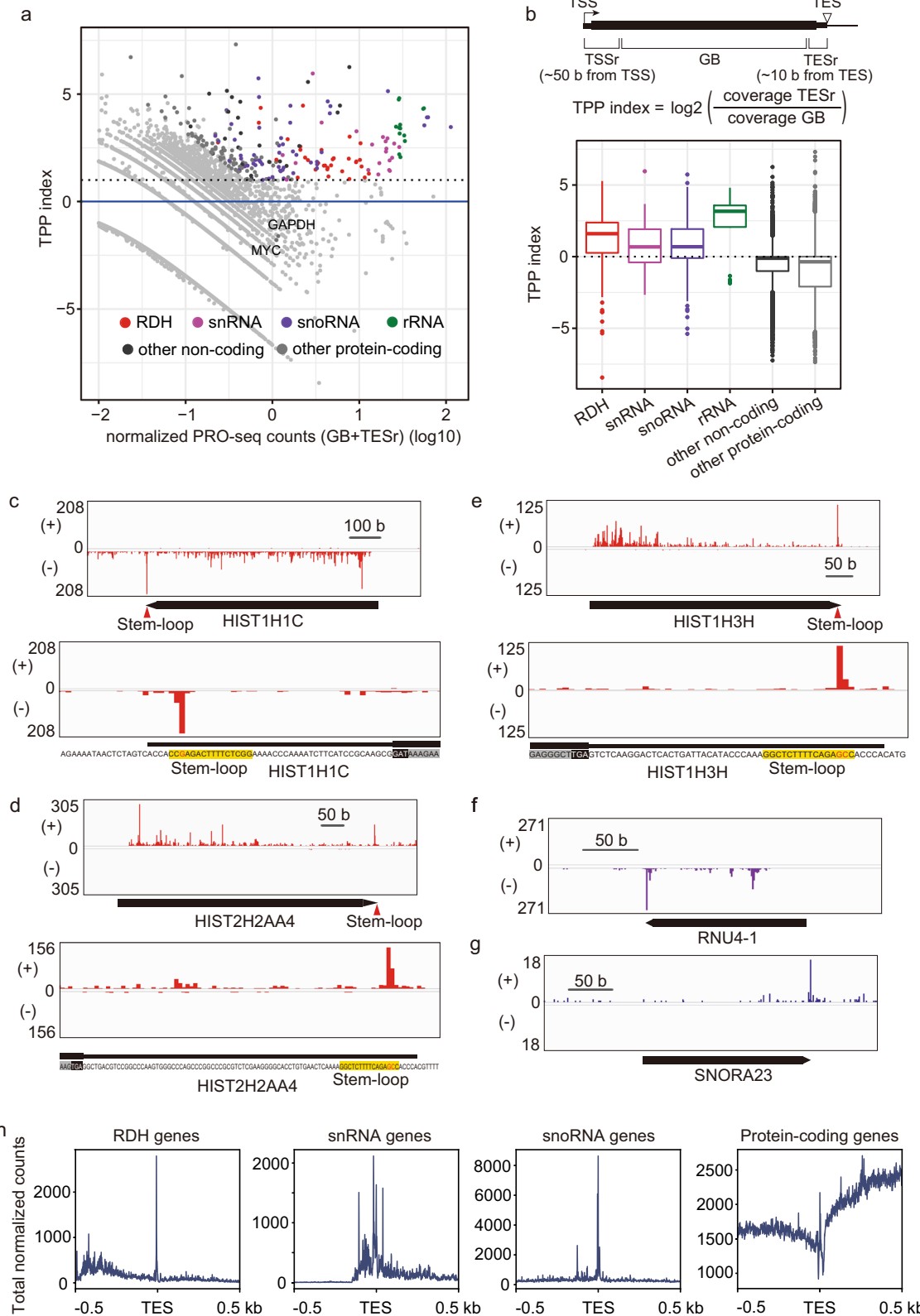

$$\text{TPP index} = \log_2\left(\frac{\text{coverage TESr}}{\text{coverage GB}}\right)$$

degradation[69,70], raising the possibility that biotinylated NTPs are added post-transcriptionally in the PRO-seq and the resulting 3′ end of the intermediate transcripts is detected as 3′ pausing sites. To address this possibility, we performed the PRO-seq analysis in the presence of α-amanitin, which blocks transcription by Pol II but not Pol I or Pol III. As shown in Supplementary Fig. 4a, b, PPP of the protein-coding genes such as *Fos* was markedly

decreased by α-amanitin, which was accompanied by the marked decreases of the elongated transcripts of the genes. Consistent with the ability of α-amanitin to block the activity of Pol II, but not that of Pol I, the PRO-seq peaks of the rRNA genes which are transcribed by Pol I, were not affected by α-amanitin (Supplementary Fig. 4c). We found that α-amanitin markedly decreased the signals of TPP at RDH, snRNA and both independently

**Fig. 1 Pol II paused at the TES proximal region of non-polyadenylated genes. a** PRO-seq revealed 3′ Pol II pausing at RDH genes, snRNA/snoRNA genes, and rRNA genes. Scatter plot showing TES proximal pausing (TPP) index relative to the normalized PRO-seq counts. **b** The TPP index was determined using the formula shown in the upper panel. Boxplots representing the TPP index for RDH genes ($n = 68$), snRNA genes ($n = 57$), snoRNA genes ($n = 231$), rRNA genes ($n = 17$), other noncoding genes ($n = 6275$), and other protein-coding genes ($n = 18,952$). The center line of each boxplot represents the median. Upper and lower fences of each boxplot represent upper and lower quartiles, respectively. Source data are provided as a Source Data file. **c–g** Genome browser tracks showing the distribution of PRO-seq reads at *HIST1H1C* (**c**), *HIST2H2AA4* (**d**), *HIST1H3H* (**e**), *RNU4-1* (**f**), and *SNORA23* (**g**) in wild-type HEK293T cells. The positive and negative strands of DNA are indicated by (+) and (−), respectively. Lower panels of (**c**), (**d**), and (**e**) show detailed genome browser tracks, indicating the distribution of PRO-seq reads at the transcript end site of the *HIST1H1C* (**c**), *HIST2H2AA4* (**d**), and *HIST1H3H* (**e**) genes in wild-type HEK293T cells. The stem-loop sequence located immediately upstream of TES of each RDH gene is highlighted in yellow. **h** Meta-gene analysis of PRO-seq reads around TESs of Pol II-transcribed genes. TSSr: Transcription start site (TSS) proximal region, TESr: Transcript end site proximal region.

transcribed snoRNA and intronic snoRNA genes (Supplementary Fig. 4d–m). The decrease of TPP at RDH genes was accompanied by the marked decrease of PPP at the genes (Supplementary Fig. 4h). These results indicate that the TPP as well as PPP observed at the RDH, snRNA and snoRNA genes are derived from Pol II and not a post-transcriptional modification. Of note, α-amanitin markedly, but not completely, blocked the Pol II activity in our PRO-seq, consistent with the evidence that α-amanitin slows the Pol II elongation and reduces substrate specificity through interactions with the trigger loop of Pol II but does not completely inhibit Pol II elongation activity[71–73]. In this study, we detected the TPP in our PRO-seq using permeabilized cells. In contrast, we did not detect the TPP in our previous study, in which we used Dounce homogenized cells[22]. The original protocol of PRO-seq recommends using permeabilized cells rather than Dounce homogenized cells because the recovery of transcripts is higher from permeabilized cells than Dounce homogenized cells[66] and therefore we used permeabilized cells in this study. One possible reason for the differences in TPP detection is that the homogenizing process disrupts the factors that are involved in TPP.

As we previously found that the human Mediator subunit MED26 plays a role in transcription termination of non-polyadenylated genes through direct interaction with LEC[22], we took advantage of ChIP-seq data of MED26 and compared these TPP genes with the genes at which MED26 is present. We found that TPP genes include many of the non-polyadenylated genes that were detected in MED26 ChIP-seq analysis (Supplementary data 1). These results raised the possibility that the human Mediator subunit MED26 plays a role in TPP of non-polyadenylated genes through interacting with LEC.

**Knockdown of NELF or CBP80 decreases TPP at RDH genes.** Studies have shown that CBC–NELF plays a critical role in transcription termination of snRNA and RDH genes[21,27,74], and we previously found that LEC plays a role in transcription termination of these genes through interaction with CBC–NELF[22], and therefore we tested whether knockdown of the components of CBC–NELF, NELF-E, or CBP80 affects TPP at RDH, snRNA, and snoRNA genes. Knockdown of NELF-E or CBP80 significantly decreased TPP at RDH genes (Fig. 2a–c, e and Supplementary Fig. 5a). In snRNA genes, knockdown of NELF-E or CBP80 mildly decreased one of the multiple TPPs, which occurs immediately upstream of TES (Fig. 2d and Supplementary Fig. 5b), although overall 3′ Pol II pausing at snRNA genes was not significantly decreased by knockdown of NELF-E or CBP80 (Fig. 2e). In contrast, knockdown of NELF-E or CBP80 did not affect TPP at snoRNA and transcription-engaging Pol II around the TES of protein-coding genes (Fig. 2e and Supplementary Fig. 5c, d). This indicates that CBC–NELF is specifically involved in TPP at RDH and snRNA genes, raising the possibility that TPP plays a role in repressing read-through of Pol II at the

genes for subsequent 3′-end processing of non-polyadenylated gene transcripts.

**Point mutation of EAF1 abolishes Mediator's interaction with LEC–CBC–NELF.** To assess the role of Mediator's interaction with LEC in TPP, we generated a mutant cell line in which Mediator's interaction with LEC is specifically abolished but that of SEC is not. Both SEC and LEC interact with Mediator via MED26's N-terminal domain (NTD) and substitution of the 61st arginine and 62nd lysine residues of MED26-NTD to alanines interferes with MED26–NTD's interaction with SEC and LEC[64,65]. In addition, we previously showed that the C-terminal region of EAF1 functions as a MED26-binding interface of LEC[22,65]. Meanwhile, it has been shown that EAF1 is a shared component of SEC and LEC, AF4 and AFF4, which are components of SEC, also directly bind to MED26[75]. Thus, the MED26-binding interface of SEC has not been clearly determined. Against this background, we tested the possibility that AF4 or AFF4 also contributes to the MED26-binding interface of SEC. As SEC components—AF4 and AFF4—contain amino acid sequences similar to the EAF1 C-terminal region (Fig. 3a), we produced recombinant proteins of the mutant form of AF4 or AFF4 in which three amino acid residues of the region were substituted for alanines (Fig. 3a). In addition, we took advantage of the wild-type of MED26-NTD and mutant form of MED26-NTD in which the 61st arginine and 62nd lysine residues were substituted for alanines. We tested whether the wild-type or mutant form of AF4 or AFF4 directly binds to MED26-NTD. As shown in Fig. 3b, c, the wild-type of AF4 or AFF4 directly bound to the wild-type of MED26-NTD, but not its mutant form. In contrast, the mutant form of AF4 or AFF4 did not bind to the wild-type of MED26-NTD. This raises the possibility that AF4 and AFF4 as well as EAF1 contribute to the MED26-binding interfaces of SEC. As we previously showed that only EAF family member proteins, EAF1/2, function as LEC docking sites for Mediator, we expected that the mutation of MED26's -binding site of EAF1 would specifically interfere with Mediator's interaction with LEC but not that with SEC. Additionally, LEC contains substantially more EAF1 than EAF2[22], raising the possibility that mutation of EAF1 is sufficient to inhibit LEC's interaction with Mediator. We took advantage of the clustered regularly interspaced short palindromic repeat (CRISPR) system and generated an EAF1-point-mutant HEK293T cell line, in which the four amino acids of MED26's-binding site of EAF1 are mutated to alanines (Fig. 3d). To confirm that the mutation of EAF1 specifically interferes with Mediator's interaction with LEC, we stably expressed FLAG-tagged MED26 (F-MED26) in wild-type or EAF1-mutant HEK293T cells. F-MED26 copurified with the components of LEC, ICE1 and EAF1 in wild-type cells, but did not in EAF1-mutant cells (Fig. 3e). We previously showed that LEC interacts with CBC–NELF through the direct interaction of ICE1 and CBP80[22], a large subunit of CBC, and therefore we tested whether

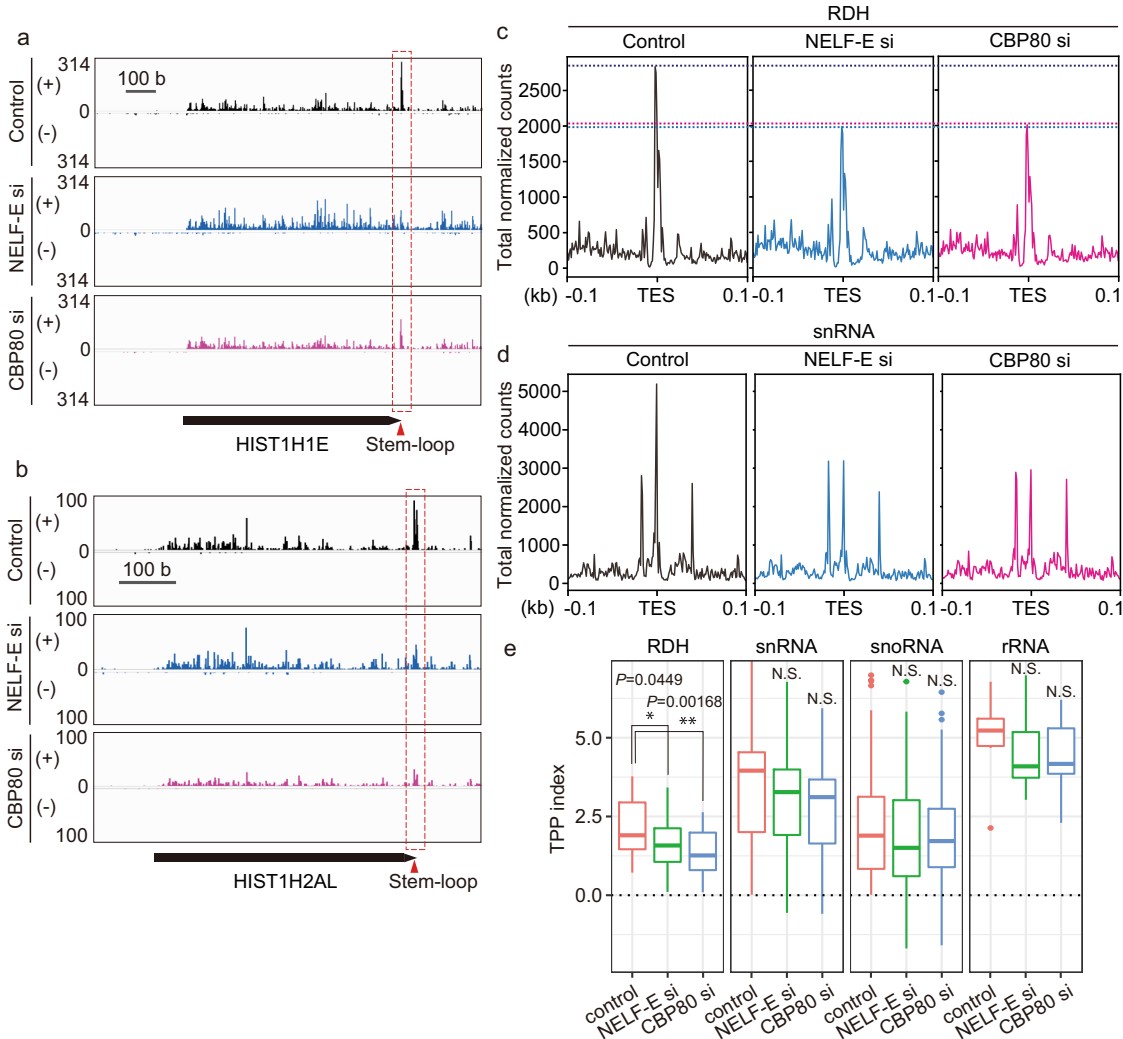

**Fig. 2 Knockdown of NELF or CBP80 decreased the Pol II pausing at the TES proximal region of RDH genes. a, b** Genome browser tracks showing the distribution of PRO-seq reads at *HIST1H1E* (**a**) and *HIST1H2AL* (**b**) in control, NELF-knockdown (NELF-E si) and CBP80-knockdown (CBP80 si) cells. The positive and negative strands of DNA are indicated by (+) and (−), respectively. **c, d** Meta-gene analysis of PRO-seq reads around TESs of RDH genes (**c**) and snRNA genes (**d**) in control, NELF-knockdown (NELF-E si) and CBP80-knockdown (CBP80 si) cells. **e** Boxplots representing the TPP index for RDH genes (*n* = 33), snRNA genes (*n* = 25), snoRNA genes (*n* = 80) and rRNA genes (*n* = 8) in control, NELF-knockdown (NELF-E si) and CBP80-knockdown (CBP80 si) cells. The center line of each boxplot represents the median. Upper and lower fences of each boxplot represent upper and lower quartiles, respectively. The *P*-values were determined by two-sided Wilcoxon's signed-rank test (*$P < 0.05$; **$P < 0.01$). N.S., not significant. Source data and exact *P*-values are provided as a Source Data file.

EAF1 mutation affects Mediator's interaction with CBC–NELF in cells. F-MED26 copurified with CBP80, and a component of NELF, NELF-B, in wild-type cells, but did not in EAF1-mutant cells (Fig. 3e). As studies have shown that Integrator as well as NELF colocalized with CBs[21,76], we tested whether EAF1 mutation affects Mediator's interaction with the components of CBs. F-MED26 copurified with Coilin and INTS4, a component of Integrator, as well as NELF-B in wild-type cells, but the levels of these components copurified with F-MED26 were decreased in EAF1-mutant cells (Fig. 3e). In contrast, F-MED26 copurified with similar amounts of the components of HLBs, NPAT, and the components of SEC and Mediator in both wild-type and EAF1-mutant cells (Fig. 3e). This indicates that mutation of EAF1 specifically interferes with Mediator's interaction with LEC–CBC–NELF. In addition, it is possible that the loss of Mediator's interaction with LEC interferes with Mediator's association with CBs but not its association with HLBs. We found that much more EAF2 was copurified with F-MED26 in EAF1-mutant cells and that only small amounts of ICE1 were still

copurified with F-MED26 in EAF1-mutant cells (Fig. 3e), raising the possibility that EAF2 also compensatorily contributes to LEC's interaction with Mediator in cells.

Next, we reconstituted LEC by co-expressing HA-tagged ICE1, FLAG-tagged ICE2, HA-tagged ELL and HA-tagged EAF1 wild-type (wt) or mutant (mut) in a baculoviral expression system and confirmed that point mutation of EAF1 does not interfere with the complex formation of LEC. Consistent with our expectations, the compositions of LEC containing EAF1-wt or EAF1-mut were similar (Fig. 3f). Taken together, these results indicate that, first, EAF1 functions as an interface by which LEC–CBC–NELF interact with Mediator and, second, multiple components of SEC, including AF4, AFF4 and EAF, can contribute to SEC's interaction with Mediator (Fig. 3g).

**TPP is abolished at RDH and snRNA genes in EAF1-mutant cells.** We next examined how the loss of Mediator and LEC–CBC–NELF interaction impacts on the genome-wide

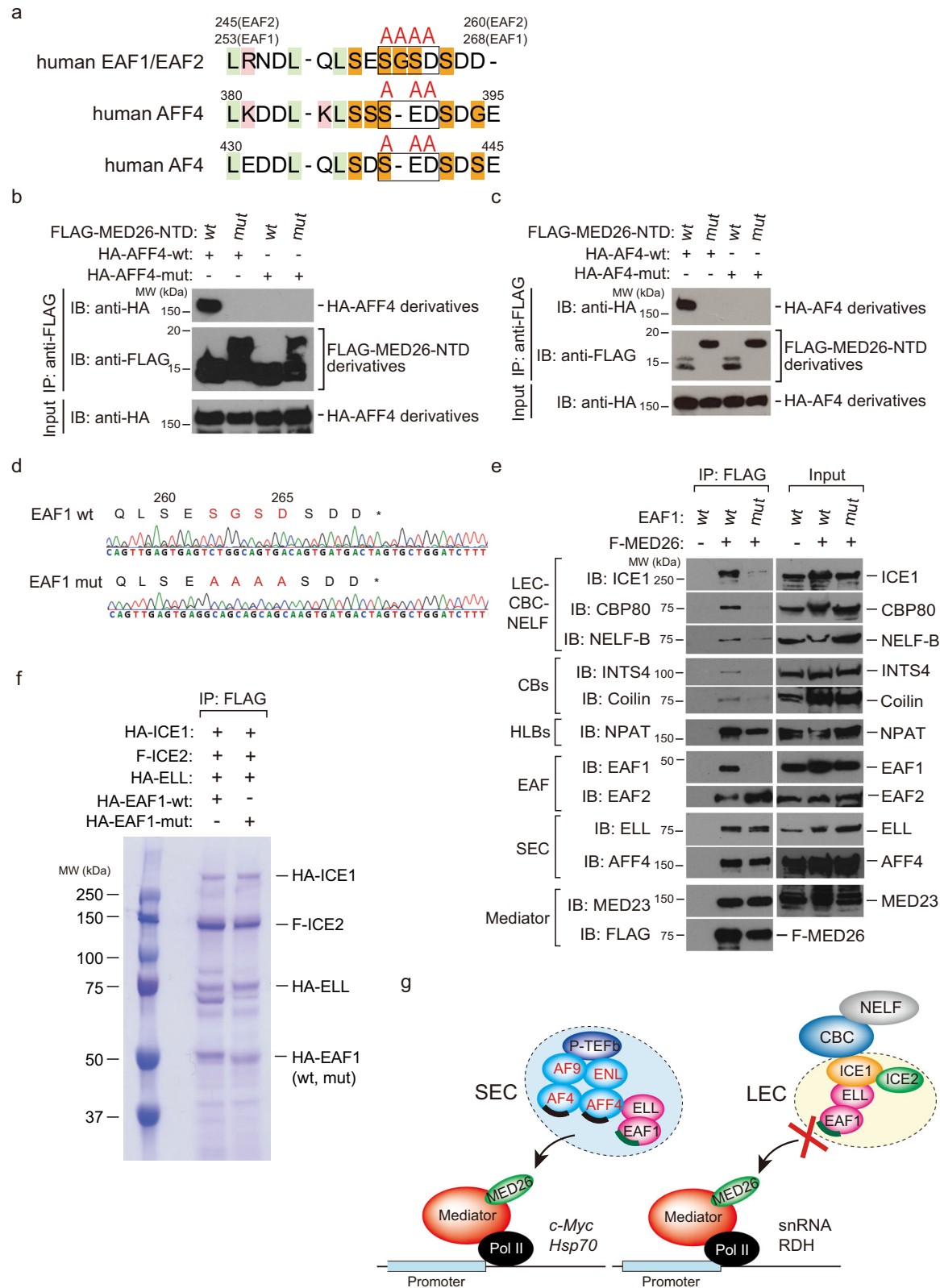

transcription of genes in cells. RNA-seq analysis revealed that the expression of snRNA and RDH genes was downregulated in EAF1-mutant cells (Fig. 4a and Supplementary data 2), which is consistent with our previous findings that LEC recruitment by Mediator is required for the expression of snRNA and RDH genes[22]. Notably, the expression of ELL, a component of LEC, and MED26-associated RDH genes was higher than that of RDH

genes, which had lower association with ELL and MED26 (Fig. 4b). As we expected, the expression of ELL and MED26-associated RDH genes was decreased more than that of other RDH genes in EAF1-mutant cells (Fig. 4b). Furthermore, the occupancy of total Pol II, phosphorylated Pol II (S5P, S2P, and S7P), in which a seven-peptide repeat of Pol II C-terminal domain (CTD) is phosphorylated, and ELL at RDH genes was

**Fig. 3 Point mutation of MED26-binding site of EAF1 abolished Mediator's interaction with LEC–CBC–NELF. a** Sequence similarity among human EAF1/ EAF2, AFF4, and AF4. SEC components AF4 and AFF4 harbor amino acid sequences similar to the EAF1 C-terminal region critical for interaction with MED26. In mutant forms of EAF1, AFF4, and AF4, the three or four amino acids indicated by a black frame were replaced by alanines. The light-green column indicates hydrophobic residues, the pink column indicates basic residues and the orange column indicates neutral residues. **b, c** FLAG-immunopurified complexes from baculovirus-infected Sf9 cells expressing the indicated proteins were analyzed by western blotting. F-MED26-NTD mut has a larger molecular weight than F-MED26-NTD wt, since the linker between the N-terminal hexa-Histidine tag and the FLAG tag of MED26-NTD mut is longer than that of MED26-NTD wt. **d** DNA sequencing of the EAF1 gene in wild-type (wt) and EAF1-C-terminal-mutant (mut) cells. The amino acids serine (S), glycine (G), serine (S), and aspartate (D) of the EAF1 C-terminal region were replaced by alanines (A) in EAF1-mutant HEK293T cells. **e** Western blotting for immunoprecipitates copurified with FLAG-tagged MED26 (F-MED26) by anti-FLAG antibody. F-MED26 was stably expressed in HEK293T cells expressing wild-type EAF1 (wt) or EAF1-mutant (mut). Anti-FLAG-immunopurification was performed and the resulting immunoprecipitates were subjected to western blotting. **f** Coomassie Brilliant Blue staining of reconstituted LEC. LEC containing wild-type EAF1 or mutant EAF1 was reconstituted using the baculoviral expression system. HA-ICE1, F-ICE2, HA-ELL, and HA-EAF1-wt or HA-EAF1-mut were co-expressed and purified through anti-FLAG affinity chromatography. **g** Model of SEC or LEC recruitment by Mediator. SEC contains three subunits, namely, AFF4, AF4, and EAF, which contribute to the interaction between SEC and MED26; in contrast, LEC contains only EAF, which functions as an adaptor for LEC's interaction with MED26.

significantly decreased in EAF1-mutant cells, while total Pol II occupancy at the promoter of the *c-Myc* gene, one of the SEC-target genes, or *GAPDH* gene was not affected in EAF1-mutant cells (Fig. 4c, d and Supplementary Fig. 6a–d). This is consistent with our previous finding that LEC recruited by Mediator plays a role in the transcription elongation of RDH genes[22]. Furthermore, meta-gene analysis of RDH gene transcripts revealed that RDH transcripts of the gene body (GB) were decreased, and read-through (RT) transcripts were increased at RDH genes in EAF1-mutant cells (Fig. 4e, f), suggesting that the 3′-end processing of RDH gene transcripts was also decreased in EAF1-mutant cells. Point mutation of EAF1 resulted in little change in the levels of LEC components EAF1, ICE1 and ELL; Mediator components MED26, MED1 and MED23; total Pol II and phosphorylated Pol II (S5P, S2P and S7P); and a component of the 3′-end processing factor LSM11 (Supplementary Fig. 7a), suggesting that point mutation of EAF1 does not lead to defects of transcription termination of RDH genes by affecting the expression of factors needed for 3′-end processing.

Next, we considered the possibility that the defect in transcription termination observed in EAF1-mutant cells was caused by decreased TPP of non-polyadenylated genes through the loss of Mediator's interaction with LEC–NELF–CBC. To address this possibility, we performed PRO-seq to detect the alteration in transcription-engaging Pol II in EAF1-mutant cells. This analysis revealed that TPP of RDH genes clearly disappeared in EAF1-mutant cells (Fig. 5a–d). Meta-gene analysis of the RDH genes also revealed that the PRO-seq peaks immediately upstream of TES observed in wild-type cells were clearly decreased in EAF1-mutant cells (Fig. 5e). Meta-gene analysis also revealed that multiple TPPs at snRNA genes were mildly decreased in EAF1-mutant cells (Fig. 5f), consistent with the results showing that knockdown of NELF-E or CBP80 also mildly decreased TPP at snRNA genes (Fig. 2d). In contrast, we did not find any change in transcription-engaging Pol II around the TES of other protein-coding genes (Fig. 5g). In addition, PPP at the genes including *GAPDH* and other protein-coding genes was slightly changed in EAF1-mutant cells (Fig. 5h, i), suggesting that Mediator's interaction with LEC–NELF–CBC impacts on the TPP of RDH genes and snRNA genes. We also found that the read-through of Pol II at RDH genes was significantly increased in EAF1-mutant cells (Fig. 5j). We calculated the read-through ratio of RDH and snRNA genes by dividing the sum of read counts 200 bp downstream of TES by the sum of those of gene bodies. By this approach, we found that the read-through of Pol II at RDH and snRNA genes was significantly increased in EAF1-mutant cells (Fig. 5k). We next used a CRISPR-generated, MED26-hypomorphic-mutant HEK293T cell line that expresses mutant MED26 lacking the NTD required for LEC's interaction with Mediator[22] and performed PRO-seq. Similar to the results of

EAF1-mutant cells, the TPP of RDH genes disappeared in MED26-mutant cells (Supplementary Fig. 7b–e). Meta-gene analysis of the RDH genes also revealed that TPP observed in wild-type cells was clearly decreased in MED26-mutant cells, which was accompanied by the increased read-through ratio of RDH genes in MED26-mutant cells. However, the PPP of the genes was not significantly affected (Supplementary Fig. 7f–h). In addition, meta-gene analysis revealed that TPP at snRNA genes was mildly decreased in MED26-mutant cells (Supplementary Fig. 7i, j), consistent with the results of the EAF1-mutant cell line (Fig. 5f). In contrast, we found little change in PPP at the *GAPDH* gene in MED26-mutant cells (Supplementary Fig. 7k). The results of the PRO-seq observed in MED26-mutant cells were similar to those of EAF1-mutant cells (Fig. 5a–f), strongly supporting the idea that Mediator's interaction with LEC–CBC–NELF is required for TPP and subsequent 3′-end processing of RDH and snRNA genes.

**Decreased association of CBs with HLBs in EAF1-mutant cells.** It has been shown that CBs frequently associate with HLBs, so we next investigated whether Mediator–LEC interaction contributes to CBs' association with HLBs. We observed CBs and HLBs in wild-type or EAF1-mutant cells by immunostaining of Coilin and NPAT, respectively. We extracted the particles of NPAT from images and calculated the area of these particles that are occupied with Coilin particles and the intensity of the particles of NPAT and Coilin. This analysis revealed that the association of Coilin and NPAT was decreased in EAF1-mutant cells (Fig. 6a), suggesting that the Mediator–LEC interaction is required for CBs' association with HLBs. We found that 148 of 300 Coilin particles associated with NPAT particles in wild-type cells. In contrast, only 12 of 300 Coilin particles associated with NPAT in EAF1-mutant cells (Fig. 6c). Consistent with this result, NPAT colocalization with other CB components, including ICE1, MED26, or LSM11 was also decreased in EAF1-mutant cells (Supplementary Fig. 8a–c). Furthermore, point mutation of EAF1 resulted in little change in the protein expression levels of the components of CBs and HLBs, including Coilin, NPAT, and LSM11 (Supplementary Fig. 8d), indicating that point mutation of EAF1 does not lead to defects of CBs' association with HLBs by affecting the expression of the factors needed for their association. We next tested whether CBs' association with HLBs was affected in MED26-hypomorphic-mutant HEK293T cells. CBs' association with HLBs was also decreased in these MED26-hypomorphic-mutant cells (Fig. 6b). We found that 106 of 200 Coilin particles associated with NPAT particles in the wild-type cells. In contrast, only 38 of 200 Coilin particles associated with NPAT in the MED26-hypomorphic-mutant cells (Fig. 6d). Notably, although the association between CBs and HLBs was drastically decreased by

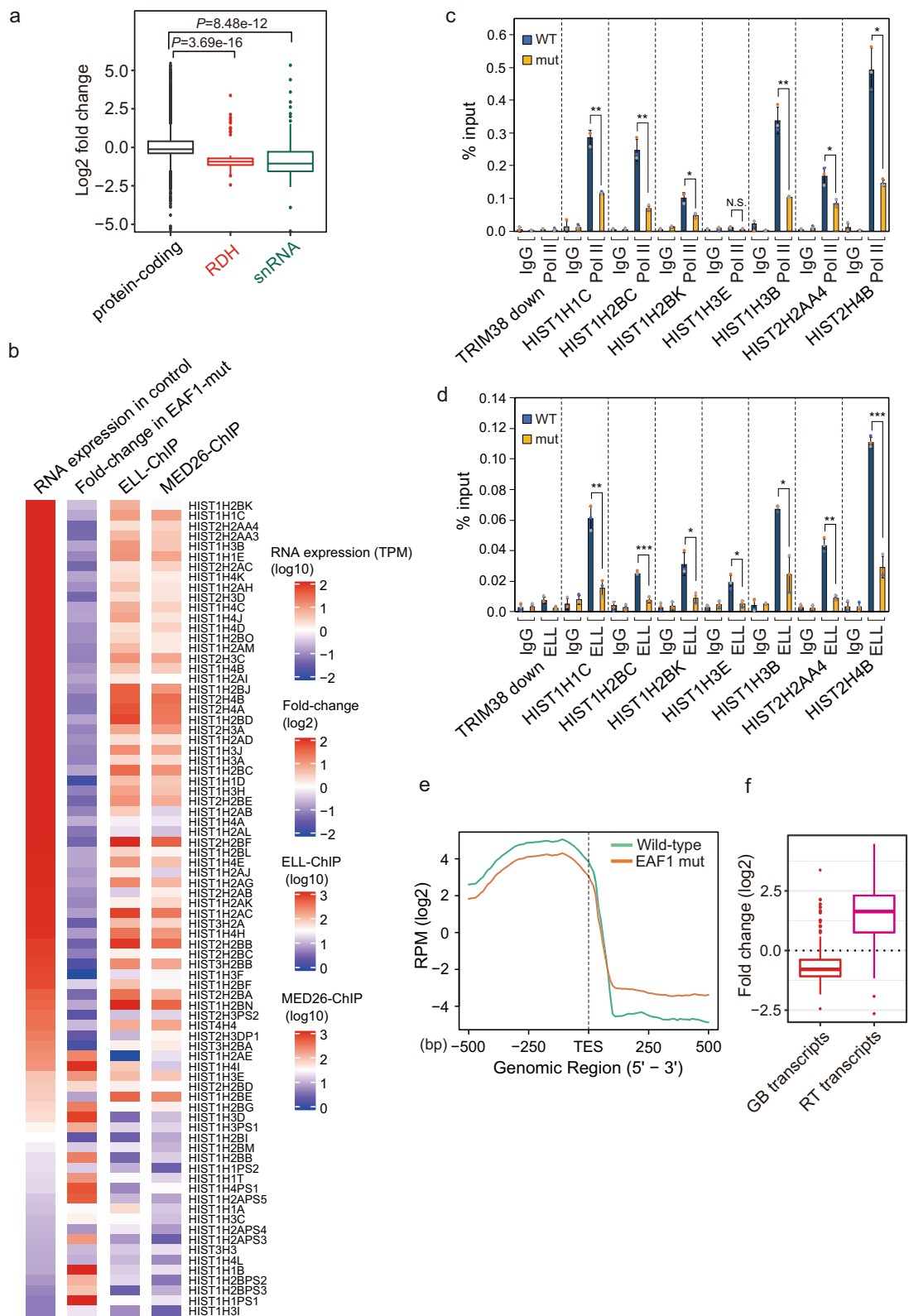

mutation of EAF1 or MED26, the number of CBs was not significantly decreased by these mutations (Fig. 6c, d).

To assess that the Coilin particles, which we observed contain other CB components, we performed immunofluorescence of SMN1 and WRAP53, which colocalize at CBs[32,77,78]. SMN1 is one of the components of the SMN complex that plays a critical role for snRNP biogenesis through translocating snRNPs from the cytoplasm to CBs[79–82]. WRAP53 is essential for CB formation and plays a role in targeting Small Cajal body-specific RNAs (scaRNAs) to CBs through direct interaction[78,83]. We performed triple immunofluorescence analysis of NPAT, Coilin and SMN1 or WRAP53 using HEK293T wild-type and EAF1-mutant cell. As shown in Supplementary Fig. 9a, b, Coilin particles that contain SMN1 or WRAP53 were associated with NPAT in HEK293T

**Fig. 4 Transcription termination defects of RDH and snRNA genes in EAF1-mutant cells. a** Boxplots representing relative mRNA expression levels in EAF1-mutant cells. Relative expression levels (adjusted *P*-value < 0.1) of protein-coding genes (*n* = 13,846, black), RDH genes (*n* = 67, red) and snRNA genes (*n* = 3837, green) were determined and are shown by boxplots. The center line of each boxplot represents the median. Upper and lower fences of each boxplot represent upper and lower quartiles, respectively. The *P*-values were determined by two-sided Wilcoxon's signed-rank test. Source data are provided as a Source Data file. **b** Heatmap showing basal mRNA expression levels of RDH genes (left), fold-change in mRNA expression level in EAF1-mutant cells (middle) and ELL and MED26-ChIP enrichment (right). **c, d** Pol II or ELL occupancy (ChIP/input) at each genomic locus was analyzed in wild-type (WT) and EAF1-mutant (mut) cells. Ct values of each ChIP were normalized to that of the input. Each value is the mean of three independent experiments. Error bars show standard deviation. The *P*-values were determined by two-sided Student's *t* test (\**P* < 0.05; \*\**P* < 0.01; \*\*\**P* < 0.001). *n* = 3 biologically independent samples. N.S., not significant. Source data and exact *P*-values are provided as a Source Data file. **e** Meta-gene analysis of RDH gene transcripts. Increased transcripts were detected downstream of the transcription termination site of RDH genes. **f** Boxplots representing RDH gene transcripts (*n* = 81) of gene body (GB) and read-through (RT) in EAF1-mutant cells. The center line of each boxplot represents the median. Upper and lower fences of each boxplot represent upper and lower quartiles, respectively.

wild-type cells. In contrast, the association between Coilin particles containing SMN1 or WRAP53 and NPAT were decreased in EAF1-mutant cells. Triple immunofluorescence analysis revealed that a large population of Coilin particles contain WRAP53 and that their association with NPAT was decreased in EAF1-mutant cells (Supplementary Fig. 9c). These results showed that Coilin particles in both wild-type and EAF1-mutant cells contain other CBs components including SMN1 and WRAP53, suggesting that they have functions for targeting scaRNAs to CBs and snRNP biogenesis. In addition, we performed RNA-FISH of scaRNAs followed by immunofluorescence of Coilin and NPAT. Consistent with the previous findings that scaRNAs colocalize with CBs[84], Coilin particles containing scaRNA12 were colocalized with NPAT in HeLa cells (Supplementary Fig. 10a).

To investigate whether the distance between CBs and HLBs is increased in EAF1-mutant cells, we next investigated the localization of CBs, HLBs and Mediator by a super-resolution imaging technique. We found that, as well as a significant decrease in the frequency of colocalization between NPAT (HLBs) and Coilin (CBs) in EAF1-mutant cells, the distance between NPAT (HLBs) and Coilin (CBs) was increased in these cells, while MED26 remained colocalized with NPAT (HLBs) (Fig. 6e, f). Consistent with our idea that Mediator–LEC interaction contributes to CBs' association with HLBs, F-MED26 copurified with components of CBs, including Coilin, INTS4, and NELF-B in wild-type cells, but the levels of these components copurified with F-MED26 were decreased in EAF1-mutant cells (Fig. 3e). In contrast, F-MED26 copurified with similar amounts of NPAT and the components of Mediator in wild-type and EAF1-mutant cells (Fig. 3e) and the colocalization of MED26 and NPAT (HLBs) was not affected in EAF1-mutant cells (Fig. 6e), suggesting that the loss of Mediator–LEC interaction interferes with Mediator's association with CBs but not its association with HLBs.

**Mediator localized at the region between CBs and HLBs**. We considered the possibility that Mediator contributes to CBs' association with HLBs through interaction with LEC, so we next investigated the relative positions of the components of Mediator, LEC and 3′-end processing factors at CBs and HLBs. We performed triple immunofluorescence analysis using HeLa cells and detected NPAT particles that associate with both particles of Coilin and other factors including the components of Mediator, LEC, or 3′-end processing factors. We generated averaged particle images from more than 50 particles and calculated the length between the center of each Coilin or NPAT particle and the center of each particle of Mediator components, MED26 or MED1, LEC components, ELL or ICE1, and 3′-end processing factors, LSM11 or FLASH (Fig. 7a–f). Coilin and NPAT were the most distant, consistent with them being markers for CBs and

HLBs, respectively (Fig. 7a–f). Intriguingly, the components of Mediator, MED1 and MED26, and the component of 3′-end processing factors, LSM11, localized between CBs and HLBs (Fig. 7a, b, e). In contrast, the components of LEC, ELL and ICE1 mainly colocalized at CBs and the component of 3′-end processing factors, FLASH, mainly colocalized at HLBs (Fig. 7c, d, f). Comparable localization patterns were also observed in HEK293T cells and HCT116 cells (Supplementary Fig. 10b, c). To further address the possibility that Mediator localizes between CBs and HLBs, we employed stimulated emission depletion (STED) super-resolution microscopy. As shown in Fig. 7g, h, particles of NPAT were clearly distinguished from those of Coilin and particles of MED26 much more closely colocalized with Coilin. We calculated what proportions of ICE1, ELL, LSM11, MED1, MED26 and FLASH colocalized with CBs, HLBs, or both CBs and HLBs (Fig. 7i). This analysis revealed that, first, LEC mainly colocalized with CBs, second, FLASH mainly colocalized with HLBs, and, third, Mediator and LSM11 colocalized with both CBs and HLBs (Fig. 7i). Taken together, these results suggest that Mediator and LSM11 localized between CBs and HLBs (Fig. 7j).

We next investigated how knockdown of these factors affected the formation of CBs and HLBs. Knockdown of ICE1 drastically decreased the formation of CBs (Supplementary Fig. 11a, b), consistent with our finding that ICE1 is a core component of CBs (Fig. 7c). Knockdown of MED26, ELL or EAF1 moderately decreased the formation of CBs, but knockdown of SEC components, AF4 or AFF4, did not (Supplementary Fig. 11a, b). Notably, knockdown of EAF2 hardly affected the formation of CBs (Supplementary Fig. 11a, b), consistent with our previous finding that a large population of LEC copurified with EAF1 but not with EAF2[22]. In contrast, knockdown of MED26 or the components of LEC did not affect the formation of HLBs (Supplementary Fig. 11c, d). These results suggest that the factors that colocalize closer to CBs contribute more to CB formation.

We found that, first, Mediator localized between HLBs and CBs and, second, the interaction between Mediator and LEC was required for CBs' association with HLBs, so we next tested whether MED26 contributes to CBs' association with HLBs. As shown in Supplementary Fig. 12a, b, knockdown of MED26 decreased CBs' association with HLBs. As members of the EAF protein family directly bind to MED26 and play roles as adaptors for LEC's interaction with Mediator[65], we next tested whether knockdown of EAF1 or EAF2 also affects CBs' association with HLBs. As shown in Supplementary Fig. 12a, b, knockdown of EAF1 decreased CBs' association with HLBs. In contrast, we observed only a small effect on CBs' association with HLBs upon knockdown of EAF2 (Supplementary Fig. 12a, b), consistent with our previous findings that much more EAF1 than EAF2 was included in LEC in cells[22]. As expected, knockdown of both EAF1 and EAF2 more strongly decreased the association of CBs and

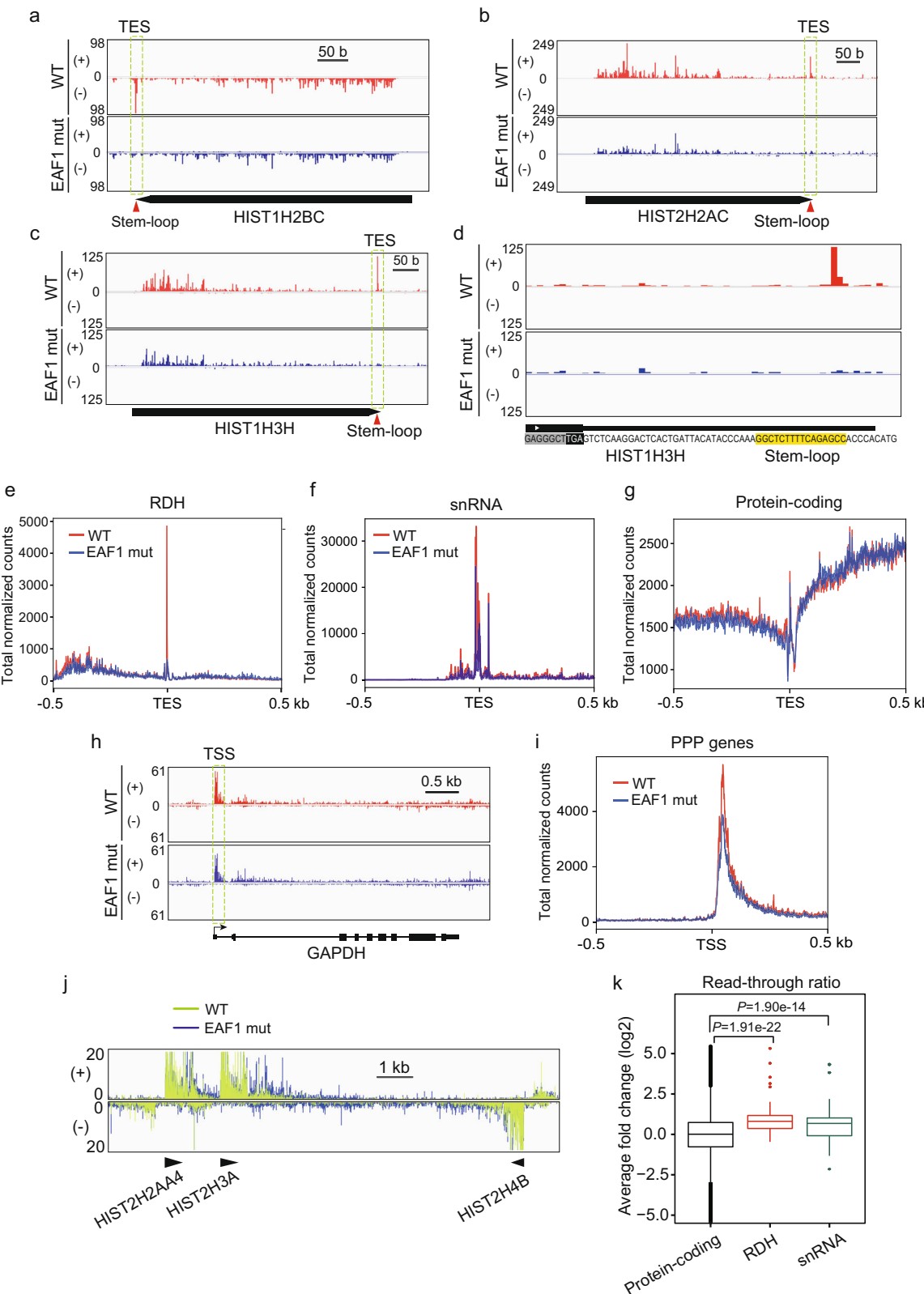

HLBs than knockdown of ELL (Supplementary Fig. 12a, b). These results support the idea that Mediator's interaction with LEC contributes to CBs' association with HLBs.

**Binding of CBs to RDH gene clusters is decreased in EAF1-mutant cells**. It has been shown that CBs are sites for various co-transcriptional events such as mRNA processing[29]. As noted

above, we found that TPP and subsequent 3′-end processing of RDH genes were defective in EAF1-mutant cells in which CBs' association with HLBs was decreased, raising the possibility that 3′-end processing factors for RDH gene transcripts are supplied from CBs through their association with HLBs. To evaluate how much CBs associate with histone gene loci in HLBs, we developed the novel technique of "in situ biotin-labeling of protein with

**Fig. 5 TPP at RDH and snRNA genes was diminished in EAF1-mutant cells. a–c** PRO-seq revealed that TPP at RDH genes was decreased in EAF1-mutant cells. Genome browser tracks showing the distribution of PRO-seq reads at *HIST1H2BC* (**a**), *HIST2H2AC* (**b**), and *HIST1H3H* (**c**) in wild-type (WT) and EAF1-mutant (EAF1 mut) cells. The positive and negative strands of DNA are indicated by (+) and (−), respectively. **d** Detailed genome browser tracks showing the distribution of PRO-seq reads at the transcript end site of the *HIST1H3H* gene in wild-type (WT) and EAF1-mutant (EAF1 mut) cells. The stem-loop sequence located immediately upstream of TES of the *HIST1H3H* gene is highlighted in yellow. **e–g** Meta-gene analysis of PRO-seq reads around TESs of RDH (**e**), snRNA (**f**), and protein-coding genes (**g**). **h** Genome browser tracks showing the distribution of PRO-seq reads at *GAPDH*. **i** Meta-gene analysis of PRO-seq reads around TSSs of PPP genes. **j** Overlaid genome browser tracks of PRO-seq reads showing the Pol II read-through around histone gene cluster 2. The relative position and direction of each RDH gene are represented by arrowheads. The positive and negative strands of DNA are indicated by (+) and (−), respectively. **k** Boxplots representing the PRO-seq read-through ratio of protein-coding genes (*n* = 87,796, black), RDH genes (*n* = 140, red) and snRNA genes (*n* = 256, green). Read-through ratio was determined using transcripts around TES, which have more than 10-fold PRO-seq signals normalized by spike-in control. The center line of each boxplot represents the median. Upper and lower fences of each boxplot represent upper and lower quartiles, respectively. The *P*-values were determined by two-sided Wilcoxon's signed-rank test. Source data are provided as a Source Data file.

high-throughput DNA sequencing" using an antibody-based in situ biotinylation technique[85]. We then used this technique to identify the genomic regions included in CBs. The proteins present within approximately 20 nm from Coilin were labeled with biotin by an anti-Coilin antibody–based biotinylation reaction in cells (Supplementary Fig. 13a, b) and then biotin-labeled proteins with genomic DNA were purified, followed by high-throughput sequencing (Fig. 8a). Two RDH gene loci and snRNA/snoRNA gene loci were strongly detected as the genes present in the proximity of Coilin (Fig. 8b–e), indicating that the CB association with two RDH gene loci and snRNA/snoRNA gene loci were successfully detected by this method. This result is consistent with the recent genome-wide 4C-seq analysis of CBs indicating that CBs are associated with these gene loci[57]. Intriguingly, the interaction of Coilin with RDH gene loci was dramatically attenuated in EAF1-mutant cells and ICE1- or MED26-knockdown cells (Fig. 8c, d, and Supplementary Fig. 13c, d), suggesting that Mediator's interaction with LEC contributes to CB association with RDH gene loci. In contrast, we found that CBs association with snRNA/snoRNA gene loci was only mildly decreased in EAF1-mutant cells compared with CB association with RDH gene loci (Fig. 8d, e), consistent with the results that the number of CBs, TPP at snRNA genes and read-through of Pol II at snRNA genes were mildly affected in EAF1-mutant cells (Figs. 5f, k and 6c). LSM11 is one of the components of U7 snRNP, which plays an essential role in 3′-end processing of RDH gene transcripts. As we found that LSM11 localized between CBs and HLBs similarly to Mediator (Fig. 7e), we performed an antibody-based in situ biotinylation technique with anti-LSM11 antibody to test whether EAF1 mutation affected LSM11's association with RDH gene loci. As shown in Fig. 8f, g, and Supplementary Fig. 13e, this association was also reduced in EAF1-mutant cells. These results suggest that the recruitment of U7 snRNP, which is enriched in CBs, to RDH gene loci was impaired according to the decrease in CBs' association with HLBs in EAF1-mutant cells. Moreover, decreased association of CBs with RDH loci was accompanied by significant decrease in the expression of RDH genes and significant loss of TPP at RDH genes (Fig. 8h and Supplementary Fig. 13f). Considering that TPP was diminished more strikingly at the RDH genes than at the snRNA genes in EAF1-mutant cells (Fig. 5e, f), these results indicate that CBs' association with HLBs is essential for RDH gene transcription and TPP at RDH genes.

A recent study showed that cyclin-dependent kinase 11 (CDK11) binds to the 3′ region of RDH gene transcripts, just before the stem-loop, and plays an important role in the 3′-end processing of those transcripts through Ser2 phosphorylation of the Pol II CTD[86]. Since we found that CDK11 colocalized at CBs (Fig. 8i), we took advantage of the individual-nucleotide-resolution UV crosslinking and immunoprecipitation (iCLIP)–seq data of Coilin and CDK11 and compared them at the RDH genes[45,86]. This analysis revealed that both Coilin and

CDK11 bind to the region immediately upstream of the stem-loop of RDH gene transcripts (Fig. 8j). This raises the possibility that CBs associate with the nascent RDH gene transcripts synthesized by Pol II, which pauses within the stem-loop region of RDH genes.

**Interaction between two RDH gene clusters is abolished in EAF1-mutant cells.** A recent study has shown that the formation of CBs plays a role in higher-order chromosome conformation[29]. It has been shown that CBs are required for the spatial clustering of snRNA/snoRNA and RDH gene loci, and they are thought to be involved in regulating their expression as well as mRNA processing[29,57]. We examined the spatial positions of RDH gene cluster 1 present at chromosome 6 and RDH gene cluster 2 present at chromosome 1 in CBs and HLBs. We performed DNA-FISH followed by immunofluorescence of Coilin and NPAT. Coilin and NPAT were colocalized at two RDH gene clusters (Fig. 9a, b). Notably, the frequency of Coilin's association with RDH gene cluster 1 or RDH gene cluster 2 was decreased in EAF1-mutant cells (Fig. 9a). The association between RDH gene cluster 1 and RDH gene cluster 2 was also decreased in EAF1-mutant cells (Fig. 9b). In contrast, NPAT's association with these two RDH gene loci was not affected in EAF1-mutant cells (Supplementary Fig. 14a). This is consistent with the results showing that knockdown of a series of factors present at CBs including MED26 and components of LEC did not affect the formation of HLBs (Supplementary Fig. 11c, d). Furthermore, we performed DNA-FISH of RDH gene cluster 1 followed by immunofluorescence of both Coilin and WRAP53. As shown in Supplementary Fig. 14b, c, RDH gene cluster 1 colocalized with the particles containing both Coilin and WRAP53 in wild-type cells. In contrast, we observed decreased colocalization between RDH gene cluster 1 and the particles containing both Coilin and WRAP53 in EAF1-mutant cells. These results also support the idea that the Coilin particles that we observed in wild-type and EAF1-mutant cells contain the components required for CB function and that Mediator's interaction with LEC is required for CBs' association with RDH gene loci, but not for HLBs' association with the loci. These results showed that CBs' association with two RDH gene clusters and the inter-chromosome association between RDH gene cluster 1 and RDH gene cluster 2 were decreased in EAF1-mutant cells. Thus, our data indicate that, first, Mediator's interaction with LEC plays a role in bringing CBs to RDH gene loci and, second, CBs' association with RDH gene loci is also required for an appropriate inter-chromosome structure between two RDH gene clusters.

To further confirm that Mediator's interaction with LEC contributes to the inter-chromosome association between two RDH gene loci, we performed 4C-seq and analyzed the chromosome conformations in EAF1-mutant cells. We used *HIST1H2BK* and *HIST2H2AB* as representatives of RDH gene

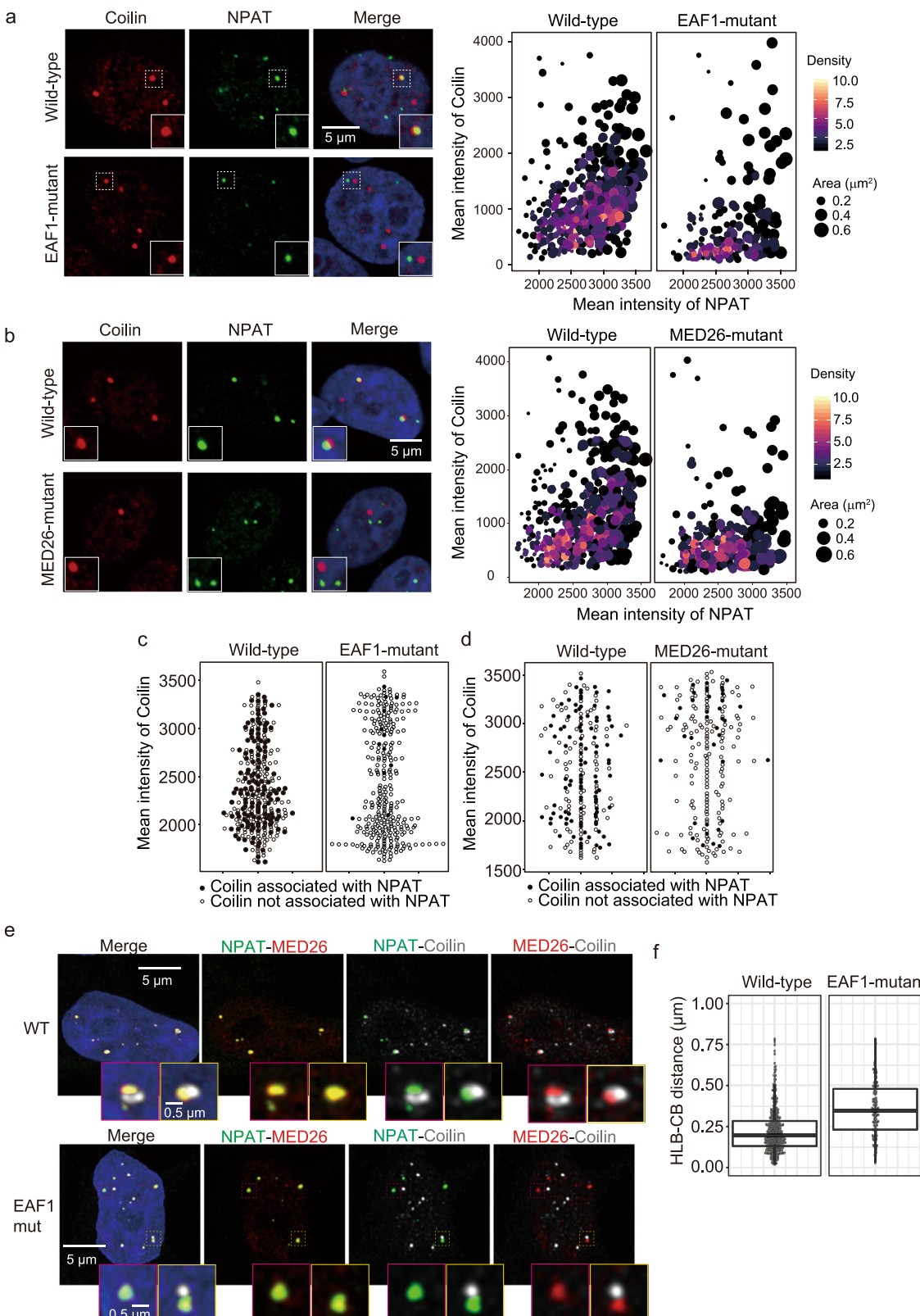

clusters 1 and 2, respectively. For the *HIST1H2BK* gene locus, we observed inter-chromosome interaction with histone gene cluster 2, as well as intra-chromosome interaction in histone gene cluster 1 (Fig. 9c). Similarly, for the *HIST2H2AB* gene locus, we observed inter-chromosome interaction with histone gene cluster 1, as well as intra-chromosome interaction in histone gene cluster 2 (Fig. 9e). Intriguingly, inter-chromosome interaction between the two histone gene clusters was decreased in EAF1-mutant cells (Fig. 9c–f), however; intra-chromosome interaction in each of the two histone gene clusters was slightly affected in EAF1-mutant cells (Fig. 9c–f). Taken together, these results suggest that Mediator's interaction with LEC and CBs' resulting association with HLBs were more critical for appropriate inter-chromosome association between the two RDH gene loci than intra-

**Fig. 6 Decreased association of CBs with HLBs in EAF1-mutant cells. a, b** EAF1 C-terminal region and MED26 N-terminal region are required for CB–HLB association. Wild-type cells and EAF1-mutant cells (**a**) or MED26-mutant cells (**b**) were fixed with paraformaldehyde, and immunofluorescence staining for Coilin and NPAT was performed. Scale bar, 5 μm. Enlarged images for representative particles are shown in the lower part of each image. Evaluation of CB–HLB association in each mutant cell line is shown in the right panels. The NPAT particles were extracted from images. The area of NPAT particles occupied by Coilin particles and the intensity of the particles of NPAT and Coilin were calculated. The density shows the degree to which dots overlap with others. Signal intensities were evaluated using more than 200 nuclei for each condition. Wild-type, $n = 308$; EAF1 mutant, $n = 204$ in **a**, and wild-type, $n = 278$; MED26 mutant, $n = 318$ in **b. c, d** Quantification of the number and intensity of Coilin particles. All detected Coilin particles in EAF1-mutant cells (EAF1-mutant) (**c**) or MED26-mutant cells (MED26-mutant) (**d**) are plotted. NPAT-associating Coilin particles are indicated by black dots and free Coilin particles are indicated by white dots. Quantification was performed using 300 (**c**) or 200 (**d**) Coilin particles. **e** Super-resolution images showing relative positions of HLBs, Mediator, and CBs. Wild-type cells (WT, upper) and EAF1-mutant cells (EAF1 mut, lower) were fixed with paraformaldehyde, and triple immunofluorescence staining for NPAT (green), MED26 (red), and Coilin (gray) was performed. Enlarged images for representative particles are shown in the lower part of each image. **f** Calculation of the distance between HLBs and CBs. Immunofluorescence staining for NPAT and Coilin was performed with wild-type and EAF1-mutant HEK293T cells. For Coilin-associated NPAT particles, the distance between the center of the NPAT particle and Coilin particle was measured, and the distance between each associating particle is displayed as boxplots. The center line of each boxplot represents the median. Upper and lower fences of each boxplot represent upper and lower quartiles, respectively. Wild-type, $n = 465$ particles from 301 nuclei; EAF1 mutant, $n = 145$ particles from 354 nuclei. Source data are provided as a Source Data file.

chromosome association in each of the two RDH gene loci, consistent with a previous report showing that CBs are involved in higher-order chromosome conformation[57].

On the basis of our findings, we propose a model for the role of Mediator's interaction with LEC–CBC–NELF in TPP and subsequent 3′-end processing of RDH genes. In this model, MED26-containing Mediator plays a role in the association of CBs with HLBs through interaction with LEC–CBC–NELF. CBs' association with HLBs leads to TPP at RDH genes and subsequent 3′-end processing by supplying 3′-end processing factors from CBs (Fig. 10).

## Discussion

In this study, we found 3′ Pol II pausing at the transcript end site (TES) proximal region of non-polyadenylated genes, including RDH, snRNA, and snoRNA genes. We found Pol II paused immediately upstream of the TES of RDH and snoRNA genes; in contrast, it paused at multiple sites around TES in snRNA genes (Fig. 1). Notably, Pol II pauses within the stem-loop region of RDH genes, indicating that it pauses before the stem-loop structure is formed at RDH genes (Fig. 1). We called such pausing of Pol II "TES proximal pausing" (TPP). Since we previously showed that MED26 plays a role in transcription termination of non-polyadenylated genes including RDH and snRNA genes through the recruitment of LEC[22], we tested the possibility that Mediator's interaction with LEC contributes to this TPP and subsequent 3′-end processing of non-polyadenylated gene transcripts. We generated an EAF1-mutant cell line and found that Mediator's interaction with LEC was specifically abolished in it, but that of SEC was not (Fig. 3). Consistent with the previous finding that LEC interacts with CBC–NELF in cells[22], Mediator's interaction with CBC-NELF was also abolished in EAF1-mutant cells (Fig. 3). Intriguingly, EAF1 mutation interfered with TPP at the RDH and snRNA genes (Fig. 5), resulting in increased levels of aberrant unprocessed transcripts of these genes (Fig. 4). In addition, knockdown of NELF or CBC abolished TPP (Fig. 2), indicating that Mediator's interaction with LEC–CBC–NELF is required for TPP.

Several lines of evidence have indicated that Pol II is arrested or pauses downstream of TESs of RDH genes[13–16]. Since PRO-seq enables precise detection of the 3′ end of nascent transcripts, our results do not contradict previous reports based on in vitro transcription assay, ChIP-seq and GRO-seq. PRO-seq analysis revealed that Pol II pauses immediately upstream of TESs of RDH and snRNA genes (Fig. 1). NELF, which is responsible for the PPP of Pol II, has been shown to colocalize with CBs and plays a role in the 3′-end processing of RDH genes[21]. Intriguingly, TPP

was decreased in EAF1-mutant cells or in cells in which CBP80 or NELF was depleted (Figs. 2 and 5). These results raised the possibility that TPP plays a critical role in the 3′-end processing of RDH and snRNA genes. Although the mechanism by which NELF regulates the 3′-end processing of RDH gene transcripts has not been fully elucidated[21], our results raise the possibility that Mediator's interaction with LEC–CBC–NELF is required for TPP and subsequent 3′-end processing of RDH and snRNA genes. It has been shown that PPP is a key checkpoint for gene expression through regulating the transition from transcription initiation to productive elongation[87]. Our results raise the possibility that TPP is a key checkpoint process regulating the transition from transcription elongation to transcription termination (Fig. 10). A recent report showed that CDK11 binds immediately upstream of the stem-loop of RDH gene transcripts and plays an essential role in the recruitment of 3′-end processing factors to RDH genes through phosphorylation of Ser2 of Pol II CTD[86]. In addition, we found that CDK11 also colocalized with CBs (Fig. 8) and that Coilin binds to the same region immediately upstream of the stem-loop of RDH gene transcripts by analyzing the iCLIP data[45] (Fig. 8). This raises the possibility that CBs associated with HLBs include the nascent RDH gene transcripts synthesized by Pol II, which pauses immediately upstream of the TES. Considering that Ser2 phosphorylation of Pol II CTD is decreased at RDH genes in EAF1-mutant cells (Supplementary Fig. 6), it is possible that TPP allows sufficient Ser2 phosphorylation of Pol II CTD by CDK11 to recruit 3′-end processing factors to RDH genes (Fig. 10). The detailed mechanism by which the transcription termination of RDH genes is regulated by TPP should be elucidated in future studies. In addition, recent reports showed that protein phosphatase 1 (PP1) or protein phosphatase 2A (PP2A) dephosphorylates Spt5, a component of DSIF, and/or Pol II CTD to decelerate Pol II elongation downstream of the TES of RDH or snRNA genes, respectively[17–20]. These reports suggest that deceleration of Pol II beyond TES contributes to the final process of transcription termination through facilitating 5′→3′ degradation of RNAs associated with Pol II.

It has been shown that CBs associate with HLBs, but the mechanism by which CBs frequently associate with HLBs containing RDH gene loci and regulate their transcription has remained unclear. Intriguingly, the components of Mediator, MED1 and MED26, and the component of 3′-end processing factors, LSM11, localized between CBs and HLBs. In contrast, the components of LEC, ELL and ICE1, mainly colocalized at CBs and the component of 3′-end processing factors, FLASH, mainly colocalized at HLBs (Fig. 7). We employed STED super-resolution microscopy and found that the Mediator

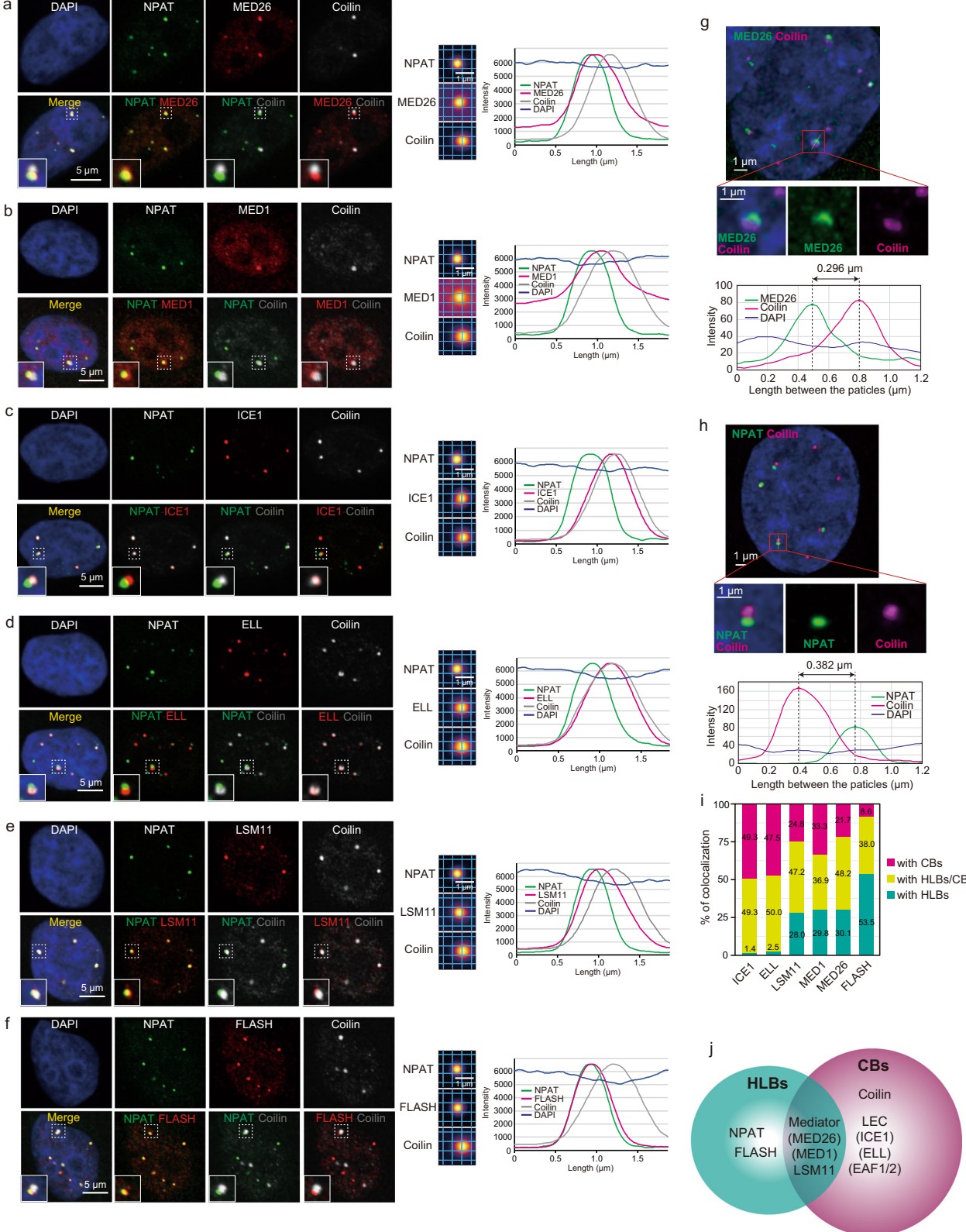

components, MED26 and MED1, localized between CBs and HLBs (Fig. 7). Intriguingly, we found that EAF1 mutation interfered with the frequency of association between CBs and HLBs, while the number of CBs was not decreased (Fig. 6). These results support our idea that Mediator's interaction with LEC contributes to CBs' association with HLBs. Detection of the genomic region associating with CBs using an in situ

biotinylation approach revealed that CBs' association with RDH gene loci was drastically decreased in EAF1-mutant cells (Fig. 8). Moreover, 4C-seq analysis revealed that inter-chromosome interaction between RDH gene clusters was disrupted in EAF1-mutant cells (Fig. 9). Thus, point mutation of EAF1 that interferes with Mediator's interaction with LEC–CBC–NELF resulted in decreased CB–HLB association, leading to the impaired TPP and

**Fig. 7 Mediator localized in the region between CBs and HLBs. a–f** HeLa cells were fixed with paraformaldehyde and subjected to triple immunofluorescence staining. MED26 (red), MED1 (red), ICE1 (red), ELL (red), LSM11 (red), and FLASH (red) were co-stained with NPAT (green) and Coilin (gray) using specific antibodies. Scale bar, 5 μm. Averaged signals of immunofluorescence centered at NPAT signal and respective line plots for each immunofluorescence experiment are shown on the right. Scale bar, 1 μm. **g, h** HCT116 cells were stained with anti-MED26 (green), anti-NPAT (green), and anti-Coilin (magenta) antibodies, and the nuclear positions of MED26 (Mediator), NPAT (HLBs), and Coilin (CBs) were observed by stimulated emission depletion (STED) microscopy. Scale bar, 1 μm. Line intensity plots across the particles of NPAT (green), Coilin (magenta), and MED26 (magenta) are shown in the lower panels. **i** Ratios of ICE1, ELL, LSM11, MED1, MED26, or FLASH colocalization with CBs, HLBs, or both CBs and HLBs. The ratio was calculated using more than 100 nuclei. ICE1, $n = 71$ particles from 196 nuclei; ELL, $n = 122$ particles from 240 nuclei; LSM11, $n = 161$ particles from 242 nuclei; MED1, $n = 84$ particles from 237 nuclei; MED26, $n = 83$ particles from 204 nuclei; FLASH, $n = 187$ particles from 291 nuclei. Source data are provided as a Source Data file. **j** Model of the relative position of each protein at CBs and HLBs. Mediator was located between HLBs and CBs.

subsequent 3′-end processing of RDH gene transcripts. On the basis of our findings, we propose a novel model in which CBs are recruited to HLBs through the interaction of MED26-containing Mediator and LEC to facilitate TPP and subsequent 3′-end processing of RDH genes (Fig. 10). Importantly, a number of 3′-end processing factors for RDH genes, including the components of NELF, U7, snRNP, and HCC are enriched in CBs or were biochemically copurified with Coilin. It was also shown that U2 snRNP, which is present at CBs, binds to pre-mRNAs of RDH genes to facilitate U7 snRNP-dependent 3′-end processing[56]. It has been shown that NPAT and FLASH colocalize at HLBs and play an essential role in RDH genes transcription[34,88]. Recently, it was shown that HLBs have a core-shell structure in which the internal core contains RDH genes that are actively transcribed. The N-terminus of Mxc, which is a fly homolog of human NPAT, is enriched in the HLB core, and the C-terminus of Mxc is enriched in the HLB outer shell where FLASH is enriched through direct interaction with the C-terminus of Mxc[89]. Thus, it was speculated that U7 snRNP interacting with FLASH translocates from shell to core to process the nascent RDH pre-mRNAs. Consistent with this report, we found that FLASH mainly colocalized at HLBs; in contrast, LSM11 localized between CBs and HLBs (Fig. 7). This previous report and our findings raise the possibility that U7 snRNP containing LSM11 localizes outside the outer shell of HLBs, which is close to CBs, and translocates to the core of HLBs to process the nascent RDH gene transcripts. Considering these previous findings and our results, it is highly possible that 3′-end processing factors are supplied from CBs to HLBs.

We observed changes in chromosome conformation in EAF1-mutant cells, in which most of the CBs are dissociated from HLBs. These results suggest that CBs' association with HLBs is required for the higher-order chromosome structure of histone gene clusters, consistent with previous reports showing that CBs organize the genome-wide clustering of RDH genes[57]. Intriguingly, decreased association of CBs with RDH loci was accompanied by significant decreases of RDH gene expression and significant loss of TPP at RDH genes in EAF1-mutant cells (Fig. 8 and Supplementary Fig. 13). These results suggest that CBs' association with HLBs is required for RDH gene transcription and TPP at RDH genes through higher-order chromosome structure at RDH genes.

Many recent studies have shown that membrane-less organelles such as nuclear bodies and nuclear speckles are formed by mechanisms of liquid–liquid phase separation (LLPS)[90]. Recently, increasing evidence has shown that LLPS is involved in many of the nuclear functions, including gene transcription, RNA processing, and DNA repair[91–93]. It is thought that membrane-less organelles are formed by weak multivalent interactions among proteins containing a low-complexity region or intrinsically disordered region (IDR)[90]. Considering that IDRs constitute large parts of both Coilin and NPAT, it is highly possible that LLPS is involved in the formation of CBs and HLBs. Recently, it has been demonstrated that NPAT phosphorylation by Cyclin E/CDK2

regulates LLPS-dependent HLB formation and the 3′-end processing of RDH gene transcripts[41]. Moreover, it has been reported that, first, CBs undergo fusion and fission, which are key features of LLPS-mediated droplets[94], and that, second, CBs are sensitive to aliphatic alcohol 1,6-hexanediol[95–97], suggesting that LLPS is involved in CB formation. Furthermore, the human Mediator subunit MED1 has been shown to play a role in super-enhancer formation through LLPS[98]. Considering our results showing that MED26-containing Mediator is involved in the association of CBs with HLBs, it is possible that Mediator contributes to the fusion of CBs with HLBs through LLPS.

In this study, we propose a model in which LEC recruitment by MED26-containing Mediator plays a role in CBs' association with HLBs to facilitate the 3′-end processing of RDH genes. However, the trigger inducing CBs' association with HLBs has not been elucidated. It is speculated that cell cycle-dependent events contribute to the Mediator-dependent association of CBs with HLBs. NPAT is phosphorylated at the $G_1$/S transition by cell cycle-dependent kinase, CDK2/Cyclin E, resulting in the formation of HLBs at RDH gene loci in an LLPS-dependent manner[41]. Considering that the MED26-binding site of EAF1 contains multiple serine residues, it is possible that the cell cycle-dependent phosphorylation of EAF1 leads to the interaction of LEC with Mediator and triggers CBs' association with HLBs. Although further studies are needed to demonstrate this, it is reasonable that the strictly controlled induction of synthesis of RDH mRNAs is switched on and off in a cell cycle-dependent manner[89,99,100].

## Methods

**Cell culture and generation of cell lines.** Human embryonic kidney 293T cells and HeLa cells were cultured in Dulbecco's modified Eagle's medium (08458-45; Nacalai Tesque, Kyoto, Japan), and HCT116 cells were cultured in McCoy's 5A medium (16600-082; Gibco, Grand Island, NY), supplemented with 10% fetal bovine serum and penicillin/streptomycin under 5% $CO_2$ at 37 °C. The EAF1-point-mutant HEK293T cell line was generated by a CRISPR-mediated knock-in strategy. sgRNA complementary to exon 6 of the endogenous EAF1 gene was chemically synthesized, phosphorylated using T4 polynucleotide kinase (0201S; NEB, Ipswich, MA) and ligated into the BbsI site downstream of the U6 promoter in pX330 plasmids (72833; Addgene, Watertown, MA). pX330 plasmids containing sgRNA were transfected into HEK293T cells with single-stranded DNA containing the sequence needed for knock-in of the EAF1 point mutant. Each cell clone was screened by genomic PCR and DNA sequencing, and, in the resulting EAF1-point-mutant cell line, E/2-24, both alleles of the EAF1 gene had mutations, leading to replacement of the amino acids from positions 262 to 265 by alanines. Oligonucleotide sequences for the guide RNAs and single-stranded DNA are shown in Supplementary Information.

**siRNA transfection.** HeLa cells in 12-well tissue culture plates (~$6 \times 10^4$ cells per well) or 6 cm dishes (~$1.5 \times 10^5$ cells per dish) were transfected with 50 nM siRNA targeting human MED26 (ON-TARGET plus SMART pool, L-011948-00; Dharmacon, Pittsburgh, PA), siRNA targeting human ICE1 (ON-TARGET plus SMART pool, L-024272-02; Dharmacon), siRNA targeting human AFF4 (ON-TARGET plus SMART pool, L-020276-00; Dharmacon), siRNA targeting human AF4 (ON-TARGET plus SMART pool, L-020074-02; Dharmacon), siRNA targeting human EAF1 (ON-TARGET plus SMART pool, L-019284-01; Dharmacon), siRNA targeting human EAF2 (ON-TARGET plus SMART pool, L-006313-00; Dharmacon), siRNA targeting human ELL (ON-TARGET plus SMART pool, L-008176-00; Dharmacon), siRNA targeting human CBP80 (ON-TARGET plus SMART pool, L-019672-00; Dharmacon), siRNA targeting human NELF-E (ON-

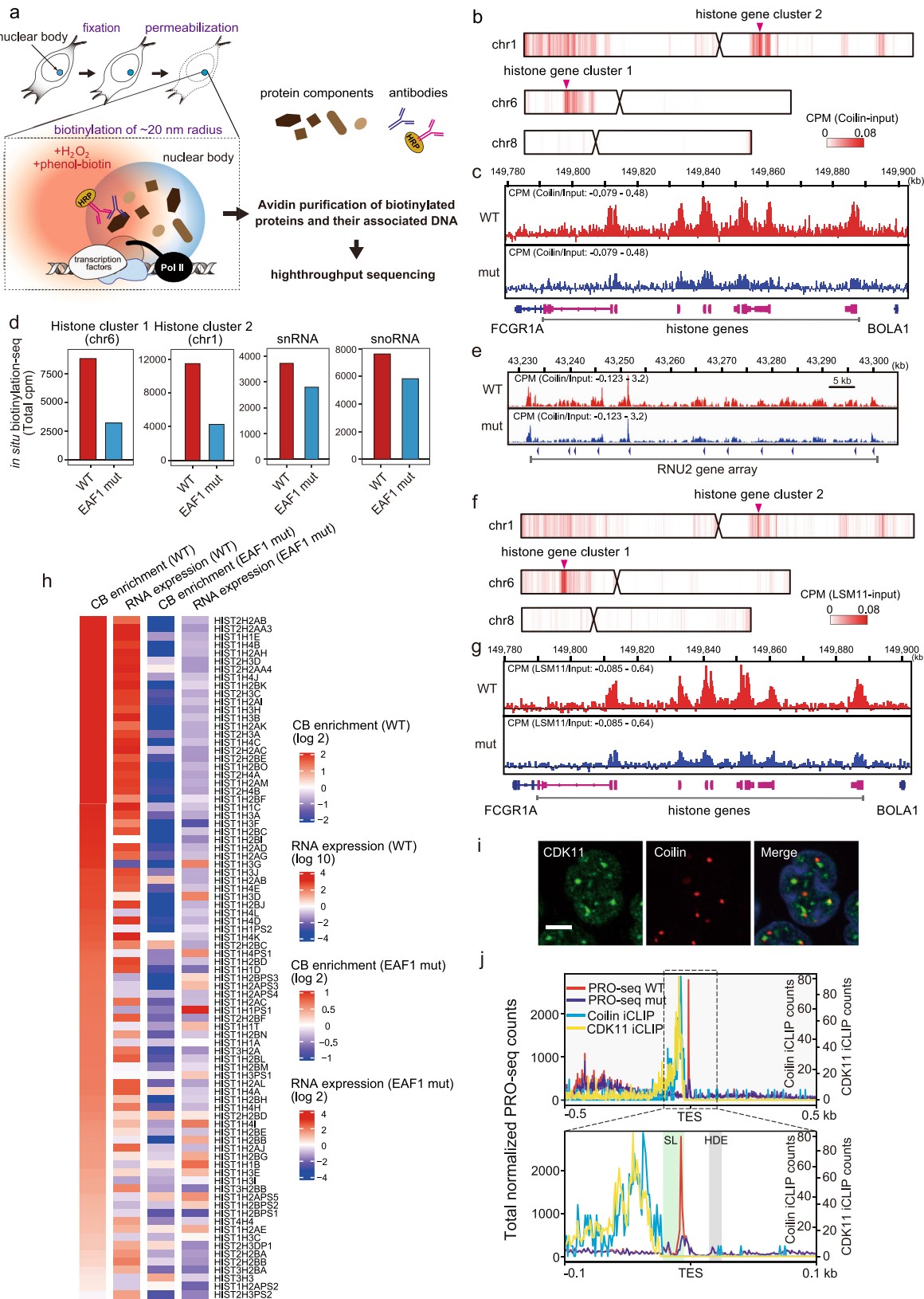

TARGET plus SMART pool, L-011761-01; Dharmacon) or with 50 nM siGEN-OME NON-TARGETING siRNA Pool #2 (D-001206; Dharmacon), using Lipofectamine$^{TM}$ RNAiMAX Transfection Reagent (Invitrogen, Carlsbad, CA).

**Production of recombinant proteins**. N-terminally 6×His- and FLAG-tagged ICE2 and N-terminally 6×His- and HA-tagged ICE1, ELL and EAF1 wild-type or

mutant were subcloned into pFastBac HTb (Life Technologies, Carlsbad, CA) and expressed together with the BAC-to-BAC expression system (Takara Bio Inc., Kusatsu, Japan).

**Western blotting**. Proteins were separated by SDS-PAGE, transferred to an Immobilon-P PVDF membrane (Millipore, Burlington, MA) and detected by ECL

**Fig. 8 Decreased CB association with histone gene cluster in EAF1-mutant cells. a** Schematic illustration of the strategy for in situ biotinylation of nuclear bodies. Cells were fixed and permeabilized, and HRP-conjugated antibodies were deposited onto target proteins (Coilin or LSM11). Then, the proximal chromatin was biotinylated by adding biotin–phenol and $H_2O_2$. After cell lysis, biotinylated proteins and their associated DNA were avidin-purified, followed by qPCR or high-throughput DNA sequencing. **b–d** CBs were dissociated from RDH gene clusters in EAF1-mutant cells. **b** Heatmap showing the CB (Coilin)-association profile in wild-type HEK293T cells. The avidin-purified DNA was analyzed by high-throughput sequencing. Two RDH gene clusters located at chromosome 1 and chromosome 6 are indicated by arrowheads. **c** Genome browser tracks showing the distribution of in situ biotinylation-seq reads using anti-Coilin antibodies at RDH gene cluster 2 in wild-type (WT) and EAF1-mutant (mut) cells. **d** Total counts of Coilin in situ biotinylation-seq reads at RDH genes, snRNA genes and snoRNA genes, and CB enrichment levels were compared between wild-type (WT) and EAF1-mutant (EAF1 mut) cells. Each value is the mean of two independent experiments. Source data are provided as a Source Data file. **e** Genome browser tracks showing the distribution of in situ biotinylation-seq using anti-Coilin antibodies at snRNA gene RNU2 in wild-type (WT) and EAF1-mutant (mut) cells. **f** Heatmap showing the U7 snRNP (LSM11)-association profile in wild-type HEK293T cells. The avidin-purified DNA was analyzed by high-throughput sequencing. Two RDH gene clusters located at chromosome 1 and chromosome 6 are indicated by arrowheads. **g** Genome browser tracks showing the distribution of in situ biotinylation-seq reads using anti-LSM11 antibodies at RDH gene cluster 2 in wild-type (WT) and EAF1-mutant (mut) cells. **h** Heatmap showing CB enrichment and mRNA expression levels of RDH genes in wild-type (WT) HEK293T cells, and fold-change of CB enrichment and mRNA expression levels of RDH genes in EAF1-mutant (EAF1 mut) cells. **i** Immunofluorescence images showing CDK11 localization at CBs. HCT116 cells were fixed with methanol and subjected to immunofluorescence staining. CDK11 (green) and Coilin (red) were stained using specific antibodies. Scale bar, 10 μm. **j** Meta-gene analysis of PRO-seq reads around TESs. Coilin (light blue) and CDK11 (yellow) iCLIP data by Machyna et al.[45] and Sathyan et al.[68] are overlaid with PRO-seq data. Approximate positions of stem-loop (SL) and HDE are indicated in the lower panel.

Western Blotting Detection Reagents (GE Healthcare, Chicago, IL). The following primary antibodies were used: anti-EAF1 antibodies (1:200 dilution, sc-398450; Santa Cruz Biotechnology, Inc., Santa Cruz, CA), anti-MED26 antibodies (D4B1X, 1:1000 dilution, 14950; Cell Signaling Technology, Danvers, MA), anti-ICE1 antibodies (1:1000 dilution, HPA054452; Sigma-Aldrich Corp., St. Louis, MO), anti-ELL antibodies (1:1000 dilution, 14468; Cell Signaling Technology), anti-MED1 antibodies (1:1000 dilution, ab64965; Abcam, Cambridge, UK), anti-MED23 antibodies (1:1000 dilution, A300-425A; Bethyl Laboratories, Montgomery, TX), anti-Rpb1-NTD antibodies (1:2000 dilution, 14958; Cell Signaling Technology), anti- phospho-Rpb1-CTD (Ser5) antibodies (1:2000 dilution, 13523; Cell Signaling Technology), anti-phospho-Rpb1-CTD (Ser7) antibodies (1:2000 dilution, 13780; Cell Signaling Technology), anti-RNA polymerase II-CTD (phospho-2) antibodies (1:1000 dilution, ab5095; Abcam), anti-Coilin antibodies (1:2000 dilution, 14168; Cell Signaling Technology), anti-NPAT antibodies (1:300 dilution, sc-136007; Santa Cruz Biotechnology), anti-LSM11 antibodies (1:500 dilution, HPA039587; Sigma-Aldrich Corp.) and anti-β-actin antibodies (1:2000 dilution, sc-47778; Santa Cruz Biotechnology, Inc.).

**Immunoprecipitation and affinity purification.** Protein complexes were purified from nuclear extract fractions of cell lines stably expressing FLAG-tagged MED26 using anti-FLAG M2 agarose (E2220; Sigma-Aldrich Corp.), as described previously[22]. Briefly, nuclear extracts and S100 fractions were prepared in the presence of Benzonase® Nuclease (E8263; Sigma-Aldrich Corp.), basically in accordance with the method of Dignam et al., from wild-type or EAF1-mutant HEK293T cells stably expressing FLAG-tagged MED26[101]. Each of the nuclear extracts was incubated with anti-FLAG agarose beads for 2 h at 4 °C. The beads were washed five times with a 100-fold excess of a buffer containing 50 mM HEPES-NaOH (pH 7.9), 0.15 M NaCl, 0.1% Triton X-100 and 10% (v/v) glycerol, and then eluted with 100 μl of a buffer containing 0.1 M NaCl, 50 mM HEPES-NaOH (pH 7.9), 0.05% Triton X-100, 10% (v/v) glycerol and 0.25 mg/ml FLAG peptide. The eluates were subjected to western blotting.

**Chromatin immunoprecipitation (ChIP) assay.** Crosslinking was performed with 1% formaldehyde for 7 min. Crosslinked HEK293T cells were washed with ice-cold PBS and then resuspended with hypotonic buffer [10 mM Hepes-KOH (pH 7.8), 10 mM KCl, 0.1 mM EDTA and 0.1% NP-40] containing protease inhibitors and pelleted by centrifugation. The nuclear pellet was resuspended with ChIP Lysis Buffer [0.5% SDS, 10 mM EDTA and 50 mM Tris-HCl (pH 8.0)] and subjected to mild sonication using Bioruptor Sonicator (Diagenode, Denville, NJ). The lysate was diluted 10-fold using ChIP Dilution Buffer [1% Triton X-100, 1.2 mM EDTA, 167 mM NaCl and 16.7 mM Tris-HCl (pH 8.0)]. The lysates containing 25 μg of DNA per reaction were incubated with specific antibodies overnight at 4 °C with rotation. Antibodies used for ChIP assays were as follows: anti-ELL (14468; Cell Signaling Technology), anti-RPB1 (D8L4Y, 14958; Cell Signaling Technology), anti-phospho-RPB1 CTD (Ser2) (ab5095; Abcam), anti-phospho-RPB1 CTD (Ser5) (D9N5I, 13523; Cell Signaling Technology), anti-phospho-RPB1 CTD (Ser7) (E2B6W, 13780; Cell Signaling Technology) and normal rabbit IgG antibodies (PM035; MBL). Thirty microlitres of Dynabeads M280 Sheep Anti-Rabbit IgG (Invitrogen) was added to each sample and further incubated for 3 h at 4 °C with rotation. Beads were washed once with low-salt buffer [0.1% SDS, 1% Triton X-100, 2 mM EDTA, 150 mM NaCl and 20 mM Tris-HCl (pH 8.0)], once with high-salt buffer [0.1% SDS, 1% Triton X-100, 2 mM EDTA, 500 mM NaCl and 20 mM Tris-HCl (pH 8.0)], once with LiCl buffer [0.25 M LiCl, 1% NP-40, 1% sodium deoxycholate, 1 mM EDTA and 10 mM Tris-HCl (pH 8.0)] and twice with TE buffer

[1 mM EDTA and 10 mM Tris-HCl (pH 8.0)]. DNA–protein complexes were eluted with 250 μl of elution buffer (1% SDS, 0.1 M NaHCO$_3$ and 10 mM DTT). Crosslinks were reversed by heating at 65 °C in the presence of 0.2 M NaCl for at least 10 h, followed by RNase A treatment at 37 °C for 0.5 h and Proteinase K treatment at 55 °C for 2 h. DNA was purified using the QIAquick PCR purification kit (28106; Qiagen, Valencia, CA). Quantification of purified DNA was performed using NanoDrop One (Thermo Fisher Scientific). ChIP signals were detected by quantitative PCR (qPCR) using iCycler iQ Real-Time PCR Detection System (Bio-Rad, Hercules, CA), iQ™ SYBR Green Supermix (Bio-Rad) and the primer sets listed in Supplementary Information. Relative quantification of qPCR data was performed using the ΔCt method.

**Antibody-based in situ biotinylation assay.** HEK293T cells cultured in a 10 cm dish were fixed with 4% PFA for 10 min and permeabilized with 0.5% Triton X-100 for 20 min. Then, cells were blocked with blocking buffer containing 10% BSA, 10% horse serum and 0.2% Tween 20 in PBS. After blocking, cells were incubated with primary antibodies to Coilin (ab11822; Abcam) or LSM11 (HPA039587; Sigma-Aldrich Corp.) for 1 h at room temperature. The cells were subsequently washed with PBS containing 0.5% Triton X-100. The cells were then incubated with HRP-labeled goat antibody to mouse IgG or rabbit IgG. After washing, HRP-based in situ biotinylation was performed by incubating cells with biotinylation buffer (200 μM biotin-tyramide and 0.0015% $H_2O_2$ in PBS) for 1 min. The cells were immediately washed with PBS, and resuspended and lysed in 1% SDS RIPA buffer [150 mM NaCl, 1% Triton X-100, 0.5% Sodium deoxycholate (Doc), 1% SDS and 50 mM Tris-HCl (pH 8.0)]. The samples were sonicated with Bioruptor Sonicator (Diagenode, Denville, NJ) for 25 min. Then, the lysate was diluted twice with RIPA buffer without SDS (final concentration of SDS was 0.5%). SoftLink™ Soft Release Avidin Resin (Promega) was added to the lysate and incubated for 90 min at room temperature for the purification of biotinylated proteins. The avidin resin was washed with 0.5% SDS RIPA buffer [150 mM NaCl, 1% Triton X-100, 0.5% Doc, 0.5% SDS and 50 mM Tris-HCl (pH 8.0)], 0.5 M NaCl RIPA buffer [0.5 M NaCl, 1% Triton X-100, 0.5% Doc, 0.1% SDS and 50 mM Tris-HCl (pH 8.0)], 1.2 M NaCl RIPA buffer [1.2 M NaCl, 1% Triton X-100, 0.5% Doc, 0.1% SDS and 50 mM Tris-HCl (pH 8.0)], 0.5% SDS RIPA buffer again and TE buffer twice. Bound protein–chromatin complex was eluted and reverse-crosslinked in TE buffer containing 1% SDS and 0.2 M NaCl at 65 °C overnight. Precipitated DNA and input DNA were treated with RNase A for 30 min at 37 °C and proteinase K for 3 h at 55 °C. DNA was purified using the QIAquick PCR purification kit (28106; Qiagen). Immunoprecipitated and input materials were analyzed by qPCR or high-throughput DNA sequencing.

**ChIP-seq, in situ biotinylation-seq, and gene annotation.** Libraries for ChIP-seq and in situ biotinylation-seq were prepared using TruSeq ChIP Library Prep Kit (Illumina, San Diego, CA) and IDT for Illumina-TruSeq RNA UD Index (Illumina). For in situ biotinylation-seq, chromatin from *Drosophila melanogaster* S2 cells was added to each sample as a spike-in control for normalization. Sequencing reads were acquired using the NovaSeq 6000 or HiSeq 2500 platform. Adapter sequences were trimmed using Trim Galore (v 0.64_dev), and reads were mapped to human genome GRCH38 using the Bowtie2 alignment tool (version 2.3.5.1) with the default settings. The mapped reads from each ChIP-seq and in situ biotinylation-seq dataset were subjected to count per million (CPM) normalization and then used for downstream analysis. IP/input or IP-input enrichment per gene was calculated with R (v. 4.0.2) and its package rtracklayer (version 1.48.0) using the Ensembl GRCH38.92 gene annotation model.

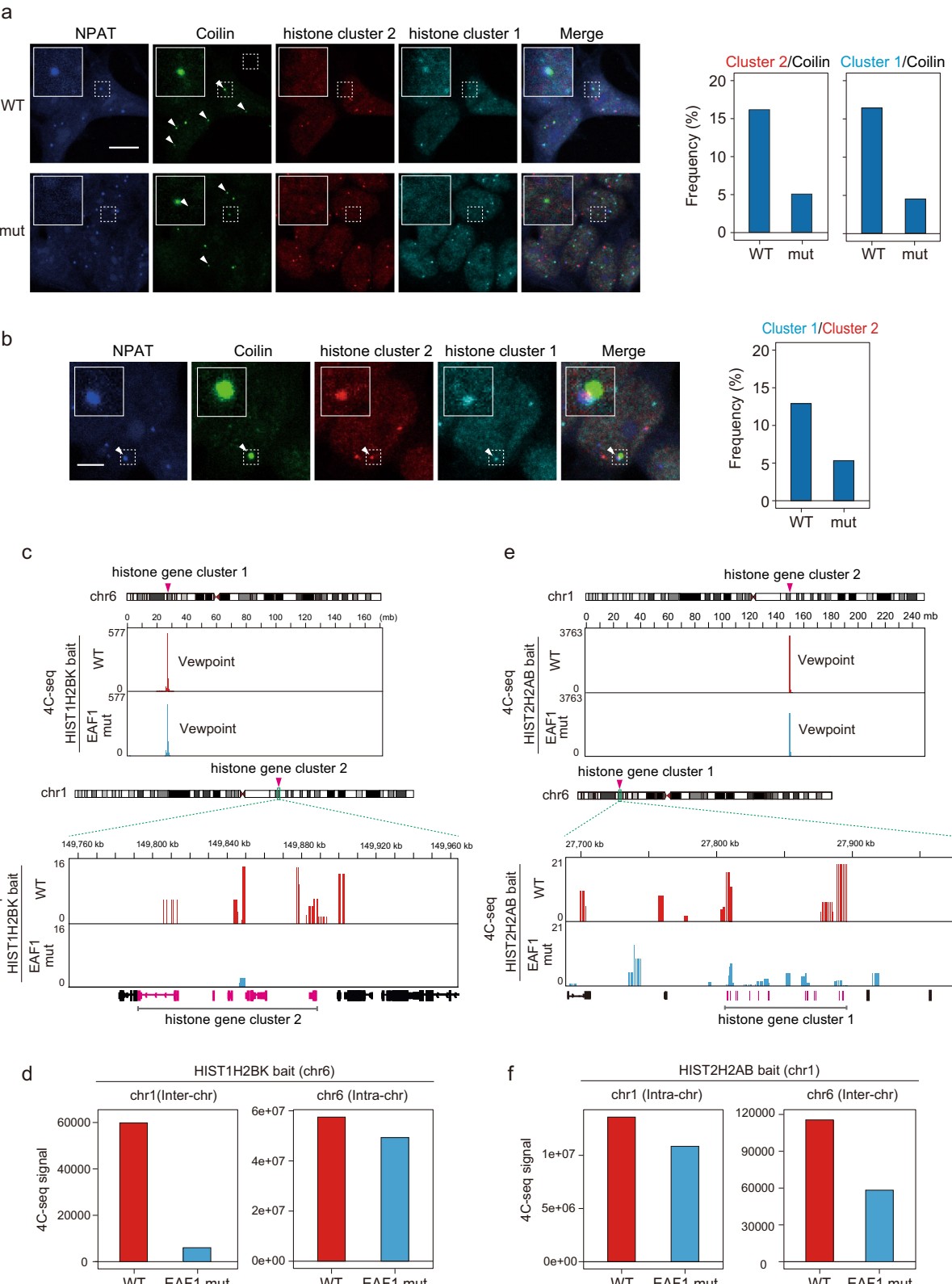

**PRO-seq analysis**. PRO-seq was performed as described previously[22]. Briefly, 25 million cells were collected and washed with ice-cold PBS. *Drosophila melanogaster* S2 cells (10% of the human cell number) were added to each sample as a spike-in control for normalization. The combined cells were resuspended in cold permeabilization buffer [10 mM Tris-HCl, pH 7.4, 300 mM sucrose, 10 mM KCl, 5 mM MgCl₂, 1 mM EGTA, 0.05% Tween-20, 0.1% NP40 substitute, 0.5 mM DTT, 1:100 protease inhibitor cocktail and 4 U/ml SUPERaseIN (Invitrogen)] and incubated on ice. The permeabilized cells were then pelleted, washed with permeabilization

buffer twice and resuspended in ice-cold storage buffer (10 mM Tris-HCl, pH 8.0, 25% glycerol, 5 mM MgCl₂, 0.1 mM EDTA, and 5 mM DTT) to $2 \times 10^7$ nuclei per 100 µl. Nuclear run-on (NRO) assays were performed with biotin-11-NTPs. In total, $2 \times 10^7$ nuclei per 100 µl were thoroughly mixed with an equal amount of pre-heated 2×NRO reaction mixture [10 mM Tris-HCl, pH 8.0, 5 mM MgCl₂, 300 mM KCl, 1 mM DTT, 1% Sarkosyl, 50 µM each of Biotin-11-A/G/C/UTP (PerkinElmer, Waltham, MA) and 0.8 U/µl RNase inhibitor] and incubated at 37 °C for 3 min in a heat block. Nascent RNA was extracted, purified and fragmented by base

**Fig. 9 Decreased higher-order inter-chromosome structure between two RDH gene clusters in EAF1-mutant cells. a** Wild-type HEK293T (WT) and EAF1-mutant cells (mut) were fixed with paraformaldehyde containing 0.5% Triton X-100 and subjected to DNA-FISH. Four-color microscopy images are shown. RDH gene clusters 1 (cyan) and 2 (red) were stained using Cy5- or Cy3-labeled probe. NPAT (blue) and Coilin (green) were stained using specific antibodies. Enlarged images for representative particles are shown in the upper left of each image. Scale bar, 10 μm. The frequencies of each RDH gene cluster 2 association with CB (Coilin) in wild-type (WT, $n = 359$ Coilin particles) and mutant cells (mut, $n = 556$ Coilin particles) were determined and are shown in the left panel. The frequencies of each RDH gene cluster 1 association with CB (Coilin) in wild-type (WT, $n = 359$ Coilin particles) and mutant cells (mut, $n = 556$ Coilin particles) were determined and are shown in the right panel. **b** Representative four-color microscopy images showing the overlap of CB, HLB, and two RDH gene clusters in wild-type HEK293T cells. RDH gene clusters, NPAT, and Coilin were stained as described above. Enlarged images for representative particles are shown in the upper left of each image. Scale bar, 5 μm. The frequency of inter-chromosome interaction between two RDH gene clusters is shown in the right panel (WT, $n = 1130$ RDH gene cluster 2; mut, $n = 771$ RDH gene cluster 2). **c–f** 4C-seq analysis revealed that the higher-order inter-chromosome structure between two RDH gene clusters was abolished in EAF1-mutant cells. **c, e** Representative 4C-seq profile using RDH gene cluster 1 (*HIST1H2BK*, chromosome 6) bait or RDH gene cluster 2 (*HIST2H2AB*, chromosome 1) bait. The detailed contact profile at RDH gene cluster 2 (chromosome 1) (**c**) or at RDH gene cluster 1 (chromosome 6) (**e**) is shown in the lower panels. **d, f** Comparisons of 4C-seq counts detected in each RDH gene cluster region in wild-type (WT) or EAF1-mutant (EAF1 mut) cells under each 4C-seq condition. The read counts of 4C-seq using chromosome 6 *HIST1H2BK* bait (**d**) and chromosome 1 *HIST2H2AB* bait (**f**) are shown. These plots represent the mean of $n = 2$ experiments. Source data are provided as a Source Data file.

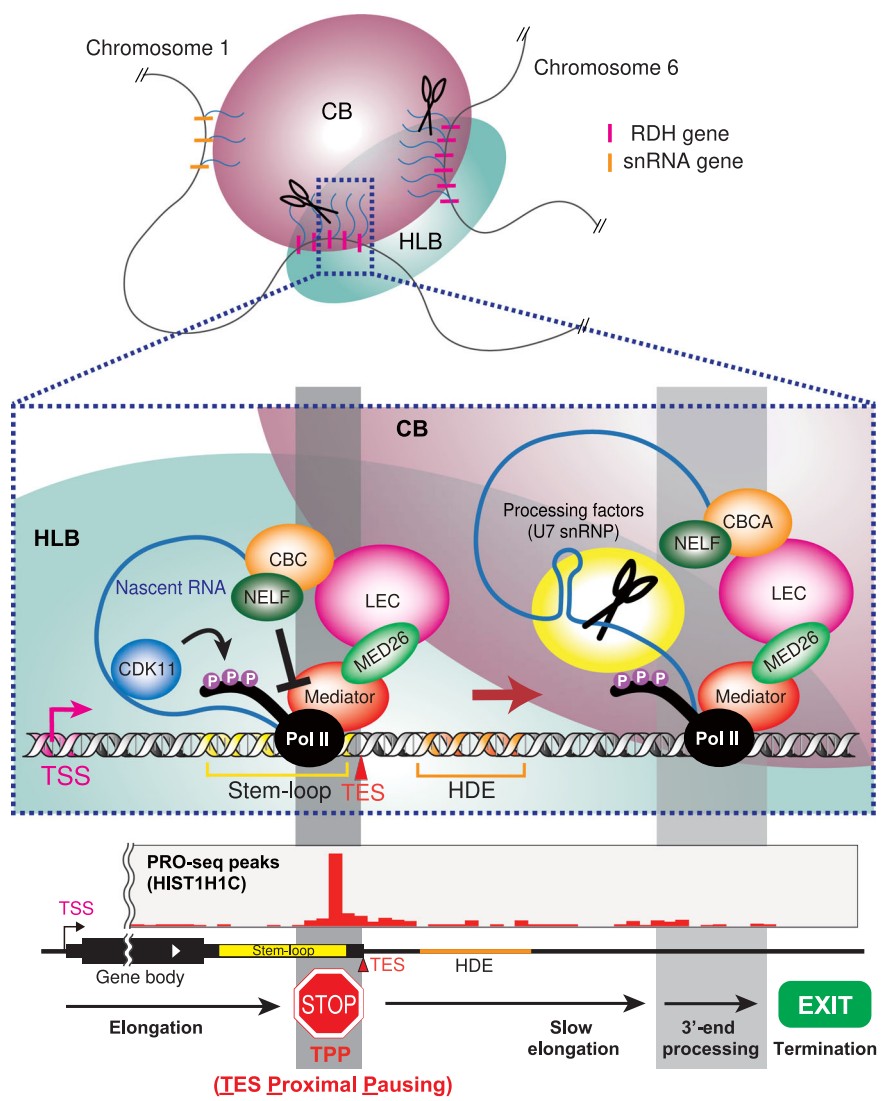

**Fig. 10 Model of the role of LEC recruitment by Mediator in CBs' association with HLBs to facilitate TPP and subsequent 3′-end processing of RDH genes.** Schematic illustration representing a model of the role of Mediator in CBs' association with HLBs to facilitate TPP and subsequent 3′-end processing of RDH genes. In this model, MED26-containing Mediator plays a role in the association of CBs with HLBs through interaction with LEC–CBC–NELF. CBs' association with HLBs leads to Pol II pausing immediately upstream of the TES (TPP) of RDH genes and subsequent 3′-end processing by supplying 3′-end processing factors from CBs. TPP plays a role in sufficient Ser2 phosphorylation of Pol II CTD by CDK11 to recruit 3′-end processing factors to the RDH genes. Thus, TPP acts as a checkpoint from transcription elongation to transcription termination for slow elongation past the HDE, appropriate 3′-end processing and production of non-polyadenylated RDH gene transcripts.

hydrolysis in 0.2 N NaOH on ice for 10 min. After neutralization, fragmented nascent RNA was bound to Dynabeads™ M-280 Streptavidin magnetic beads (Invitrogen) and incubated for 20 min at 4 °C. The beads were sequentially washed twice in high-salt buffer (2 M NaCl, 50 mM Tris-HCl, pH 7.4, and 0.5% Triton X-100), twice in medium-salt buffer (300 mM NaCl, 10 mM Tris-HCl, pH 7.4, and 0.1% Triton X-100) and once in low-salt buffer (5 mM Tris-HCl, pH 7.4, and 0.1% Triton X-100). Biotinylated RNA was extracted from the beads and precipitated in ethanol. 3′ RNA adaptors were ligated to biotinylated RNA and a second round of biotin–streptavidin purification was performed. The mRNA cap was then removed and the reverse 5′ RNA adaptor was ligated. After the third round of biotin–streptavidin purification, adaptor-ligated nascent RNA was reverse-transcribed (RT) into complementary DNA (cDNA) using RP1 primer. cDNA was amplified with index primers and amplicons of 120–350 bp were selected using AMPure XP beads (Beckman Coulter, Brea, CA). Equimolar concentrations of library fractions were then pooled together and sequenced using a high-output flow cell on the NovaSeq 6000 platform (Illumina).

Raw data of sequences were trimmed using Trim Galore (v 0.64_dev), and the reads were mapped to the human genome GRCh38 and *Drosophila melanogaster* genome build5.41 using the Bowtie2 alignment tool (version 2.3.5.1) with the default settings. Read counts were normalized according to the genomic coverage of mapped *Drosophila* reads using bedtools (version 2.29.2) and samtools (version 1.7). For α-amanitin-treated samples, the read counts were normalized with million rRNA reads to evaluate effect of α-amanitin on the incorporation of biotin-labeled NTPs into RNA by Pol II. The rRNA reads were defined as reads mapped to chrUn_GL000220v1: 105424-118780 and counted using bedtools. Pileup tracks of the last base pair of the reads were generated using bedtools (version 2.29.2) and used for downstream analysis. For read-through analysis, read counts from TSS to +200 bp downstream of it and from TES to +200 bp downstream of it were piled up by R package rtracklayer (version 1.48.0). refGene of GRCh38.p13 was used as a reference of transcripts. Then, the read-through ratio was defined using the following equation: (from TES to TES + 200 bp)/(from TSS to TSS + 200 bp). Meta-gene plots were generated with deeptools (version 3.3.1) using refGene as a reference. Multiple transcripts sharing the same TSS or TES were analyzed as a single gene transcript.

**RNA-seq analysis**. For ribo-depleted RNA-seq analysis, total RNA was isolated using miRNeasy Mini Kit (217004; Qiagen). One microgram of total RNA was subjected to the depletion of ribosomal RNA using the Ribo-Zero Kit (MRZH11124; Illumina), and libraries were prepared using the TruSeq Stranded Total RNA Library Prep Gold (Illumina). Raw reads from sequencing were demultiplexed allowing up to one mismatch using Illumina bcl2fastq2 v2.18. Adapter sequences were trimmed using Trim Galore (v 0.64_dev), and paired-end reads were mapped to human genome GRCH38 with hisat2 (version 2.3.4.1), using Ensembl gene annotation of GRCh38.p13. Read counts were obtained using Rsubread (version 2.2.6). Differential gene expression analysis was performed using R (v. 4.0.2) package DESeq2 (v. 1.28.1). Genes with an adjusted *P*-value <0.01 and absolute log2 fold-change >0.5 were included in the downstream analysis. A read-through plot was created with modified ngsplot (2.6.1) to show a logarithmic *y*-axis. *HIST2H2AC*, *HIST2H2AB* and *HIST1H3J* were excluded from this analysis because their TESs were too near to each other or the TES positions appeared to be incorrect.

**Immunofluorescence**. HeLa cells, 293T cells or HCT116 cells grown on coverslips were fixed with 4% PFA for 10 min or ice-cold methanol for 5 min and permeabilized with 0.5% Triton X-100 for 20 min. After blocking the cells with PBS containing 10% BSA, 10% horse serum and 0.2% Tween 20, they were incubated with primary antibodies to MED26 (D4B1X, 14950S; Cell Signaling Technology), MED1 (ab64965; Abcam), ELL (D7N6U, 14468 S; Cell Signaling Technology), ICE1 (HPA054452; Sigma-Aldrich Corp.), Coilin (ab11822; Abcam), NPAT (HPA066370, Sigma-Aldrich Corp.; or sc-136007, Santa Cruz), LSM11 (HPA039587; Sigma-Aldrich Corp.), FLASH (HPA053573; Sigma-Aldrich Corp.), WRAP53 (HPA029928; Sigma-Aldrich Corp.), SMN1 (2F1, 12976 S; Cell Signaling Technology) or CDK11 (HPA073626; Sigma-Aldrich Corp.). The cells were incubated with Alexa 488-labeled goat polyclonal antibody to rabbit IgG at 1:2000 dilution or Alexa 555-labeled goat polyclonal antibody to mouse IgG at 1:2000 dilution (Life Technologies). For triple-stain imaging, cells were stained using fluorescein-labeled NPAT antibody and HiLyte Fluor™ 647-labeled Coilin antibody after staining with primary antibodies and Alexa 555-labeled goat polyclonal antibody. Direct labeling of primary antibodies was performed using Ab-10 Rapid Fluorescein Labeling Kit (LK32; Dojindo) and Ab-10 Rapid HiLyte Fluor™ 647 Labeling Kit (LK36; Dojindo). Then, the cells were covered with a drop of Prolong Glass antifade reagent (Invitrogen) and photographed with a Zeiss LSM 700 Laser Scanning Microscope (Carl Zeiss, Oberkochen, Germany). Three-dimensional super-resolution images were acquired using a Leica TCS SP8 STED 3× Gated 660 system with a ×100 objective lens (HC PL APO CS2 ×100/1.40 NA OIL). The excitation was provided by a white light laser, the depletion was from a 660 nm STED laser with the three-dimensional slider adjusted to 60% and the fluorescence signal was acquired using a Leica HyD™ in time-gated mode. All images were deconvolved and arranged using Huygens software (Scientific Volume Imaging

B.V., The Netherlands) and Photoshop (Adobe, USA), respectively. Quantification of the signal intensity of the particles was performed using ImageJ Fiji software.

**The 4C-seq analysis**. The 4C-seq analysis was performed in accordance with the protocol of Krijger et al.[102]. HEK293T cells cultured in a 10 cm dish (~1 × 10⁷ cells) were fixed with 4% formaldehyde for 10 min at room temperature. Crosslinked HEK293T cells were washed with ice-cold PBS, resuspended with cell lysis buffer [150 mM NaCl, 50 mM Tris-HCl (pH 8.0), 1% Triton X-100, 0.5% NP-40 and 5 mM EDTA] containing protease inhibitors and incubated on ice for 20 min. The nuclei pelleted down by centrifugation were resuspended in 500 μl of 1.2× restriction enzyme 1 (RE1) buffer. The samples were warmed up to 37 °C and SDS was added at a final concentration of 0.3%, followed by incubation for 1 h at 37 °C while shaking at 750 rotations per min (rpm). Triton X-100 was added at a final concentration of 2.5% and incubated for 1 h at 37 °C while shaking at 750 rpm. One hundred units of RE1 was added to the samples and incubated overnight at 37 °C while shaking at 750 rpm. The resulting samples were then incubated at 65 °C for 20 min to inactivate RE1. A total of 700 μl of 10× ligation buffer, 50 units of T4 DNA ligase (10799009001; Roche Diagnostics, Tokyo, Japan) and Milli-Q water were added to the samples up to 7 ml, followed by incubation overnight at 16 °C. Samples were then reverse-crosslinked by adding 30 μl of Protease K (P2308; Sigma-Aldrich Corp.) and incubated overnight at 65 °C. The resulting 3C-templates were purified by P-beads (NucleoMag® NGS Clean-up and Size Select, 744970.5; Takara Bio Inc.) and eluted with 450 μl of 5 mM Tris-HCl (pH 8.0). Then, 50 μl of 10× restriction enzyme 2 (RE2) buffer and 50 units of RE2 were added to the samples, and the resulting samples were incubated overnight at 37 °C with shaking at 500 rpm. The samples were then incubated at 65 °C for 20 min to inactivate RE2. Next, 25 μg of DNA, 50 units of ligase and 450 μl of 10× ligation buffer were mixed and incubated overnight at 16 °C. The resulting 4C-templates were purified by P-beads and subjected to sequencing for library preparation. As representative targets, we used *HIST1H2BK* and *HIST2H2AB*, which are present on chromosomes 1 and 6, respectively. For 4C-seq analysis, we used *Nla*III (R0125S; NEB) and *Dpn*II (R0543S; NEB) as RE1 and RE2, respectively. For the data analysis, pipe4C was used with the default settings. The total 4C-seq signal of each histone locus (histone gene cluster 1: the genomic region from 27130000 to 27150000 and the region from 27801000 to 27903000 in chromosome 6; histone gene cluster 2: the genomic region from 149780000 to 149890000 in chromosome 1) was calculated with rtracklayer.

**Hybridization probes**. BAC DNA clones of RP11-116E21 and RP11-368M17 that cover histone cluster 1 (chr.6) and histone cluster 2 (chr.1), respectively, and scaRNA12 cDNA were used for the generation of hybridization probes. scaRNA12 cDNA was amplified by PCR from HeLa cDNA and cloned into the pBlueScript II vector (212205; Addgene). Probes were labeled with Cy3 or Cy5 using a Nick Translation Kit (32-801300; Abbott Laboratories, Chicago, IL, USA).

**Fluorescence in situ hybridization (FISH)**. HEK293T cells or HeLa cells grown on coverslips were fixed with 4% paraformaldehyde containing 0.5% Triton X-100 in PBS for 15 min. After rinsing with PBS, the cells were then permeabilized with 0.5% Triton X-100 in PBS for 20 min. Samples were subsequently immersed in 20% glycerol in PBS for more than 30 min and subjected to freeze-thawing four times using liquid nitrogen. Subsequently, the cells were incubated in 0.1 N HCl for 15 min, in 0.1 mg/ml RNase A in 2×SSC for 30 min and in 50% formamide in 2×SSC for 30 min. For denaturation and hybridization, the cells were incubated in a hybridization mixture (2×SSC, 50% formamide, 10% dextran sulfate, 1 mg/ml tRNA and 5 μg/ml probe DNA) at 85 °C for 5 min, and then incubated overnight at 37 °C. After hybridization, the cells were washed with 2×SSC and 50% formamide at 37 °C for 5 min and 2×SSC again for 5 min. RNA FISH was performed using the same protocol as DNA FISH except for the DNA denaturation step. For RNA FISH, 10 units/ml of RNasein Plus RNase inhibitor (N2611, Promega) was added to each incubation step.

**Measurement of the mean intensity of particles and the distance between two distinct particles**. At least five images containing ~1000 cells were taken with a Zeiss LSM 700 Laser Scanning Microscope (Carl Zeiss). Signals from outside of the nuclei were deleted to reduce background noise, and particles were recognized using ImageJ plugin particle analyzer with a fixed threshold at which particles of the control sample could be clearly recognized. Size, intensity, and center of mass for each particle, as well as the distance between two distinct particles, were calculated.

**Statistics and reproducibility**. At least three biological replicates were performed for each experiment. It was confirmed that each experiment produced similar results.

**Reporting summary**. Further information on research design is available in the Nature Research Reporting Summary linked to this article.

## Data availability

The data that support this study are available from the corresponding author upon reasonable request. ChIP-seq, RNA-seq, 4C-seq, antibody-based in situ biotinylation-seq, and PRO-seq data are deposited in GEO under accession number GSE164144. Source data are provided with this paper.

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

## Acknowledgements

We thank Miho Uchiumi for help in preparing the manuscript, Naomichi Matsumoto for providing us with BAC DNA clones and Tetsuro Kokubo for helpful discussions. We also thank Edanz (https://jp.edanz.com/ac) for editing a draft of this manuscript. This work was supported in part by KAKENHI [2020K15718 to H.S., 16H06279 (PAGS), 17K19578, 221S0002, 18H02378, 19K22401, 21H05159, 21H02405, and 21K19356 to H.T.] and by Takeda Science Foundation (H.T.), Suhara Memorial Fund (H.T.), Taka-matsu Cancer Research Fund (H.T.), Leukemia Research Fund (H.T.), The Ichiro Kanehara Foundation (H.T.), Friends of Leukemia Research Fund (H.T.), Ono Cancer Research Fund (H.T.), Kobayashi Foundation for Cancer Research (H.T.), MSD Life Science Foundation (H.T.), The Naito Foundation (H.T.), The Tokyo Biochemical Research Foundation (H.T.), Yokohama Foundation for Advancement of Medical Science (H.T.), Nakatani Foundation (H.T.) and by a grant from the Helen Nelson Medical Research Fund to the Stowers Institute (J.W.C.).

## Author contributions

H.S. planned the research and performed most of the experiments. H.S. performed PRO-seq. R.A. generated EAF1-mutant cells. R.A. analyzed the data from next-generation sequencing and immunostaining. M.S. performed reconstitution of LEC. S.S. and C.S. generated MED26-mutant cells. R.A., H.H., T.Y., and N.S. performed FISH experiments. A.T. and Y.S. performed next-generation sequencing. H.H., Y.I., N.Y., and K.F. performed immunostaining and T.H. performed STED microscopy. Y.I. performed western blotting. H.H. and K.N. helped with the antibody-based in situ biotinylation experiments. Y.Y. helped with the interpretation of results. H.S., R.A., J.W.C., and R.C.C. contributed to the writing of the manuscript. H.T. supervised the research and wrote the manuscript.

## Competing interests

The authors declare no competing interests.

**Additional information**

