## [Peer Review File · Nature Communications]

REVIEWER COMMENTS

Reviewer #1 (Remarks to the Author):

In this manuscript, Suzuki et al. study the mechanisms by which replication-dependent histone (RDH) genes are transcribed by RNA polymerase II (Pol II) and how the pausing of Pol II at the 3' end of the RDH genes is regulated. The authors reproduce previous observations showing that Pol II pauses immediately upstream of transcription end sites of RDH genes. In addition, they describe that the Mediator complex plays a role in this Pol II pausing at the 3' end of the RDH genes through association with Cajal bodies (CBs) at histone locus bodies (HLBs). Thus, the authors propose "that the Mediator is involved in CBs' association with HLBs to facilitate transcription termination of RDH genes by supplying 3-end processing factors."

The manuscript starts by rediscovering that Pol II pausing immediately upstream of transcription end sites of RDH genes is different from genes of which the transcripts are polyadenylated, and continues with a number of interesting novel mechanistic explanations about how this RDH gene-specific 3' pausing could be regulated by LEC. In conclusion, this reviewer would suggest to shorten the ms, by deleting all the non-essential "confirmatory" data, and rather by concentrating on the role of LEC (EAF1) in regulating Pol II pausing at the 3' end sites of RDH genes, as well as at the interphase of CBs and HLBs.

Concerns:

- The authors have to cite the paper(s) previously describing the unexpected Pol II pausing pattern immediately upstream of RDH genes (PMID: 22701709; PMID: 23131668). In the light of these publications, the title and several parts of the manuscript have to be rewritten. Consequently the "novel 3' Pol II pausing at replication-dependent histone genes" has lost its "novelty", "unexpectedness", or its "striking-ness". Authors should consider showing Figure 1 as Supplemental Figure 1.

-What do the authors call exactly "snoRNA genes"? SnoRNAs are either independently transcribed, or intronic snoRNAs processed by exonucleases from polyA+ mRNA transcripts.

-Pro-seq signals are different between RDH (and snoRNA) and snRNA genes, as on snRNA genes there are always multiple Pro-seq peaks compared to unique Pro-seq peak on RDH (and snoRNA) genes. What are the multiple peaks corresponding to at the snRNA genes?

-In their previous Nature Com paper (PMID: 32102997) authors from the same labs demonstrated that that the MED complex recruited the little elongation complex (LEC) and negative elongation factor (NELF) to regulate transcription termination of RDH transcripts at Cajal bodies (PMID: 32102997). Moreover, other studies showed that NELF and the cap-binding complex (CBC) interact to regulate 3' end processing of replication-dependent histone mRNAs. In this study now, the authors show that the knockdown of NELF or CBC subunits decreased Pol II pausing (Pro-seq signal) at the 3' end of RDH genes (Figure 2). These results are rather confirmatory.

-How were the different Pro-seq and ChIP-seq data sets normalized for comparisons?

-For general, readers the authors need to better introduce Coilin and NPAT. They are markers of what exactly?

-The authors point out that the Mediator complex, CBs and HLBs have been suggested to be involved in the formation of LLPS-mediated droplets. Interestingly, their beautiful IF staining experiments (Figure 6 a and b, HEK293 cells) do not reveal any of these phase-

separated droplets with CB and HLB markers, or with Med complex (in Figure 7, in HeLa cells). Could the authors comment and discuss these important observations?

-To be able to demonstrate more precisely the localization of CBs, HLBs and especially the Mediator complex, the authors should carry out super resolution microscopy. Is the CB (coilin)-HLB (NPAT) distance increasing when the EAF1 subunit of LEC is mutated?

-Would it be possible to localize LEC subunits by IF and compare their localization to CBs and HLBs?

-In Figure 6 the authors use HEK293 cells, but in Figure 7 they use HeLa cells, do the experiments carried out in the two different cell lines give similar or comparable conclusions?

-In Figure 9a (and b) the four-color microscopy images are difficult to see. The authors should consider showing some magnified insets.

-Are the histone gene clusters (1 and 2) transcribed when they are not associated with CBs and HLBs?

-Figure 9c and e: what are the upper panels showing exactly? The EAF1 mutant effects are rather modest, while on other figure panels the effects are quite strong, why?

-In Figure 10, 'poximal' should be 'proximal'.

Reviewer #2 (Remarks to the Author):

In this study Suzuki et al set out to examine the mechanisms of transcription termination of non-polyadenylated mRNAs encoded by replication-dependent histone genes (RDHs). As a start they found that Pol II is paused at the TES proximal region (TER) of many genes including RDH and snoRNA genes. This pausing at non-polyadenylated genes was termed TES proximal pausing (TPP). In the RDH genes Pol II paused before a stem-loop structure was formed. Next, they KD proteins involved in 3'-end processing of RDH and found that KD of NELF-E or CBP80 significantly decreased TPP at RDH and snoRNA genes, but not at snoRNA genes. Since they had previously found that Mediator subunit MED26 is also involved in this termination, they generated a cell line containing a MED26 mutant that cannot interact with the Little Elongation Complex (LEC). Also, they mutated AF4 and AFF4 which are subunits of the Super Elongation Complex (SEC). WT AF4 and AFF4, but not the mutated forms, interacted with MED26. The interactions of WT and mutant EAF1 were also tested for many components, showing that a mutation in EAF1 specifically abrogates Mediator's interaction with LEC-CBC-NELF, but not with the formation of LEC. Expression of snoRNA and RDH genes was downregulated in EAF1-mutant cells as was 3'-end processing of RDH transcripts. Importantly, TPP of RDH genes disappeared in EAF1-mutant cells and read-through of Pol II increased substantially (+200bp), showing that Mediator's interaction with LEC-CBC-NELF is required for TPP and 3'-end processing. Next, the authors examined whether mediator plays a role in the interaction between CBs and HLBs and found a decrease in the overlap between the two bodies in EAF1 and MED26 mutant cells. The data suggested that the interaction between Mediator and LEC is required for CB-HLB association. Imaging of the bodies showed that Mediator localized between CBs and HLBs. Next, using the "in situ biotin-labelling of protein with high throughput DNA sequencing" technique they developed, they identified the genomic regions with which CBs associate, which included the RDH genes. Interactions were reduced in EAF1 and ICE1 mutant cells

and loss of TPP at RDH genes was observed. Finally, FISH analysis showed that CB association (and not HLB) with RDH genes requires Mediator's interaction with LEC, and 4C-seq showed that this interaction is required for inter-chromosome association. Altogether, in this study the authors delineate a compelling model in which LEC recruitment by Mediator plays a role in CB-HLB association and facilitates 3'-end processing of RDHs. They also define the areas in the genome associate with CBs. This is a very elaborate study with much well-presented data obtained using a multitude of high-end approaches. I find the study clear and convincing and worthwhile for publication in Nature Communications pending a few comments below.

Fig. 3b,c: What are Mw of the proteins? Why do the mutants run as several bands and why are they heavier in MW, shouldn't they run faster in the gel? The bands are very strong for the WT as well, and it seems that in b there are 2 bands for the WT – why, and which of the two is IP'd (in c we see two bands as well so same question)?

Fig. 6a, b – would be nice to see an enlarged are of the images in general to get to see the spatial relationship between CBs and HLBs, and more so to see a visual example of the reduction in area (not only in the plots).

Fig. 7a – "Mediator components MED26 and MED1, and the LEC components ICE1 and ELL colocalized with CBs in HeLa cells" – actually for ICE1, the bottom nucleus does not show much colocalization. Same comment for the NPAT images.

Fig. 9a suggest to enlarge, really hard to see what the authors are pointing at.

Page 12 line 20,21 – "meta-gene analysis of RDH gene transcripts revealed that increased RDH transcripts were detected downstream" – increased what? Levels?

Reviewer #3 (Remarks to the Author):

The authors propose a function for pol II pausing regulated by mediator at histone genes that is relevant to transcription termination. They have previously presented data on other non-polyadenylated RNAs.

A major issue with this paper is that they do not discuss the major advances in transcription termination on these genes which has occurred over the past two years. Several years ago Price and coworkers (Guo et al., 2014, Adamson and Price, 2003) and Tora (Anamika et al., 2012) demonstrated that pol II stalls shortly after the histone processing signals and terminates. Both Bentley and West's group for histone mRNAs have recently shown transcription termination on histone genes occurs shortly after transcribing past the processing site by activation of a protein phosphatase, leading to polymerase stalling and termination (Eaton and West, 2020, Eaton et al., 2020, Eaton et al., 2018, Cortazar et al., 2019), well beyond the proposed pause site in this paper and Wagner (Huang et al., 2020) has shown similar results with a different phosphatase on snRNA genes.

Major problem:

1. The PRO-SEQ data, which the authors use to measure where the pause sites are, is clearly seriously contaminated with cellular RNAs. Each histone gene shows a major peak right at the end of the transcript, and the sequence in Fig. 3I shows it is at the place where histone mRNA is processed. Thus that peak is processed histone mRNA and not paused transcripts. Similar results are found for all the histones (Fig. 5A-D), and snRNA genes, and even for 28S rRNA. These cannot be polymerase pause sites relevant to processing and termination, since the processing signals haven't been transcribed yet and the 3' stemloop in histone genes would still be inside pol II. Given this technical issue any discussion of termination in this paper is unwarranted. Note that this is not an issue for the

polyadenylated mRNAs since any reads would not have been mapped.

2. The proposal that Cajal bodies and HLBs exchange components during histone gene expression seems very unlikely. While in some cells coilin is present in HLBs, in many cells the two bodies are separate. U7 snRNP is primarily found in HLBs, and not in all Cajal bodies, although it is certainly possible that U7 snRNP passes through the Cajal body as part of its maturation process. In the initial Frey and Matera paper (ref. 27) the point of the paper was that U7 snRNA was found only in the Cajal bodies near histone genes (these are the HLBs, and this paper was published before the HLB had been defined). I am not aware of any data that NPAT or FLASH are found in Cajal bodies, except for HLBs that contain some coilin. It has recently been definitively shown that histone gene transcription and processing occurs inside the HLB (Kemp et al., 2021).

3. They define HLF, heat labile factor, incorrectly. Kolev and Steitz showed in 2006 that HLF is symplekin. The complex they define as HLF is actually the HCC (histone cleavage complex) which is part of the active U7 snRNP, which contains symplekin, CPSF73, CPSF100 and CstF64. This complex is bound to the core U7 snRNP through FLASH which specifically binds Lsm11 and the HCC. This complex was described in ref. 8 in their paper.

Reference List

- Adamson, T. E. & Price, D. H. (2003). Cotranscriptional processing of *Drosophila* histone mRNAs. *Mol Cell Biol*, 23, 4046-55.
- Anamika, K., Gyenis, A., Poidevin, L., Poch, O. & Tora, L. (2012). RNA polymerase II pausing downstream of core histone genes is different from genes producing polyadenylated transcripts. *PLoS ONE*, 7, e38769.
- Cortazar, M. A., Sheridan, R. M., Erickson, B., Fong, N., Glover-Cutter, K., Brannan, K. & Bentley, D. L. (2019). Control of RNA Pol II Speed by PNUITS-PP1 and Spt5 Dephosphorylation Facilitates Termination by a "Sitting Duck Torpedo" Mechanism. *Mol Cell*, 76, 896-908 e4.
- Eaton, J. D., Davidson, L., Bauer, D. L. V., Natsume, T., Kanemaki, M. T. & West, S. (2018). Xrn2 accelerates termination by RNA polymerase II, which is underpinned by CPSF73 activity. *Genes Dev*, 32, 127-139.
- Eaton, J. D., Francis, L., Davidson, L. & West, S. (2020). A unified allosteric/torpedo mechanism for transcriptional termination on human protein-coding genes. *Genes Dev*, 34, 132-145.
- Eaton, J. D. & West, S. (2020). Termination of Transcription by RNA Polymerase II: BOOM! *Trends Genet*, 36, 664-675.
- Guo, J., Turek, M. E. & Price, D. H. (2014). Regulation of RNA polymerase II termination by phosphorylation of Gdown1. *J. Biol. Chem*, 289, 12657-12665.
- Huang, K. L., Jee, D., Stein, C. B., Elrod, N. D., Henriques, T., Mascibroda, L. G., Baillat, D., Russell, W. K., Adelman, K. & Wagner, E. J. (2020). Integrator Recruits Protein Phosphatase 2A to Prevent Pause Release and Facilitate Transcription Termination. *Mol Cell*.
- Kemp, J. P., Jr., Yang, X. C., Dominski, Z., Marzluff, W. F. & Duronio, R. J. (2021). Superresolution light microscopy of the *Drosophila* histone locus body reveals a core-shell organization associated with expression of replication-dependent histone genes. *Mol Biol Cell*, 32, 942-955.

POINT-BY-POINT RESPONSES TO REVIEWERS' COMMENTS

The reviewers' comments are listed below in boldface black characters, and our point-by-point responses to the comments are shown in regular green characters. Sentences quoted from the revised manuscript are shown in regular brown characters.

Reviewer #1 (Remarks to the Author):

In this manuscript, Suzuki at al. study the mechanisms by which replication-dependent histone (RDH) genes are transcribed by RNA polymerase II (Pol II) and how the pausing of Pol II at the 3' end of the RDH genes is regulated. The authors reproduce previous observations showing that Pol II pauses immediately upstream of transcription end sites of RDH genes. In addition, they describe that the Mediator complex plays a role in this Pol II pausing at the 3' end of the RDH genes through association with Cajal bodies (CBs) at histone locus bodies (HLBs). Thus, the authors propose "that the Mediator is involved in CBs' association with HLBs to facilitate transcription termination of RDH genes by supplying 3-end processing factors."

The manuscript starts by rediscovering that Pol II pausing immediately upstream of transcription end sites of RDH genes is different from genes of which the transcripts are polyadenylated, and continues with a number of interesting novel mechanistic explanations about how this RDH gene-specific 3' pausing could be regulated by LEC. In conclusion, this reviewer would suggest to shorten the ms, by deleting all the non-essential "confirmatory" data, and rather by concentrating on the role of LEC (EAF1) in regulating Pol II pausing at the 3' end sites of RDH genes, as well as at the interphase of CBs and HLBs.

Concerns:

1) The authors have to cite the paper(s) previously describing the unexpected Pol II pausing pattern immediately upstream of RDH genes (PMID: 22701709; PMID: 23131668). In the light of these publications, the title and several parts of the manuscript have to be rewritten. Consequently the "novel 3' Pol II pausing at replication-dependent histone genes" has lost its "novelty", "unexpectedness", or its "striking-ness". Authors should consider showing Figure 1 as Supplemental Figure 1.

We thank the reviewer for this comment. In accordance with the reviewer's suggestion, we have cited the papers. We stated in the Introduction section: "Several lines of evidence have indicated that Pol II is arrested or pauses downstream of transcription end sites (TESs) of RDH genes. *In*

in vitro transcription assay with RDH genes of *Drosophila melanogaster* revealed that Pol II is arrested 32 to 35 nucleotides downstream of TESS^{11, 12}. Genome-wide analysis including ChIP-seq of Pol II and Global Run On sequencing (GRO-seq) using human cells showed that Pol II pauses downstream of TESSs of RDH genes^{13, 14}. ” (Page 4, lines 2–6)

In accordance with the reviewer’s comment that “novel 3’ Pol II pausing at RDH genes” has lost its “novelty”, “unexpectedness”, or its “striking-ness”, we removed “A novel” from the title. Now, the revised title is “The 3’ Pol II pausing at replication-dependent histone genes is regulated by Mediator through Cajal bodies’ association with histone locus bodies”.

Notably, these previous reports showed that Pol II pauses downstream of the transcription end site (TES) of RDH genes; in contrast, our PRO-seq analysis revealed that Pol II pauses immediately upstream of the TES of RDH genes. Because PRO-seq enables precise detection of the 3’ end of nascent transcripts, which are produced by elongating or paused polymerases, but not stalled or arrested ones, it is thought that our results do not contradict the previous reports based on ChIP-seq or GRO-seq, and that we precisely detected the 3’ end of the nascent transcripts of RDH genes. Now, we have mentioned this issue in the Result section and stated: “In previous studies, ChIP-seq of Pol II and GRO-seq revealed that Pol II pauses downstream of RDH genes^{13, 14}. Because PRO-seq enables precise detection of the 3’ end of nascent transcripts, which are produced by elongating or paused polymerases, but not stalled or arrested polymerases, it is thought that our results do not contradict the previous reports based on ChIP-seq and GRO-seq. Taking the findings together, our analysis revealed that Pol II pauses immediately upstream of the TES of RDH genes.” (Page 10, lines 9–14)

While the reviewer pointed out that 3’ Pol II pausing at RDH genes has lost its “novelty”, our PRO-seq analysis revealed two novel points that (i) Pol II pauses immediately upstream of the TES of RDH genes (revised Fig. 1c–e, h, revised Supplementary Fig. 2a, b, k) and that (ii) Pol II pauses within the stem-loop region of most RDH genes (new Supplementary Fig. 3). These results indicate that Pol II pauses before the stem-loop structure is formed at RDH genes. Considering these results of ours, we made a revised version of Fig. 1 instead of moving original Fig. 1 to Supplementary Fig. 1. We now show these two points in revised Fig. 1c–e, h, revised Supplementary Fig. 2a, b, k, and new Supplementary Fig. 3.

2. What do the authors call exactly "snoRNA genes"? SnoRNAs are either independently transcribed, or intronic snoRNAs processed by exonucleases from polyA+ mRNA transcripts.

We thank the reviewer for raising this issue. As the reviewer pointed out, there are two types of snRNA. One is poly- or mono-cistronic snoRNAs, which are transcribed independently, and the

other is intronic snoRNAs, which are present in introns of snoRNA host genes and processed by exonucleases from the introns of pre-mRNAs of the genes¹. Because both types of snoRNA were included in our analysis, we have not distinguished them. As shown in new Supplementary Fig. 2g, we found that Pol II pauses immediately upstream of TESs of both types of snoRNA gene.

3. Pro-seq signals are different between RDH (and snoRNA) and snRNA genes, as on snRNA genes there are always multiple Pro-seq peaks compared to unique Pro-seq peak on RDH (and snoRNA) genes. What are the multiple peaks corresponding to at the snRNA genes?

There are two classes of snRNAs: Sm and Lsm classes. The Sm class of snRNAs comprises U1, U2, U4, U4atac, U5, U7, U11, U12 and a subset of U6 snRNAs, and is transcribed by Pol II, whereas the Lsm class of snRNAs comprises U6 snRNAs and U6atac snRNA, which are transcribed by Pol III¹. We detected TPP at a number of Sm-class snRNAs. Sm-class snRNAs contain a 3'-stem loop immediately upstream of the TES¹. We applied the computational analysis called “MXfold2” to detect the 3'-stem loop around the TESs of snRNA genes². As shown in the figure below, there is a stem loop around the TESs of snRNA genes. Comparing with the stem loop of RDH genes, which are usually located at almost the same position as the

TES, the 3'-stem loops of snRNA genes are located at a variety of positions relative to the TES. Although we do not have any evidence showing that the stem loop sequence of RDH genes is involved in TPP at RDH genes, if we assume so, it is possible that variation of the position of the 3'-stem loop sequence of snRNA genes affects the multiple positions of TPP at snRNA genes.

4. In their previous Nature Com paper (PMID: 32102997) authors from the same labs demonstrated that that the MED complex recruited the little elongation complex (LEC) and negative elongation factor (NELF) to regulate transcription termination of RDH transcripts at Cajal bodies (PMID: 32102997). Moreover, other studies showed that NELF and the cap-binding complex (CBC) interact to regulate 3' end processing of replication-dependent histone mRNAs. In this study now, the authors show that the knockdown of NELF or CBC subunits decreased Pol II pausing (Pro-seq signal) at the 3' end of RDH genes (Figure 2). These results are rather confirmatory.

We thank the reviewer for raising this issue. In our previous report (PMID: 32102997)³, we showed that LEC recruited by MED26-containing Mediator plays a role in transcription termination of non-polyadenylated genes including RDH and snRNA genes. In the previous paper, proteomic and biochemical analyses revealed that LEC interacts with CBC–NELF through direct interaction between CBP80, a component of CBC, and ICE1, a component of LEC. In addition, other reports demonstrated that CBC–NELF plays a role in repressing Pol II read-through at snRNA and RDH genes^{4,5}. Taking these findings together, we previously proposed a model that LEC recruited by MED26-containing Mediator plays a role in transcription termination of RDH and snRNA genes through interacting with CBC–NELF.

In this study, we found that Pol II pauses immediately upstream of the TES of RDH genes (revised Fig. 1a–h and revised Supplementary Fig. 2a–g, k). Notably, Pol II pauses within the stem-loop region of most RDH genes (new Supplementary Fig. 3). We called such pausing of Pol II “TES proximal pausing” (TPP). Knockdown of NELF-E, one of the components of NELF, or CBP80 significantly abolished Pol II TPP at RDH genes (Fig. 2a–c, and Supplementary Fig. 4a, b). Because we previously found that MED26 plays a role in transcription termination of RDH genes through direct interaction with LEC–CBC–NELF³, we thought it possible that Mediator’s interaction with LEC contributes to TPP and subsequent 3’-end processing of RDH gene transcripts. To test this possibility, we mutated the MED26 binding site of EAF1, which plays a role as a docking site of LEC on Mediator. In this EAF1-point-mutant cell line, Mediator’s interaction with LEC–CBC–NELF was specifically abolished, but that of SEC was not (Fig. 3a–e). Intriguingly, EAF1 mutation drastically interfered with Pol II TPP at RDH genes (Fig. 5a–e), resulting in increased aberrant unprocessed RDH gene transcripts (Fig. 4a–e, new Fig. 4f).

On the basis of our findings, we propose a model for the role of Mediator in TPP of Pol II and subsequent 3’-end processing of RDH genes. In this model, MED26-containing Mediator plays a role in TPP through interacting with LEC–CBC–NELF. Thus, our results raise

the possibility that TPP is a key checkpoint of the transition from transcription elongation to transcription termination for appropriate 3'-end processing of RDH genes and the production of non-polyadenylated transcripts (revised Fig. 10).

5. How were the different Pro-seq and ChIP-seq data sets normalized for comparisons?

We thank the reviewer for raising this issue. As the reviewer pointed out, appropriate normalization between different data sets is very important. In this study, we performed comparison of PRO-seq or *in situ* biotinylation-seq (ChIP-seq). In both cases, we took advantage of the spike-in method using *Drosophila* S2 cells. For PRO-seq, equal numbers of *Drosophila* S2 cells were mixed as a spike-in control when nuclear run-on reaction was performed. For ChIP-seq, we added an equal amount of ChIPed *Drosophila* chromatin to the samples. The mapped read count of each experimental data set was normalized using the mapped *Drosophila* read count.

6. For general, readers the authors need to better introduce Coilin and NPAT. They are markers of what exactly?

We thank the reviewer for this suggestion. In accordance with this suggestion, we introduced Coilin and NPAT in the Introduction section.

For the introduction of Coilin, we stated: “Coilin is a marker protein for CBs and has been used in immunofluorescent analysis of them³⁷. Coilin interacts with a number of the components of CBs including snRNPs, Survival of Motor Neuron protein (SMN) and WD40 Repeat-Containing Protein encoding RNA antisense to p53 (WRAP53), and noncoding RNAs including snRNAs and snoRNAs³⁸. Coilin plays an important role in CB formation and small nuclear ribonucleoprotein (snRNP) assembly³⁸.” (Page 5, lines 14–19)

For the introduction of NPAT, we stated: “Nuclear Protein, Coactivator of Histone Transcription (NPAT) is an HLB marker protein that plays a critical role in RDH gene transcription^{29, 30, 31}. It has been shown that NPAT interacts with a DNA binding transcription activator, POU class 2 homeobox 1 (POU2F1), and functions as a critical coactivator in RDH gene transcription through recruiting a variety of coactivators including the transformation/transactivation domain-associated protein (TRRAP)–Tip60 complex^{32, 33}. Notably, NPAT is phosphorylated by Cyclin E–CDK2 at the beginning of S phase, triggering the transcription of RDH genes^{31, 34}. ” (Page 5, lines 5–12)

7. The authors point out that the Mediator complex, CBs and HLBs have been suggested

to be involved in the formation of LLPS-mediated droplets. Interestingly, their beautiful IF staining experiments (Figure 6 a and b, HEK293 cells) do not reveal any of these phase-separated droplets with CB and HLB markers, or with Med complex (in Figure 7, in HeLa cells). Could the authors comments and discuss these important observations?

We thank the reviewer for this suggestion. In accordance with this suggestion, we discussed the possibility that MED26-containing Mediator, CBs and HLBs are involved in the formation of LLPS-mediated droplets in the Discussion section. We stated: “Many recent studies have shown that membrane-less organelles such as nuclear bodies and nuclear speckles are formed by mechanisms of liquid–liquid phase separation (LLPS)⁶⁵. Recently, increasing evidence has shown that LLPS is involved in many of the nuclear functions, including gene transcription, RNA processing and DNA repair^{66, 67, 68}. It is thought that membrane-less organelles are formed by weak multivalent interactions among proteins containing a low-complexity region or intrinsically disordered region (IDR)⁶⁵. Considering that IDRs constitute large parts of both Coilin and NPAT, it is highly possible that LLPS is involved in the formation of CBs and HLBs. Recently, it has been demonstrated that NPAT phosphorylation by Cyclin E/CDK2 regulates LLPS-dependent HLB formation and the 3'-end processing of RDH gene transcripts³⁴. Moreover, it has been reported that, first, CBs undergo fusion and fission, which are key features of LLPS-mediated droplets⁶⁹, and that, second, CBs are sensitive to aliphatic alcohol 1,6-hexanediol^{70, 71, 72}, suggesting that LLPS is involved in CB formation. Furthermore, the human Mediator subunit MED1 has been shown to play a role in super-enhancer formation through LLPS⁷³. Considering our results showing that MED26-containing Mediator is involved in the association of CBs with HLBs, it is possible that Mediator contributes to the fusion of CBs with HLBs through LLPS.” (Page 27, lines 19–24, and Page 28, lines 1–10)

8. To be able to demonstrate more precisely the localization of CBs, HLBs and especially the Mediator complex, the authors should carry out super resolution microscopy. Is the CB (coilin)-HLB (NPAT) distance increasing when the EAF1 subunit of LEC is mutated? Would it be possible to localize LEC subunits by IF and compare their localization to CBs and HLBs?

We thank the reviewer for pointing out this issue. To demonstrate the relative positions of the components of Mediator, LEC and 3'-end processing factors at CBs and HLBs, we performed triple immunofluorescence analysis using HeLa cells and detected NPAT particles, which associate with both particles of Coilin, and the other factor, which includes the components of Mediator, LEC or 3'-end processing factors. We generated averaged particle images from more

than 50 particles and calculated the distance between the centre of each Coilin or NPAT particle and the centre of each particle of Mediator components, MED26 or MED1, LEC components, ELL or ICE1, and 3'-end processing factors, LSM11 or FLASH (new Fig. 7a–f). Intriguingly, the components of Mediator, MED1 and MED26, and the component of 3'-end processing factors, LSM11, localized between CBs and HLBs (new Fig. 7a, b, e). In contrast, the components of LEC, ELL and ICE1, mainly colocalized at CBs and the component of 3'-end processing factors, FLASH, mainly colocalized at HLBs (new Fig. 7c, d, f). We calculated the ratios of ICE1, ELL, LSM11, MED1, MED26 or FLASH colocalized with CBs, HLBs, or both CBs and HLBs. This analysis revealed that, first, LEC mainly colocalized with CBs, second, FLASH mainly colocalized with HLBs, and third, Mediator and LSM11 localized between CBs and HLBs (new Fig. 7i, j).

To further address the possibility that Mediator localizes between CBs and HLBs, we employed stimulated emission depletion (STED) super-resolution microscopy and observed MED26, Coilin and NPAT in HCT116 cells. As shown in Fig. 7g and h, particles of NPAT were clearly distinguished from those of Coilin and particles of MED26 much more closely colocalized with Coilin. Taken together, these results also suggest that Mediator localized between CBs and HLBs (new Fig. 7j).

To answer the reviewer's question of whether CB (coilin)–HLB (NPAT) distance is increased in EAF1-mutant HEK293T cells, we performed immunofluorescence of MED26, Coilin and NPAT using wild-type or EAF1-mutant HEK293T cells and obtained super-resolution images. As shown in new Fig. 6e and f, as well as the frequency of CB (Coilin)–HLB (NPAT) colocalization being decreased in EAF1-mutant HEK293T cells, the distance between Coilin (CBs) and NPAT (HLBs) was increased in EAF1-mutant HEK293T cells.

9. In Figure 6 the authors use HEK293 cells, but in Figure 7 they use HeLa cells, do the experiments carried out in the two different cell lines give similar or comparable conclusions?

We appreciate the reviewer for raising this point. To answer the reviewer's question, we performed immunofluorescence analysis using HeLa cells, HEK293T cells and HCT116 cells and investigated the relative positions of the components of Mediator, LEC and 3'-end processing factors at CBs and HLBs. Comparable localization patterns observed in HeLa cells (new Fig. 7a–f) were also observed in HCT116 cells (new Supplementary Fig. 8a) and HEK293T cells (new Supplementary Fig. 8b).

10. In Figure 9a (and b) the four-color microscopy images are difficult to see. The authors

should consider showing some magnified insets.

We appreciate the reviewer's suggestion. Accordingly, we show magnified insets in revised Fig. 9a and b.

11. Are the histone gene clusters (1 and 2) transcribed when they are not associated with CBs and HLBs?

We appreciate the reviewer for raising this important point. It has been shown that the assembly of HLBs through NPAT concentration at RDH genes is required for RDH gene transcription^{6, 7, 8, 9}. In addition, recent genome-wide 4C-seq analysis revealed that CBs associate with HLBs containing RDH gene clusters and play a role in both the transcription of RDH genes and the genomic conformation of RDH gene clusters¹⁰.

We also found that the RDH genes with which CBs associate are transcribed at much higher levels than other RDH genes (new Fig. 8g). In EAF1-mutant cells, the expression levels of CB-associated RDH genes were significantly decreased accompanying CBs' dissociation (new Fig. 8g). These results are also consistent with recent reports showing that Pol II condensates with active transcription emerge and colocalize with both CBs and HLBs at the beginning of S phase¹¹, suggesting that CBs' association with HLBs plays an important role in transcription activation of RDH genes at S phase. Taken together, our results and other previous reports indicate that the association of histone gene clusters with both HLBs and CBs plays an important role for the transcription activation of RDH genes.

12. Figure 9c and e: what are the upper panels showing exactly? The EAF1 mutant effects are rather modest, while on other figure panels the effects are quite strong, why?

We appreciate the reviewer for raising this point. The upper panels of Fig. 9c and e show the 4C-seq signals of both viewpoint of the indicated RDH gene and the region around the gene. As the reviewer pointed out, while the 4C-seq signals of histone gene cluster, which is located at different chromosome of the viewpoint, were clearly decreased in EAF1-mutant cells, the 4C-seq signals around the viewpoint were slightly affected in EAF1-mutant cells (Fig. 9c–f). These results indicate that inter-chromosome interaction between the two histone gene clusters was decreased in EAF1-mutant cells, however; intra-chromosome interaction in each of the two histone gene clusters was slightly affected in EAF1-mutant cells. We clarified the “viewpoint” in the upper panels of revised Fig. 9c and e. Now, we have mentioned this issue in the Result section and stated: “Intriguingly, inter-chromosome interaction between the two histone gene

clusters was decreased in EAF1-mutant cells (Fig. 9c–f), however; intra-chromosome interaction in each of the two histone gene clusters was slightly affected in EAF1-mutant cells (Fig. 9c–f). Taken together, these results suggest that Mediator’s interaction with LEC and CBs’ resulting association with HLBs were more critical for appropriate inter-chromosome association between the two RDH gene loci than intra-chromosome association in each of the two RDH gene loci, consistent with a previous report showing that CBs are involved in higher-order chromosome conformation⁴⁴.” (Page 22, lines 14–21)

13. In Figure 10, 'poximal' should be 'proximal'.

We thank the reviewer for pointing out the typographical error in original Fig. 10. We corrected this in revised Fig. 10.

Reviewer #2 (Remarks to the Author):

In this study Suzuki et al set out to examine the mechanisms of transcription termination of non-polyadenylated mRNAs encoded by replication-dependent histone genes (RDHs). As a start they found that Pol II is paused at the TES proximal region (TER) of many genes including RDH and snoRNA genes. This pausing at non-polyadenylated genes was termed TES proximal pausing (TPP). In the RDH genes Pol II paused before a stem-loop structure was formed. Next, they KD proteins involved in 3'-end processing of RDH and found that KD of NELF-E or CBP80 significantly decreased TPP at RDH and snRNA genes, but not at snoRNA genes. Since they had previously found that Mediator subunit MED26 is also involved in this termination, they generated a cell line containing a MED26 mutant that cannot interact with the Little Elongation Complex (LEC). Also, they mutated AF4 and AFF4 which are subunits of the Super Elongation Complex (SEC). WT AF4 and AFF4, but not the mutated forms, interacted with MED26. The interactions of WT and mutant EAF1 were also tested for many components, showing that a mutation in EAF1 specifically abrogates Mediator’s interaction with LEC–CBC–NELF, but not with the formation of LEC. Expression of snRNA and RDH genes was downregulated in EAF1-mutant cells as was 3'-end processing of RDH transcripts. Importantly, TPP of RDH genes disappeared in EAF1-mutant cells and read-through of Pol II increased substantially (+200bp), showing that Mediator’s interaction with LEC–CBC–NELF is required for TPP and 3'-end processing. Next, the authors examined whether mediator plays a role in the interaction between CBs and HLBs and found a decrease in the overlap between the two bodies in EAF1 and MED26 mutant cells. The data suggested that the interaction between

Mediator and LEC is required for CB-HLB association. Imaging of the bodies showed that Mediator localized between CBs and HLBs. Next, using the "in situ biotin-labelling of protein with high throughput DNA sequencing" technique they developed, they identified the genomic regions with which CBs associate, which included the RDH genes. Interactions were reduced in EAF1 and ICE1 mutant cells and loss of TPP at RDH genes was observed. Finally, FISH analysis showed that CB association (and not HLB) with RDH genes requires Mediator's interaction with LEC, and 4C-seq showed that this interaction is required for inter-chromosome association.

Altogether, in this study the authors delineate a compelling model in which LEC recruitment by Mediator plays a role in CB-HLB association and facilitates 3'-end processing of RDHs. They also define the areas in the genome associate with CBs. This is a very elaborate study with much well-presented data obtained using a multitude of high-end approaches. I find the study clear and convincing and worthwhile for publication in Nature Communications pending a few comments below.

1. Fig. 3b,c: What are Mw of the proteins? Why do the mutants run as several bands and why are they heavier in MW, shouldn't they run faster in the gel? The bands are very strong for the WT as well, and it seems that in b there are 2 bands for the WT – why, and which of the two is IP'd (in c we see two bands as well so same question)?

We agree with the reviewer's comment. In accordance with this, we added the MW in revised Fig. 3b and c. As we indicated in the Result section, in mut of FLAG-MED26-NTD, the 61st Arg (R) and 62nd Lys (K) are substituted to Ala (A) residues. As the reviewer pointed out, FLAG-MED26-NTD mut (R61A, K62A) has a larger molecular weight than FLAG-MED26-NTD wt for the following reason. We subcloned cDNA encoding FLAG-MED26-NTD wt into pBacPak8 (Clontech) and cDNA encoding FLAG-MED26-NTD mut into pFASTBac HTb (Clontech). Both FLAG-MED26 NTD wt and mut are expressed with a Hexa-Histidine (His⁶) tag at the N-terminus of the proteins; however, the linker between the His⁶ tag and FLAG tag of MED26-NTD mut (R61A, K62A) is longer than that of MED26-NTD wt because of the difference of the vector backbone.

We added an explanation of this in the figure legends of revised Fig. 3b and c as follows: "F-MED26-NTD mut has a larger molecular weight than F-MED26-NTD wt, since the linker between the N-terminal hexa-Histidine tag and the FLAG tag of MED26-NTD mut is longer than that of MED26-NTD wt." (Page 52, lines 8–11)

2. Fig. 6a, b – would be nice to see an enlarged are of the images in general to get to see the

spatial relationship between CBs and HLBs, and more so to see a visual example of the reduction in area (not only in the plots).

We thank the reviewer for raising this issue. As the reviewer suggested, we showed enlarged images in revised Fig. 6a and b to clarify the spatial relationship between CBs and HLBs.

3. Fig. 7a – "Mediator components MED26 and MED1, and the LEC components ICE1 and ELL colocalized with CBs in HeLa cells" – actually for ICE1, the bottom nucleus does not show much colocalization. Same comment for the NPAT images.

We thank the reviewer for raising this point. As the reviewer pointed out, we observed that some NPAT particles (HLBs) colocalize with CBs, but other particles do not (original Fig. 7a).

As shown in Fig. 6c and d, we found that about 50% of Coilin particles (CBs) associated with NPAT particles (HLBs) in wild-type cells. Notably, it has been shown that RDH genes are transcribed at S phase and CBs associate with RDH genes more frequently during late G₁ and S phase¹². Consistent with the previous findings, we found that the RDH genes with which CBs associate are transcribed at much higher levels than other RDH genes (new Fig. 8g). These results indicate that CBs' association with HLBs is involved in the transcription activation of RDH genes. Because we performed immunofluorescence analysis using asynchronized cells, it is possible that we could observe more CBs associated with HLBs using the cells synchronized at S phase.

As the reviewer pointed out, some ICE1 particles colocalized with CBs, but others did not. We performed microscopic analysis to elucidate the relative localization of LEC in CBs or HLBs and found that LEC mainly colocalized with CBs rather than HLBs (new Fig. 7c, d). We next calculated the ratio of the particles of LEC components ICE1 or ELL colocalized with CBs, HLBs or CBs associated with HLBs. As shown in the right panel, in contrast to ELL, we found that about 35% of ICE1 particles have not been colocalized with CBs, CBs associated with

HLBs or HLBs (indicated as "Alone"). Because it has been shown that ICE1 has a role in processes other than transcription regulation, such as nonsense-mediated mRNA decay¹³, it is

possible that ICE1 particles that do not colocalize with CBs, CBs associated with HLBs or HLBs have roles other than in transcription regulation.

4. Fig. 9a suggest to enlarge, really hard to see what the authors are pointing at.

We appreciate the reviewer's suggestion. Accordingly, we show magnified insets in revised Fig. 9a and b.

5. Page 12 line 20,21 – "meta-gene analysis of RDH gene transcripts revealed that increased RDH transcripts were detected downstream" – increased what? Levels?

We apologize for the confusion. To aid understanding, we added a boxplot showing fold-change expression of gene-body transcripts and read-through transcripts in EAF1-mutant cells (new Fig. 4f). We also stated in the Result section that: "meta-gene analysis of RDH gene transcripts revealed that RDH transcripts of the gene body (GB) were decreased, and read-through (RT) transcripts were increased at RDH genes in EAF1-mutant cells (Fig. 4e, f)". (Page 14, lines 21–23)

Reviewer #3 (Remarks to the Author):

The authors propose a function for pol II pausing regulated by mediator at histone genes that is relevant to transcription termination. They have previously presented data on other non-polyadenylated RNAs.

A major issue with this paper is that they do not discuss the major advances in transcription termination on these genes which has occurred over the past two years. Several years ago Price and coworkers (Guo et al., 2014, Adamson and Price, 2003) and Tora (Anamika et al., 2012) demonstrated that pol II stalls shortly after the histone processing signals and terminates. Both Bentley and West's group for histone mRNAs have recently shown transcription termination on histone genes occurs shortly after transcribing past the processing site by activation of a protein phosphatase, leading to polymerase stalling and termination (Eaton and West, 2020, Eaton et al., 2020, Eaton et al., 2018, Cortazar et al., 2019), well beyond the proposed pause site in this paper and Wagner (Huang et al., 2020) has shown similar results with a different phosphatase on snRNA genes.

We appreciate the reviewer for raising these critical points. As the reviewer suggested, we cited all of these papers and discussed the recent progress in the revised version of the manuscript.

Adamson et al. (2003) established an *in vitro* transcription system and showed that Pol II is arrested 32 to 35 nucleotides downstream of transcription end sites (TESs) on RDH genes of *Drosophila melanogaster*¹⁴. Anamika et al. (2012) applied ChIP-seq and GRO-seq, and showed that Pol II pauses shortly downstream of TESs of RDH genes^{15, 16}. Eaton et al. (2018, 2020) and Cortazar et al. (2019) performed ChIP-seq, RNA-seq and mNET-seq, and showed that protein phosphatase 1 (PP1) dephosphorylates Spt5, a component of DRB-sensitivity inducing factor (DSIF), and decelerates Pol II elongation downstream of the TESs of protein-coding genes including RDH genes^{17, 18, 19, 20}. They proposed the “unified allosteric/torpedo model” or “sitting duck torpedo model” for transcription termination, in which deceleration of Pol II beyond TES is required for subsequent 5'→3' degradation of RNA associated with Pol II. We cited these papers in the Introduction section and stated: “Several lines of evidence have indicated that Pol II is arrested or pauses downstream of transcription end sites (TESs) of RDH genes. *In vitro* transcription assay with RDH genes of *Drosophila melanogaster* revealed that Pol II is arrested 32 to 35 nucleotides downstream of TESs^{11, 12}. Genome-wide analysis including ChIP-seq of Pol II and Global Run On sequencing (GRO-seq) using human cells showed that Pol II pauses downstream of TESs of RDH genes^{13, 14}. Intriguingly, recent reports have shown that protein phosphatase 1 (PP1) dephosphorylates Spt5, a component of DRB-sensitivity inducing factor (DSIF), and decelerates Pol II elongation downstream of TESs of protein-coding genes including RDH genes^{15, 16, 17, 18}. These reports propose a model that the deceleration of Pol II beyond TESs is required for subsequent 5'→3' degradation of RNA associated with Pol II.” (Page 4, lines 2–11)

All the previous papers above showed that Pol II is arrested or pauses downstream of transcription end sites (TESs) of RDH genes; in contrast, our PRO-seq analysis revealed that Pol II pauses immediately upstream of the TESs of RDH genes (revised Fig. 1c–e and revised Supplementary Fig. 2a, b, k). Notably, Pol II pauses within the stem-loop region of most RDH genes (new Supplementary Fig. 3). As shown in new Supplementary Fig. 1a, PRO-seq enables precise detection of the 3'-end of nascent transcripts at single-nucleotide resolution. Moreover, PRO-seq detects nascent RNAs transcribed by elongating or pausing Pol II, but not those of arrested or stalled Pol II, indicating that our results do not contradict the previous reports based on experiments including *in vitro* transcription assay, ChIP-seq, GRO-seq, RNA-seq and mNET-seq and that we precisely detected the 3' end of the nascent transcripts of RDH genes. We called such pausing of Pol II “TES proximal pausing” (TPP). We described this issue in the Result section and stated: “In previous studies, ChIP-seq of Pol II and GRO-seq revealed that Pol II pauses downstream of RDH genes^{13, 14}. Because PRO-seq enables precise detection of the

3' end of nascent transcripts, which are produced by elongating or paused polymerases, but not stalled or arrested polymerases, it is thought that our results do not contradict the previous reports based on ChIP-seq and GRO-seq. Taking the findings together, our analysis revealed that Pol II pauses immediately upstream of the TES of RDH genes.” (Page 10, lines 9–14)

We also found that knockdown of NELF-E, one of the components of NELF, or CBP80 significantly abolished Pol II TPP at RDH genes (Fig. 2a–c, and Supplementary Fig. 4a, b). Because we previously found that MED26 plays a role in transcription termination of RDH genes through direct interaction with LEC–CBC–NELF³, we thought it possible that Mediator’s interaction with LEC contributes to TPP and subsequent 3’-end processing of RDH gene transcripts. To test this possibility, we mutated the MED26 binding site of EAF1, which plays a role as a docking site of LEC on Mediator. In this EAF1-point-mutant cell line, Mediator’s interaction with LEC–CBC–NELF was specifically abolished, but that of SEC was not (Fig. 3e). Intriguingly, EAF1 mutation drastically interfered with Pol II TPP at RDH genes (Fig. 5a–e), resulting in increased aberrant unprocessed RDH gene transcripts (Fig. 4a–e, new Fig. 4f). On the basis of our findings, we propose a model for the role of Mediator in TPP of Pol II and subsequent 3’-end processing of RDH genes. In this model, MED26-containing Mediator plays a role in TPP through interacting with LEC–CBC–NELF. Thus, our results raise the possibility that TPP is a key checkpoint of the transition from transcription elongation to transcription termination for appropriate 3’-end processing of RDH genes and the production of non-polyadenylated RDH transcripts (revised Fig. 10). We have mentioned our progress in the Discussion section and stated: “Several lines of evidence have indicated that Pol II is arrested or pauses downstream of TESs of RDH genes^{11, 12, 13, 14}. Since PRO-seq enables precise detection of the 3’ end of nascent transcripts, our results do not contradict previous reports based on *in vitro* transcription assay, ChIP-seq and GRO-seq. PRO-seq analysis revealed that Pol II pauses immediately upstream of TESs of RDH and snRNA genes (Fig. 1). NELF, which is responsible for the PPP of Pol II, has been shown to colocalize with CBs and plays a role in the 3’-end processing of RDH genes¹⁹. Intriguingly, TPP was decreased in EAF1-mutant cells or in cells in which CBP80 or NELF was depleted (Fig. 2 and 5). These results raised the possibility that TPP plays a critical role in the 3’-end processing of RDH and snRNA genes. Although the mechanism by which NELF regulates the 3’-end processing of RDH gene transcripts has not been fully elucidated¹⁹, our results raise the possibility that Mediator’s interaction with LEC–CBC–NELF is required for TPP and subsequent 3’-end processing of RDH and snRNA genes. It has been shown that PPP is a key checkpoint for gene expression through regulating the transition from transcription initiation to productive elongation⁶³. Our results raise the possibility that TPP is a key checkpoint process regulating the transition from transcription

elongation to transcription termination (Fig. 10).” (Page 24, lines 19–24, and Page 25, lines 1–9)

Huang et al. (2020) performed PRO-seq using DL1 cells of *Drosophila melanogaster* and showed that Integrator recruits a different phosphatase, protein phosphatase 2A (PP2A), to protein-coding genes and snRNA genes²¹. They showed that PP2A dephosphorylates Spt5 and Pol II CTD and inhibits release of the pausing of Pol II, leading to transcription termination. Although we took advantage of the PRO-seq data, we did not find TPP at snRNA/snoRNA and RDH genes. Considering that we found TPP at RDH and snRNA genes using PRO-seq data of human cells in Sathyan et al. (2019) (Supplementary Fig. 2k)²², it is possible that transcription regulation by TPP differs between the species.

Notably, we took advantage of the mNET-seq results of Eaton et al. (2018) and compared the data to our PRO-seq data to see whether TPP is detected at non-polyadenylated genes

including snRNA/snoRNA and RDH genes. As shown in the right panel, we found TPP at a subset of snRNA/snoRNA genes in human HCT116 cells, although we did not find TPP at RDH genes. Furthermore, we took advantage

of the mNET-seq results of Arnold et al. (2021) to see whether TPP is detected at RDH genes²³. As shown in the right panel, we found TPP at a subset of RDH genes in human K562 cells.

The groups of West and Bentley showed that PP1 dephosphorylates Spt5 and decelerates Pol II transcription downstream of polyadenylation signal (PAS) of protein-coding genes and RDH genes^{17, 18, 19, 20}. In addition, the groups of Adelman and Wagner showed that Integrator recruits a different phosphatase, PP2, to dephosphorylate Spt5 and decelerates Pol II transcription downstream of the PAS of protein-coding genes and snRNA genes²¹. These reports propose a model that the slowing down of Pol II beyond TES is required for transcription termination, raising the possibility that protein phosphatases including PP1 or PP2A dephosphorylate Spt5 of CBC–NELF–DSIF and play a role in TPP at RDH and snRNA genes. We have mentioned this issue in the Discussion section and stated: “Intriguingly, recent reports have shown that protein phosphatase 1 (PP1) or protein phosphatase 2A (PP2A)

dephosphorylates Spt5, a component of DSIF, and/or Pol II CTD to decelerate Pol II elongation downstream of the TES of RDH or snRNA genes, respectively^{15, 16, 17, 18}. These reports suggest a model in which the deceleration of Pol II beyond TES is a prerequisite for subsequent 5'→3' degradation of RNAs associated with Pol II. These reports and our findings raise the possibility that PP1 or PP2A dephosphorylates Spt5, a component of NELF/DSIF, and/or Pol II CTD to promote TPP at RDH or snRNA genes, respectively. ” (Page 25, lines 10–16)

Major problem:

1. The PRO-SEQ data, which the authors use to measure where the pause sites are, is clearly seriously contaminated with cellular RNAs. Each histone gene shows a major peak right at the end of the transcript, and the sequence in Fig. 3I shows it is at the place where histone mRNA is processed. Thus that peak is processed histone mRNA and not paused transcripts. Similar results are found for all the histones (Fig. 5A-D), and snRNA genes, and even for 28S rRNA. These cannot be polymerase pause sites relevant to processing and termination, since the processing signals haven't been transcribed yet and the 3' stemloop in histone genes would still be inside pol II. Given this technical issue any discussion of termination in this paper is unwarranted. Note that this is not an issue for the polyadenylated mRNAs since any reads would not have been mapped.

We appreciate the reviewer for raising this issue. To rule out the possibility that our PRO-seq detected contaminated RNAs from cells, we checked the RNA purification process of PRO-seq. In this experiment, we permeabilized the cells and isolated intact nuclei, and then performed a nuclear run-on reaction. In this reaction, transcription-engaging RNA polymerases are paused by the incorporation of biotinylated NTPs (new Supplementary Fig. 1a). We purified the resulting biotinylated RNAs through avidin-biotin purification. The purified RNAs were reverse-transcribed to cDNA and subjected to the construction of a cDNA library for next-generation sequencing. We did not detect any amplified cDNAs in library construction in the absence of biotin-NTPs (new Supplementary Fig. 1b), indicating that we detected only biotinylated RNAs and eliminated contaminated RNAs from the cells.

We also compared the genome browser tracks showing the distribution of 3' ends of total RNA-seq reads and those of PRO-seq reads. At RDH, snRNA/snoRNA and rRNA genes, the peaks of TPP observed in PRO-seq were not detected in RNA-seq, opposing the possibility that the peaks of TPP are derived from contaminated RNAs (new Supplementary Fig. 1c–g).

2. The proposal that Cajal bodies and HLBs exchange components during histone gene

expression seems very unlikely. While in some cells coilin is present in HLBs, in many cells the two bodies are separate. U7 snRNP is primarily found in HLBs, and not in all Cajal bodies, although it is certainly possible that U7 snRNP passes through the Cajal body as part of its maturation process. In the initial Frey and Matera paper (ref. 27) the point of the paper was that U7 snRNA was found only in the Cajal bodies near histone genes (these are the HLBs, and this paper was published before the HLB had been defined). I am not aware of any data that NPAT or FLASH are found in Cajal bodies, except for HLBs that contain some coilin. It has recently been definitively shown that histone gene transcription and processing occurs inside the HLB (Kemp et al., 2021).

We appreciate the reviewer for raising this issue. Although the reviewer pointed out that U7 snRNP is primarily present at HLBs and not in all Cajal bodies, numerous reports have indicated that 3'-end processing factors for RDH genes including the components of NELF, U7 snRNP and HCC are enriched in CBs.

For instance, it was reported that Coilin can form a complex with the U7 snRNP²⁴. Interactome analysis of Coilin also identified a number of constituents of CBs, which include LSM10 and LSM11, the components of U7 snRNP²⁵. Although a recent report and our immunofluorescence analysis shown in new Fig. 7f indicate that FLASH mainly colocalizes with HLBs⁷, FLASH has been shown to be required for the formation of CBs²⁶. In addition, it has been shown that Negative elongation factor (NELF) and one of the components of HCC, CSTF64, colocalize with CBs and facilitate the 3'-end processing of RDH genes^{3,4}. It was also demonstrated that U2 snRNP, which is present at CBs, binds to pre-mRNAs of RDH genes to facilitate U7 snRNP-dependent 3'-end processing²⁷. Moreover, genome-wide 4C-seq analysis revealed that CBs associate with RDH gene clusters and play a critical role in both the transcription of RDH genes and the genomic conformation of RDH gene clusters¹⁰. Thus, previous reports indicate that (i) 3'-end processing factors for RDH genes are enriched in CBs, (ii) CBs are frequently and physically associated with HLBs, and (iii) CB formation is required for both the genomic conformation of RDH gene loci and the transcription of RDH genes, supporting our idea that 3'-processing factors for RDH genes are directly supplied from CBs to HLBs. We cited the papers above and stated the following in the Introduction section:

“Intriguingly, HLBs and CBs frequently and physically associate with each other, suggesting that there is a functional link between them³⁹. Notably, interactome analysis of Coilin identified a number of constituents of CBs including LSM10 and LSM11, which are components of U7 snRNP^{40,41}. It has also been shown that FLASH, which colocalizes at HLBs, is required for the formation of CBs⁴². In addition, it has been shown that Negative elongation factor (NELF) and CSTF64 colocalize with CBs and facilitate 3'-end processing of RDH genes^{19,20}. It was also

demonstrated that U2 snRNP, which is present at CBs, binds to pre-mRNAs of RDH genes to facilitate U7 snRNP-dependent 3'-end processing⁴³. Moreover, genome-wide 4C-seq analysis revealed that CBs associate with HLBs containing RDH gene clusters and play critical roles in both the transcription of RDH genes and the genomic conformation of RDH gene clusters⁴⁴. Thus, CBs have been shown to contain the 3'-end processing factors for RDH genes, including the components of NELF, U7 snRNP and HCC^{27, 38, 39, 45}. Considering that (i) 3'-end processing factors for RDH genes are enriched in CBs and (ii) CBs are frequently and physically associated with HLBs^{46, 47}, 3'-processing factors for RDH genes may be directly supplied from CBs to HLBs.” (Page 5, lines 22–24, and Page 6, lines 1–11)

Consistent with our expectations, we found that LSM11, a component of U7 snRNP, was localized in the region between CBs and HLBs, similarly to Mediator (new Fig. 7e). We performed antibody-based *in situ* biotinylation with anti-LSM11 antibody to test whether EAF1 mutation affects LSM11's association with RDH gene loci. As shown in new Fig. 8e and f, and new Supplementary Fig. 11e, LSM11's association was reduced in EAF1-mutant cells. These results suggest that the recruitment of U7 snRNP, which is enriched in CBs, to RDH gene loci was impaired according to the decrease in CBs' association with HLBs in EAF1-mutant cells, supporting our idea that a subset of 3'-processing factors for RDH genes is supplied from CBs to HLBs.

As the reviewer pointed out, recently super-resolution light microscopy of the *Drosophila* HLB revealed that RDH gene transcription and processing occur inside the HLB⁷. In this report, it was shown that HLBs have a “core-shell” organization in which the internal core contains transcriptionally active RDH genes. The N-terminus of Mxc, a fly homologue of human NPAT, which is required for Mxc oligomerization, HLB assembly and RDH gene transcription, is enriched in the HLB core. In contrast, the C-terminus of Mxc is enriched in the HLB outer shell where FLASH is enriched through direct interaction with the C-terminus of Mxc. Thus, in this report, it was speculated that U7 snRNP bound to FLASH translocates from the shell to the core to bind and process the nascent pre-mRNA. Consistent with this report, we found that FLASH mainly colocalized at HLBs (new Fig. 7f, i); in contrast, LSM11 localized between CBs and HLBs (new Fig. 7e, i). Previous reports and our findings raise the possibility that LSM11 localizes outside the outer shell of HLBs or at CBs and translocates to the core of HLBs to process the nascent RDH gene transcripts. We have mentioned this issue in the Discussion section and stated: “Recently, it was shown that HLBs have a “core-shell” structure in which the internal core contains transcriptionally active RDH genes. The N-terminus of Mxc, which is a fly homologue of human NPAT, is enriched in the HLB core, and the C-terminus of Mxc is enriched in the HLB outer shell where FLASH is enriched through direct interaction with the C-terminus of Mxc⁶⁴. Thus, it was speculated that U7 snRNP bound to FLASH

translocates from the shell to the core to process the nascent pre-mRNA. Consistent with this report, we found that FLASH mainly colocalized at HLBs; in contrast, LSM11 localized between CBs and HLBs (Fig. 7). This previous report and our findings raise the possibility that U7 snRNP containing LSM11 localizes outside the outer shell of HLBs, which is close to CBs, and translocates to the core of HLBs to process the nascent RDH gene transcripts. Considering these previous findings and our results, it is highly possible that 3'-end processing factors are supplied from CBs to HLBs. ” (Page 26, lines 16–24, and Page 27, lines 1–3)

3. They define HLF, heat labile factor, incorrectly. Kolev and Steitz showed in 2006 that HLF is symplekin. The complex they define as HLF is actually the HCC (histone cleavage complex) which is part of the active U7 snRNP, which contains symplekin, CPSF73, CPSF100 and CstF64. This complex is bound to the core U7 snRNP through FLASH which specifically binds Lsm11 and the HCC. This complex was described in ref. 8 in their paper.

We appreciate the reviewer for pointing out these important points. As the reviewer pointed out, we corrected the Introduction section and stated: “The stem-loop and HDE are bound by stem-loop binding protein (SLBP) and U7 small nuclear ribonucleoprotein (snRNP), respectively^{5,6}. FLICE-Associated Huge Protein (FLASH) is a specific factor needed for 3'-end processing of RDH genes and binds to Histone pre-mRNA Cleavage Complex (HCC) composed of CPSF2, CPSF3, Symplekin and the 64 kDa subunit of Cleavage Stimulation Factor (CSTF64)^{5,8,9}. It has been shown that FLASH plays a critical role in the recruitment of U7 snRNP through interaction with U7 snRNA-associated Sm-like protein LSM11⁸. The HCC component CPSF3 functions as an endonuclease responsible for the 3'-end cleavage of RDH gene transcripts¹⁰.” (Page 3, lines 18–24, and Page 4, line 1)

4. Reference List

- Adamson, T. E. & Price, D. H. (2003). Cotranscriptional processing of *Drosophila* histone mRNAs. *Mol Cell Biol*, 23, 4046-55.
- Anamika, K., Gyenis, A., Poidevin, L., Poch, O. & Tora, L. (2012). RNA polymerase II pausing downstream of core histone genes is different from genes producing polyadenylated transcripts. *PLoS. ONE*, 7, e38769.
- Cortazar, M. A., Sheridan, R. M., Erickson, B., Fong, N., Glover-Cutter, K., Brannan, K. & Bentley, D. L. (2019). Control of RNA Pol II Speed by PNUTS-PP1 and Spt5 Dephosphorylation Facilitates Termination by a "Sitting Duck Torpedo" Mechanism. *Mol Cell*, 76, 896-908 e4.

Eaton, J. D., Davidson, L., Bauer, D. L. V., Natsume, T., Kanemaki, M. T. & West, S. (2018). Xrn2 accelerates termination by RNA polymerase II, which is underpinned by CPSF73 activity. *Genes Dev*, 32, 127-139.

Eaton, J. D., Francis, L., Davidson, L. & West, S. (2020). A unified allosteric/torpedo mechanism for transcriptional termination on human protein-coding genes. *Genes Dev*, 34, 132-145.

Eaton, J. D. & West, S. (2020). Termination of Transcription by RNA Polymerase II: BOOM! *Trends Genet*, 36, 664-675.

Guo, J., Turek, M. E. & Price, D. H. (2014). Regulation of RNA polymerase II termination by phosphorylation of Gdown1. *J. Biol. Chem*, 289, 12657-12665.

Huang, K. L., Jee, D., Stein, C. B., Elrod, N. D., Henriques, T., Mascibroda, L. G., Baillat, D., Russell, W. K., Adelman, K. & Wagner, E. J. (2020). Integrator Recruits Protein Phosphatase 2A to Prevent Pause Release and Facilitate Transcription Termination. *Mol Cell*.

Kemp, J. P., Jr., Yang, X. C., Dominski, Z., Marzluff, W. F. & Duronio, R. J. (2021). Superresolution light microscopy of the *Drosophila* histone locus body reveals a core-shell organization associated with expression of replication-dependent histone genes. *Mol Biol Cell*, 32, 942-955.

We appreciate the reviewer for suggesting the citation of these important papers. We cited all the papers listed above and discussed the recent progress in the field in the revised manuscript.

References

1. Matera AG, Terns RM, Terns MP. Non-coding RNAs: lessons from the small nuclear and small nucleolar RNAs. *Nature reviews Molecular cell biology* **8**, 209-220 (2007).
2. Sato K, Akiyama M, Sakakibara Y. RNA secondary structure prediction using deep learning with thermodynamic integration. *Nature communications* **12**, 941 (2021).
3. Takahashi H, *et al.* The role of Mediator and Little Elongation Complex in transcription termination. *Nature communications* **11**, 1063 (2020).
4. Narita T, *et al.* NELF interacts with CBC and participates in 3' end processing of replication-dependent histone mRNAs. *Molecular cell* **26**, 349-365 (2007).
5. Hallais M, *et al.* CBC-ARS2 stimulates 3'-end maturation of multiple RNA families and favors cap-proximal processing. *Nature structural & molecular biology* **20**, 1358-1366 (2013).
6. Hur W, *et al.* CDK-Regulated Phase Separation Seeded by Histone Genes Ensures Precise Growth and Function of Histone Locus Bodies. *Developmental cell* **54**, 379-394 e376 (2020).
7. Kemp JP, Jr., Yang XC, Dominski Z, Marzluff WF, Duronio RJ. Superresolution light microscopy of the *Drosophila* histone locus body reveals a core-shell organization associated with expression of replication-dependent histone genes. *Molecular biology of the cell* **32**, 942-955 (2021).
8. Ma T, *et al.* Cell cycle-regulated phosphorylation of p220(NPAT) by cyclin E/Cdk2 in Cajal bodies promotes histone gene transcription. *Genes & development* **14**, 2298-2313 (2000).
9. Zhao J, *et al.* NPAT links cyclin E-Cdk2 to the regulation of replication-dependent histone gene transcription. *Genes & development* **14**, 2283-2297 (2000).
10. Wang Q, *et al.* Cajal bodies are linked to genome conformation. *Nature communications* **7**, 10966 (2016).

11. Imada T, Shimi T, Kaiho A, Saeki Y, Kimura H. RNA polymerase II condensate formation and association with Cajal and histone locus bodies in living human cells. *Genes to cells : devoted to molecular & cellular mechanisms* **26**, 298-312 (2021).
12. Shopland LS, Byron M, Stein JL, Lian JB, Stein GS, Lawrence JB. Replication-dependent histone gene expression is related to Cajal body (CB) association but does not require sustained CB contact. *Molecular biology of the cell* **12**, 565-576 (2001).
13. Baird TD, *et al.* ICE1 promotes the link between splicing and nonsense-mediated mRNA decay. *eLife* **7**, (2018).
14. Adamson TE, Price DH. Cotranscriptional processing of Drosophila histone mRNAs. *Molecular and cellular biology* **23**, 4046-4055 (2003).
15. Anamika K, Gyenis A, Poidevin L, Poch O, Tora L. RNA polymerase II pausing downstream of core histone genes is different from genes producing polyadenylated transcripts. *PloS one* **7**, e38769 (2012).
16. Anamika K, Gyenis A, Tora L. How to stop: the mysterious links among RNA polymerase II occupancy 3' of genes, mRNA 3' processing and termination. *Transcription* **4**, 7-12 (2013).
17. Cortazar MA, *et al.* Control of RNA Pol II Speed by PNUTS-PP1 and Spt5 Dephosphorylation Facilitates Termination by a "Sitting Duck Torpedo" Mechanism. *Molecular cell* **76**, 896-908 e894 (2019).
18. Eaton JD, Davidson L, Bauer DLV, Natsume T, Kanemaki MT, West S. Xrn2 accelerates termination by RNA polymerase II, which is underpinned by CPSF73 activity. *Genes & development* **32**, 127-139 (2018).
19. Eaton JD, Francis L, Davidson L, West S. A unified allosteric/torpedo mechanism for transcriptional termination on human protein-coding genes. *Genes & development* **34**, 132-145 (2020).
20. Eaton JD, West S. Termination of Transcription by RNA Polymerase II: BOOM! *Trends*

in genetics : TIG **36**, 664-675 (2020).

21. Huang KL, *et al.* Integrator Recruits Protein Phosphatase 2A to Prevent Pause Release and Facilitate Transcription Termination. *Molecular cell* **80**, 345-358 e349 (2020).
22. Sathyan KM, McKenna BD, Anderson WD, Duarte FM, Core L, Guertin MJ. An improved auxin-inducible degron system preserves native protein levels and enables rapid and specific protein depletion. *Genes & development* **33**, 1441-1455 (2019).
23. Arnold M, Bressin A, Jasnovidova O, Meierhofer D, Mayer A. A BRD4-mediated elongation control point primes transcribing RNA polymerase II for 3'-processing and termination. *Molecular cell* **81**, 3589-3603 e3513 (2021).
24. Bellini M, Gall JG. Coilin can form a complex with the U7 small nuclear ribonucleoprotein. *Molecular biology of the cell* **9**, 2987-3001 (1998).
25. Machyna M, *et al.* The coilin interactome identifies hundreds of small noncoding RNAs that traffic through Cajal bodies. *Molecular cell* **56**, 389-399 (2014).
26. Barcaroli D, *et al.* FLASH is an essential component of Cajal bodies. *Proceedings of the National Academy of Sciences of the United States of America* **103**, 14802-14807 (2006).
27. Friend K, Lovejoy AF, Steitz JA. U2 snRNP binds intronless histone pre-mRNAs to facilitate U7-snRNP-dependent 3' end formation. *Molecular cell* **28**, 240-252 (2007).

REVIEWER COMMENTS

Reviewer #1 (Remarks to the Author):

The authors have satisfactorily answered my concerns.

Reviewer #2 (Remarks to the Author):

I am satisfied with the author's response to my queries.

Reviewer #3 (Remarks to the Author):

The authors have responded to the previous reviews by acknowledging the work of other investigators who have characterized the transcription and transcription termination at the 3' end of histone genes. However, they have not provided an explanation for why their results differ from all others who have studied this process, where there is general agreement that RNA polymerase accumulates and terminates 3' of the processing site. Their results also seem to disagree with their previous results published last year in Nature Communications, where they did not observe these peaks at the 3' ends of histone mRNAs, snRNAs and snoRNAs reported here using PRO-SEQ.

Specific Comments:

1. As I read their method, it differs substantially from some of the other methods used. Most methods looking at the nascent transcripts have rigorously taken steps to assure they are looking at nascent RNAs bound to chromatin. As I read their method, this doesn't seem to be the case, since start with a relatively crude nuclear preparation (prepared by permeabilizing cells). What is striking in their results is they find the RNA polymerase stalled precisely (to the nt level) at the histone stemloop. There are two possible artifacts that could explain this observation: 1. Contamination with histone mRNA in the bound RNA fraction. I think they argue convincingly that this is unlikely. 2. Modification of the 3' end of histone mRNA by biotinylated oligonucleotides which I think is very possible, and would account for the sharp peak at the 3' end of the histone mRNA. Histone mRNAs are processed 5 nts after the stem-loop. They then undergo removal of 2-3 nts by an exonuclease, and there are cycles of uridylation and exonucleolytic degradation resulting in the 3' end of a large fraction of histone mRNAs ending in uridine which is added postranscriptionally (Lackey et al., 2016; Welch et al., 2015). If this modification is occurring in their in vitro incubation, the result would be precisely what they observe, a biotinylated nt or nts added precisely after the stemloop. The simple experiment to address this is to carry out the PRO-SEQ experiment in the presence of a-amanitin, which will block all pol II transcription, and make sure it inhibits all these peaks are the very end of RNAs. Many other RNAs in the cell undergo some level of terminal modification as well (addition of U's or A's) and this may account for the sharp peaks observed at the end of 28S rRNA, and snRNAs (Fig. S1). The previous results of others that showed polymerases accumulating 3' of the 3' end were not all the result of CHIP-SEQ or GRO-Seq that do not have nt resolution. The results of Eaton et al. were obtained using NET-SEQ which also gives nt resolution at the 3' end. This method, developed by Nojima and Proudfoot (Nojima et al. 2016), has been used extensively in similar studies by others. More confusing is that in their initial paper (Nature Comm. 11, 1063 (2020) they show in Fig. 4a PROSEQ done by a slightly different approach (Dounce homogenization and purification of nuclei, rather than permeabilization of nuclei). The results for histone genes look very different there with peaks throughout the transcript as expected, and they saw the same thing for U1 snRNA, U11 snRNA and U13 sno RNA (Fig. 4d-f). They definitely need to comment on the difference in these two results between their two papers.
2. In their new supplementary figure 2g, they show that "transcription" pauses at the 3' end of snoRNAs present in introns. These RNAs are formed by transcription of the pre-mRNA followed by processing to release the snoRNAs. They would never be labeled precisely by the method the authors used unless the labeling was posttranscriptional.

3. Their results do not make biological sense (which is OK since we are always finding surprising new things), but they don't try to explain how their results might be linked to histone pre-mRNA processing. If the polymerase stalls precisely at the stemloop, then there is 25 nt of nascent RNA inside the polymerase (not accessible to processing factors), and no processing factors can be recruited to the nascent mRNA. Results of Eaton et al. (also at the nt sequence level) shows pol II accumulating downstream of the processing site, which then allows the processing factors to identify the processing site, and gives them time to assemble on the nascent transcript.

4. The HLB is very specific and easy to identify since it contains NPAT, which is found only on histone genes (Kaya-Okur et al., 2019). Cajal bodies are much harder to unambiguously identify since some HLBs contain coilin (this doesn't make them Cajal bodies, which are better identified by the presence of scaRNAs. Cajal bodies are clearly involved in maturation of snRNAs so they will contain some U7 snRNP, but mature U7 snRNP is present concentrated in the HLB. Papers before 2007 (when the HLB was first described as distinct from Cajal bodies) referred to what we now know are HLBs as Cajal bodies. They do not cite (Barcaroli et al., 2006) where they identified bodies containing NPAT and FLASH at histone genes, which were not affected by manipulating coilin, and further studies by this group confirmed that there is a difference between the NPAT containing bodies and other Cajal bodies (Bongiorno-Borbone et al., 2008). These "cajal bodies" were clearly HLBs and not Cajal bodies, and that paper was adjacent to the one they cited.

There are not good reagents to track U7 snRNP in mammalian cells (good antibodies to Lsm10 and Lsm11 are not available). It is possible that once U7 snRNP is matured in Cajal bodies it then is actively directed to the HLB.

References

- Barcaroli, D., Bongiorno-Borbone, L., Terrinoni, A., Hofmann, T.G., Rossi, M., Knight, R.A., Matera, A.G., Melino, G., and De, L., V (2006). FLASH is required for histone transcription and S-phase progression. *Proc. Natl. Acad. Sci. U. S. A* 103, 14808-14812. 0604227103 [pii];10.1073/pnas.0604227103 [doi].
- Bongiorno-Borbone, L., De, C.A., Vernole, P., Finos, L., Barcaroli, D., Knight, R.A., Melino, G., and De, L., V (2008). FLASH and NPAT positive but not Coilin positive Cajal Bodies correlate with cell ploidy. *Cell Cycle* 7, 2357-2367. 6344 [pii].
- Kaya-Okur, H.S., Wu, S.J., Codomo, C.A., Pledger, E.S., Bryson, T.D., Henikoff, J.G., Ahmad, K., and Henikoff, S. (2019). CUT&Tag for efficient epigenomic profiling of small samples and single cells. *Nat Commun* 10, 1930. 10.1038/s41467-019-09982-5.
- Lackey, P.E., Welch, J.D., and Marzluff, W.F. (2016). TUT7 catalyzes the uridylation of the 3' end for rapid degradation of histone mRNA. *RNA* 22, 1673-1688. 10.1261/rna.058107.116.
- Welch, J.D., Slevin, M.K., Tatomer, D.C., Duronio, R.J., Prins, J.F., and Marzluff, W.F. (2015). EnD-Seq and AppEnD: sequencing 3' ends to identify nontemplated tails and degradation intermediates. *RNA* 21, 1375-1389. 10.1261/rna.048785.114.
- Nojima T, Gomes T, Carmo-Fonseca M, Proudfoot NJ. 2016. Mammalian NET-seq analysis defines nascent RNA profiles and associated RNA processing genome-wide. *Nat Protoc* 11: 413-428

POINT-BY-POINT RESPONSES TO THE REVIEWERS' COMMENTS

Please find our responses to the reviewers' comments listed below. The reviewers' comments are shown in a bold black font, and our responses to the comments are shown in green. Sentences quoted from the revised manuscript are shown in brown. The newly added text in the revised manuscript is shown in red.

Reviewer #1 (Remarks to the Author):

The authors have satisfactorily answered my concerns.

We are very pleased that our changes were approved by the reviewer.

Reviewer #2 (Remarks to the Author):

I am satisfied with the author's response to my queries.

We are very pleased that our changes were approved by the reviewer.

Reviewer #3 (Remarks to the Author):

The authors have responded to the previous reviews by acknowledging the work of other investigators who have characterized the transcription and transcription termination at the 3' end of histone genes. However, they have not provided an explanation for why their results differ from all others who have studied this process, where there is general agreement that RNA polymerase accumulates and terminates 3' of the processing site. Their results also seem to disagree with their previous results published last year in Nature Communications, where they did not observe these peaks at the 3' ends of histone mRNAs, snRNAs and snoRNAs reported here using PRO-SEQ.

Specific Comments:

1. As I read their method, it differs substantially from some of the other methods used. Most methods looking at the nascent transcripts have rigorously taken steps to assure they are looking at nascent RNAs bound to chromatin. As I read their method, this doesn't seem to be the case, since start with a relatively crude nuclear preparation (prepared by permeabilizing cells). What is striking in their results is they find the RNA polymerase stalled precisely (to the nt level) at the histone stemloop.

There are two possible artifacts that could explain this observation: 1. Contamination with histone mRNA in the bound RNA fraction. I think they argue convincingly that this is unlikely. 2. Modification of the 3' end of histone mRNA by

biotinylated oligonucleotides which I think is very possible, and would account for the sharp peak at the 3' end of the histone mRNA. Histone mRNAs are processed 5 nts after the stem-loop. They then undergo removal of 2-3 nts by an exonuclease, and there are cycles of uridylation and exonucleolytic degradation resulting in the 3' end of a large fraction of histone mRNAs ending in uridine which is added postranscriptionally (Lackey et al., 2016; Welch et al., 2015). If this modification is occurring in their in vitro incubation, the result would be precisely what they observe, a biotinylated nt or nts added precisely after the stemloop. The simple experiment to address this is to carry out the PRO-SEQ experiment in the presence of α -amanitin, which will block all pol II transcription, and make sure it inhibits all these peaks are the very end of RNAs.

Many other RNAs in the cell undergo some level of terminal modification as well (addition of U's or A's) and this may account for the sharp peaks observed at the end of 28S rRNA, and snRNAs (Fig. S1). The previous results of others that showed polymerases accumulating 3' of the 3' end were not all the result of CHIP-SEQ or GRO-Seq that do not have nt resolution. The results of Eaton et al. were obtained using NET-SEQ which also gives nt resolution at the 3' end. This method, developed by Nojima and Proudfoot (Nojima et al. 2016), has been used extensively in similar studies by others.

We thank the reviewer for raising the possibility that biotinylated NTPs are added post-transcriptionally in the PRO-seq and the resulting 3' end of the intermediate transcripts is detected as 3' pausing sites. To address this issue, we performed the PRO-seq analysis in the presence of α -amanitin, which blocks transcription by Pol II but not Pol I and Pol III. As shown in new Supplementary Fig. 4a and b, the promoter proximal pausing (PPP) of protein-coding genes such as *Fos* was markedly decreased by α -amanitin, which was accompanied by marked decreases of the elongated transcripts of the genes. Consistent with the ability of α -amanitin to block the activity of Pol II, but not that of Pol I, the PRO-seq peaks of rRNA genes, which are transcribed by Pol I, were not affected by α -amanitin (new Supplementary Fig. 4c). As we expected, α -amanitin markedly decreased the signals of the transcription end site (TES) proximal pausing (TPP) at RDH, snRNA and both independently transcribed snoRNA and intronic snoRNA genes (new Supplementary Fig. 4d-m). The decrease of TPP at RDH genes was accompanied by the marked decrease of PPP at the genes (new Supplementary Fig. 4h). These results indicate that the TPP as well as PPP observed at the RDH, snRNA and snoRNA genes are derived from Pol II and not a post-transcriptional modification. Of note, α -amanitin markedly, but not completely, blocked Pol II activity in our PRO-seq, consistent with the evidence that α -amanitin slows Pol II elongation and reduces substrate specificity through interactions with the trigger loop of Pol II but does not completely inhibit Pol II elongation

activity^{1,2,3}. (page 10, lines 23–24, and page 11, lines 1–24)

2. More confusing is that in their initial paper (Nature Comm. 11, 1063 (2020) they show in Fig. 4a PROSEQ done by a slightly different approach (Dounce homogenization and purification of nuclei, rather than permeabilization of nuclei). The results for histone genes look very different there with peaks throughout the transcript as expected, and they saw the same thing for U1 snRNA, U11 snRNA and U13 sno RNA (Fig. 4d-f). They definitely need to comment on the difference in these two results between their two papers.

We thank the reviewer for raising this issue. As the reviewer pointed out, we modified the protocol of PRO-seq and performed nuclei permeabilization in the current study instead of Dounce homogenization. In the original protocol of PRO-seq, permeabilization methods were recommended rather than Dounce homogenization because the recovery of transcripts is higher from permeabilized cells than from Dounce homogenized cells⁴. Therefore, we performed permeabilization methods in this study and detected TPP, which was not detected using Dounce homogenized cells (Fig. 1).

As the reviewer pointed out in comment #1, it was possible that we detected posttranscriptional modification of RNAs in our PRO-seq analysis. To exclude this possibility, we performed PRO-seq analysis in the presence of α -amanitin, which blocks transcription by Pol II, but not Pol I and Pol III, as described in comment #1 and shown in new Supplementary Fig. 4.

In accordance with the reviewer's suggestion, we have included text to address this point in the Results section as follows: In this study, we detected the TPP in our PRO-seq using permeabilized cells. In contrast, we did not detect the TPP in our previous study, in which we used Dounce homogenized cells⁵. The original protocol of PRO-seq recommends using permeabilized cells rather than Dounce homogenized cells because the recovery of transcripts is higher from permeabilized cells than Dounce homogenized cells⁴ and therefore we used permeabilized cells in this study. One possible reason for the differences in TPP detection is that the homogenizing process disrupts the factors that are involved in TPP. (page 11, lines 18–24)

To further confirm that the Mediator–LEC interaction is required for TPP at RDH genes, we performed PRO-seq analysis using a CRISPR-generated, MED26-hypomorphic-mutant HEK293T cell line that we used in our previous paper and that expresses mutant MED26 lacking the NTD required for LEC's interaction with Mediator⁵. Similar to the results of EAF1-mutant cells, the TPP of RDH genes disappeared in MED26-mutant cells (new Supplementary Fig. 7b–e). Meta-gene analysis of the RDH genes also revealed that TPP observed in wild-type cells was clearly decreased in MED26-mutant cells,

which was accompanied by the increased read-through ratio of RDH genes in MED26-mutant cells. However, the PPP of the genes was not significantly affected (new Supplementary Fig. 7f–h). In addition, meta-gene analysis revealed that TPP at snRNA genes was mildly decreased in MED26-mutant cells (new Supplementary Fig. 7i, j), which was consistent with the results of the EAF1-mutant cell line (Fig. 5f). In contrast, we found little change in PPP at the *GAPDH* gene in MED26-mutant cells (new Supplementary Fig. 7k). The results of the PRO-seq observed in MED26-mutant cells were similar to those of EAF1-mutant cells, strongly supporting the idea that Mediator’s interaction with LEC-CBC-NELF is required for TPP and subsequent 3’-end processing of RDH and snRNA genes. (page 17, lines 7–21)

3. In their new supplementary figure 2g, they show that “transcription” pauses at the 3’ end of snoRNAs present in introns. These RNAs are formed by transcription of the pre-mRNA followed by processing to release the snoRNAs. They would never be labeled precisely by the method the authors used unless the labeling was posttranscriptional.

As referenced by the reviewer, there are two types of snoRNA. One is poly- or mono-cistronic snoRNAs, which are transcribed independently, and the other is intronic snoRNAs, which are present in the introns of snoRNA host genes and processed by exonucleases from the introns of pre-mRNAs of the genes⁶. Thus, it is possible that we detected the posttranscriptional modification of snoRNAs in the PRO-seq analysis. To address this issue, we performed PRO-seq in the presence of α -amanitin. We found that α -amanitin markedly decreased the signals of TPP at both types of snoRNA genes (new Supplementary Fig. 4l, m). Our results indicate that the TPPs of snoRNA genes are derived from Pol II and not post-transcriptional modification.

4. Their results do not make biological sense (which is OK since we are always finding surprising new things), but they don’t try to explain how their results might be linked to histone pre-mRNA processing. If the polymerase stalls precisely at the stemloop, then there is 25 nt of nascent RNA inside the polymerase (not accessible to processing factors), and no processing factors can be recruited to the nascent mRNA. Results of Eaton et al. (also at the nt sequence level) shows pol II accumulating downstream of the processing site, which then allows the processing factors to identify the processing site, and gives them time to assemble on the nascent transcript.

We thank the reviewer for raising this issue. A recent finding showed that CDK11 directly interacts immediately upstream of the stem loop of RDH gene transcripts and plays an

important role in Ser2 phosphorylation of Pol II CTD to facilitate transcription termination⁷. We analyzed data of individual-nucleotide resolution cross-linking and immuno-precipitation (iCLIP) of CDK11 and Coilin^{7,8}. We found that, similar to Coilin, CDK11 interacts with the upstream region of TPP of RDH gene transcripts (Fig. 8i, j). (page 23 lines 22–24, and page 24 lines 1–6)

To investigate the role of CDK11 in transcription termination of RDH genes, we performed PRO-seq analysis using cells in which CDK11 was knocked down. Our results showed that knockdown of CDK11 increased the TPP at RDH genes, but not other protein coding genes (Figure below, a–d). These results raise the possibility that CDK11 bound upstream of TPP sites of RDH gene transcripts is involved in the release of TPP through phosphorylation of NELF and Ser2 of Pol II CTD. This result is consistent with the previous finding that CDK11 plays a role in the recruitment of 3'-end processing factors to RDH genes through phosphorylation of Ser2 of Pol II CTD and facilitate transcription termination. Based on these findings, we considered the possibility that TPP allows sufficient phosphorylation of Ser2 by CDK11 to recruit 3'-end processing factors to the RDH genes and facilitate transcription termination. Thus, we propose the possibility that TPP may have a role as a checkpoint at the transition from transcription elongation to termination for appropriate 3'-end processing to produce non-polyadenylated transcripts of RDH genes. The detailed mechanism by which the transcription termination of RDH genes is regulated by TPP should be elucidated in future studies.

We address the biological relevance of TPP at RDH genes in the Results section as follows: “On the basis of our findings, we propose a model for the role of Mediator’s interaction with LEC–CBC–NELF in TPP and subsequent 3’-end processing of RDH genes. In this model, MED26-containing Mediator plays a role in the association of CBs with HLBs through interaction with LEC–CBC–NELF. CBs’ association with HLBs leads to TPP at RDH genes and subsequent 3’-end processing by supplying 3’-end processing factors from CBs (Fig. 10). Notably, a recent report and our finding showed that CDK11 and Coilin directly interact immediately upstream of the stem-loop of RDH gene transcripts (Fig. 8j) and that CDK11 plays an important role in Ser2 phosphorylation of the Pol II CTD to recruit 3’-end processing factors^{52, 84}. We found that Ser2 phosphorylation of Pol II CTD is decreased at RDH genes in EAF1-mutant cells (Supplementary Fig. 6b). Thus, it is possible that TPP plays a role in sufficient Ser2 phosphorylation of Pol II CTD by CDK11 to recruit 3’-end processing factors to the RDH genes. Taken together, TPP may have a role as a checkpoint at the transition from transcription elongation to termination for appropriate 3’-end processing to produce non-polyadenylated transcripts of RDH genes (Fig. 10).” (page 26, lines 4–17)

We have included the following text in the Discussion section: “Our results raise the possibility that TPP is a key checkpoint process regulating the transition from transcription elongation to transcription termination (Fig. 10). A recent report showed that CDK11 binds immediately upstream of the stem loop of RDH gene transcripts and plays an essential role in the recruitment of 3’-end processing factors to RDH genes through phosphorylation of Ser2 of Pol II CTD⁸⁴. We found that CDK11 also colocalized with CBs (Fig. 8) and that both CDK11 and Coilin bind to a region immediately upstream of the sites of 3’ Pol II pausing of RDH gene transcripts (Fig. 8). This raises the possibility that CBs associated with HLBs include the nascent RDH gene transcripts synthesized by Pol II, which pauses immediately upstream of the TES. Considering that Ser2 phosphorylation of Pol II CTD is decreased at RDH genes in EAF1-mutant cells (Supplementary Fig. 6), it is possible that TPP allows sufficient Ser2 phosphorylation of Pol II CTD by CDK11 to recruit 3’-end processing factors to RDH genes (Fig. 10). The detailed mechanism by which the transcription termination of RDH genes is regulated by TPP should be elucidated in future studies. In addition, recent reports showed that protein phosphatase 1 (PP1) or protein phosphatase 2A (PP2A) dephosphorylates Spt5, a component of DSIF, and/or Pol II CTD to decelerate Pol II elongation downstream of the TES of RDH or snRNA genes, respectively^{15, 16, 17, 18}. These reports suggest that deceleration of Pol II beyond TES contributes to the final process of transcription termination through facilitating 5’→3’ degradation of RNAs associated with Pol II.” (page 28, lines 8–24, and page 29, line 1-2)

We have also included the following text in the Figure legend of Fig. 10: “TPP plays a role in sufficient Ser2 phosphorylation of Pol II CTD by CDK11 to recruit 3’-end processing

factors to the RDH genes. Thus, TPP acts as a checkpoint from transcription elongation to transcription termination for appropriate 3'-end processing and production of non-polyadenylated RDH gene transcripts.” (page 64, lines 13–16)

5. The HLB is very specific and easy to identify since it contains NPAT, which is found only on histone genes (Kaya-Okur et al., 2019). Cajal bodies are much harder to unambiguously identify since some HLBs contain coilin (this doesn't make them Cajal bodies, which are better identified by the presence of scaRNAs. Cajal bodies are clearly involved in maturation of snRNAs so they will contain some U7 snRNP, but mature U7 snRNP is present concentrated in the HLB. Papers before 2007 (when the HLB was first described as distinct from Cajal bodies) referred to what we now know are HLBs as Cajal bodies. They do not cite (Barcaroli et al., 2006) where they identified bodies containing NPAT and FLASH at histone genes, which were not affected by manipulating coilin, and further studies by this group confirmed that there is a difference between the NPAT containing bodies and other Cajal bodies (Bongiorno-Borbone et al., 2008). These "cajal bodies" were clearly HLBs and not Cajal bodies, and that paper was adjacent to the one they cited. There are not good reagents to track U7 snRNP in mammalian cells (good antibodies to Lsm10 and Lsm11 are not available). It is possible that once U7 snRNP is matured in Cajal bodies it then is actively directed to the HLB.

We thank the reviewer for raising this issue. As the reviewer pointed out, HLBs were mistaken to be CBs for decades until HLBs were identified as a distinct nuclear body localized at RDH gene clusters in *Drosophila melanogaster*⁹. HLB is easy to identify because it contains NPAT and FLASH^{10, 11}. Previous studies showed that NPAT is only present at histone gene clusters¹². After the initial finding of HLBs, CBs and HLBs have been recognized as distinct nuclear bodies in a variety of species^{10, 13, 14, 15}. In accordance with the reviewer's suggestion to avoid the confusion between CBs and HLBs, we changed the Introduction section in the revised manuscript and included the following text: “The transcription of RDH genes and snRNA genes is regulated at two nuclear bodies, histone locus bodies (HLBs) and Cajal bodies (CBs), respectively^{26, 27}. After the initial finding of HLBs as nuclear bodies localized at RDH gene clusters in *Drosophila melanogaster*²⁸, CBs and HLBs have been recognized as distinct nuclear bodies in a variety of species^{29, 30, 31, 32}. ” (page 5, lines 3–6)

In accordance with the reviewer's suggestion, we cited the papers that the reviewer has highlighted. We changed the Introduction section in the revised manuscript and included the following text: “Intriguingly, HLBs and CBs frequently and physically associate with each other, suggesting that there is a functional link between them^{27, 29, 30, 44}. The association of CBs and

HLBs was shown to be increased at S phase in which RDH genes are transcribed^{31, 32, 45, 46}. It was also shown that colocalization of CBs and HLBs is increased at the late stage of the oogenesis in both the *Xenopus* and *Drosophila*⁴⁷. HLBs formation was not affected by the depletion of Coilin^{48, 49}, in contrast, localization of Coilin was affected by the depletion of HLB components NPAT or FLASH^{50, 51}. Thus, the molecular mechanism and biological importance of the colocalization of CBs and HLBs remain unknown.” (page 6, lines 1–8)

To clarify the reviewer’s comment that Cajal bodies are harder to unambiguously identify because some HLBs contain coilin, we performed immunofluorescence of SMN1 and WRAP53, which colocalize at CBs^{14, 16, 17}. SMN1 is one of the components of the SMN complex that plays a critical role in snRNP biogenesis through translocating snRNPs from the cytoplasm to CBs^{18, 19, 20, 21}. WRAP53 is essential for CB formation and plays a role in targeting Small Cajal body–specific RNAs (scaRNAs) to CBs through direct interaction^{17, 22}. We performed triple immunofluorescence analysis of NPAT, Coilin and SMN1 or WRAP53 in wild-type or EAF1-mutant HEK293T cells. As shown in new Supplementary Fig. 9a and b, Coilin particles containing SMN1 or WRAP53 were associated with NPAT in wild-type HEK293T cells. In contrast, the NPAT association with Coilin particles containing SMN1 or WRAP53 was decreased in EAF1-mutant cells. Triple immunofluorescence analysis revealed that a large amount of Coilin particles contained WRAP53 and that their association with NPAT was decreased in EAF1-mutant cells (new Supplementary Fig. 9c). These results showed that Coilin particles in both wild-type and EAF1-mutant cells contain other CB components including SMN1 and WRAP53, suggesting that they have functions for targeting scaRNAs to CBs and snRNP biogenesis. In addition, we performed RNA-fluorescence *in situ* hybridization of scaRNAs followed by immunofluorescence of Coilin and NPAT. Consistent with the previous findings that scaRNAs colocalize with CBs²³, Coilin particles containing scaRNA12 were colocalized with NPAT in HeLa cells (new Supplementary Fig. 10a). (page 18, lines 22–24, and page 19, lines 1–15)

To evaluate how many CBs associate with histone gene loci in HLBs, we developed the novel technique of “*in situ* biotin-labelling of protein with high-throughput DNA sequencing” using an antibody-based *in situ* biotinylation technique²⁴. We used this technique to identify the genomic regions included in Coilin particles. The proteins present within approximately 20 nm from Coilin were labelled with biotin by an anti-Coilin antibody–based biotinylation reaction in cells (Supplementary Fig. 13a, b) and then biotin-labelled proteins with genomic DNA were purified, followed by high-throughput sequencing (Fig. 8a). Two RDH gene loci and snRNA/snoRNA gene loci were strongly detected as the genes present in the proximity of Coilin (Fig. 8b–e), indicating that the CB’s associations with two RDH gene loci and snRNA/snoRNA gene loci were successfully detected by this method. This result is consistent

with the recent genome-wide 4C-seq analysis of CBs showing that CBs are associated with these gene loci²⁵. Intriguingly, the interaction of Coilin with RDH gene loci was dramatically attenuated in EAF1-mutant cells and ICE1- or MED26-knockdown cells (Fig. 8c, d, and Supplementary Fig. 13c, d), suggesting that Mediator's interaction with LEC contributes to CBs' association with RDH gene loci. In contrast, we found that CB's association with snRNA/snoRNA gene loci was only mildly decreased in EAF1-mutant cells compared with CB's association with RDH gene loci (Fig. 8d, e), consistent with the results that the number of CBs, TPP at snRNA genes and read-through of Pol II at snRNA genes were mildly affected in EAF1-mutant cells (Fig. 6c, and Fig. 5e, f, k). These results indicate that CBs in wild-type cells have functions for genomic conformation at two RDH gene loci and snRNA/snoRNA gene loci. In contrast, Coilin particles in EAF1-mutant cells dissociate from HLBs including RDH gene loci; however, they retain genomic conformation function at snRNA/snoRNA gene loci. (page 22, lines 14–24, and page 23, lines 1–8)

In addition, we performed DNA-FISH of RDH gene cluster 1 followed by immunofluorescence of both Coilin and WRAP53. As shown in new Supplementary Fig. 14b and c, RDH gene cluster 1 colocalized with particles containing both Coilin and WRAP53 in wild-type cells. In contrast, we observed decreased colocalization between the RDH gene cluster 1 and the particles containing both Coilin and WRAP53 in EAF1-mutant cells. (page 24, lines 23–24, and page 25, lines 1–7)

Taken together, our results showed that Coilin particles in wild-type cells contained other CB components including WRAP53, SMN1, scaRNAs and genomic regions including RDH gene loci and snRNA/snoRNA gene loci, suggesting that they have functions for targeting scaRNAs, snRNP biogenesis and genomic conformation of RDH gene and snRNA/snoRNA gene loci. In EAF1-mutant cells, although Coilin particles dissociated from HLBs including RDH gene clusters, they still contained WRAP53, SMN1 and snRNA/snoRNA gene loci. This result suggests that Coilin particles in EAF1-mutant cells retain functions for targeting scaRNAs, snRNP biogenesis and genomic conformation at snRNA/snoRNA gene loci. Thus, our results support the idea that the Coilin particles that we observed in cells have at least some of the functions of CBs.

6. References

Barcaroli, D., Bongiorno-Borbone, L., Terrinoni, A., Hofmann, T.G., Rossi, M., Knight, R.A., Matera, A.G., Melino, G., and De, L., V (2006). FLASH is required for histone transcription and S-phase progression. Proc. Natl. Acad. Sci. U. S. A 103, 14808-14812. 0604227103 [pii];10.1073/pnas.0604227103 [doi].
Bongiorno-Borbone, L., De, C.A., Vernole, P., Finos, L., Barcaroli, D., Knight, R.A.,

Melino, G., and De, L., V (2008). FLASH and NPAT positive but not Coilin positive Cajal Bodies correlate with cell ploidy. *Cell Cycle* 7, 2357-2367. 6344 [pii].

Kaya-Okur, H.S., Wu, S.J., Codomo, C.A., Pledger, E.S., Bryson, T.D., Henikoff, J.G., Ahmad, K., and Henikoff, S. (2019). CUT&Tag for efficient epigenomic profiling of small samples and single cells. *Nat Commun* 10, 1930. 10.1038/s41467-019-09982-5.

Lackey, P.E., Welch, J.D., and Marzluff, W.F. (2016). TUT7 catalyzes the uridylation of the 3' end for rapid degradation of histone mRNA. *RNA* 22, 1673-1688. 10.1261/rna.058107.116.

Welch, J.D., Slevin, M.K., Tatomer, D.C., Duronio, R.J., Prins, J.F., and Marzluff, W.F. (2015). EnD-Seq and AppEnD: sequencing 3' ends to identify nontemplated tails and degradation intermediates. *RNA* 21, 1375-1389. 10.1261/rna.048785.114.

Nojima T, Gomes T, Carmo-Fonseca M, Proudfoot NJ. 2016. Mammalian NET-seq analysis defines nascent RNA profiles and associated RNA processing genome-wide. *Nat Protoc* 11: 413–428

We appreciate the reviewer for suggesting the citation of these important papers. We cited almost all of the papers listed above and discussed the recent progress of the field in the revised manuscript.

References

1. Rudd MD, Luse DS. Amanitin greatly reduces the rate of transcription by RNA polymerase II ternary complexes but fails to inhibit some transcript cleavage modes. *The Journal of biological chemistry* **271**, 21549-21558 (1996).
2. Kaplan CD, Larsson KM, Kornberg RD. The RNA polymerase II trigger loop functions in substrate selection and is directly targeted by alpha-amanitin. *Molecular cell* **30**, 547-556 (2008).
3. Chafin DR, Guo H, Price DH. Action of alpha-amanitin during pyrophosphorolysis and elongation by RNA polymerase II. *The Journal of biological chemistry* **270**, 19114-19119 (1995).
4. Mahat DB, *et al.* Base-pair-resolution genome-wide mapping of active RNA polymerases using precision nuclear run-on (PRO-seq). *Nat Protoc* **11**, 1455-1476 (2016).
5. Takahashi H, *et al.* The role of Mediator and Little Elongation Complex in transcription termination. *Nature communications* **11**, 1063 (2020).
6. Matera AG, Terns RM, Terns MP. Non-coding RNAs: lessons from the small nuclear and small nucleolar RNAs. *Nat Rev Mol Cell Biol* **8**, 209-220 (2007).
7. Gajduskova P, Ruiz de Los Mozos I, Rajecky M, Hluchy M, Ule J, Blazek D. CDK11 is required for transcription of replication-dependent histone genes. *Nat Struct Mol Biol* **27**, 500-510 (2020).
8. Machyna M, *et al.* The coilin interactome identifies hundreds of small noncoding RNAs that traffic through Cajal bodies. *Molecular cell* **56**, 389-399 (2014).
9. Liu JL, Murphy C, Buszczak M, Clatterbuck S, Goodman R, Gall JG. The *Drosophila melanogaster* Cajal body. *J Cell Biol* **172**, 875-884 (2006).
10. Bongiorno-Borbone L, *et al.* FLASH and NPAT positive but not Coilin positive Cajal Bodies correlate with cell ploidy. *Cell cycle* **7**, 2357-2367 (2008).

11. Barcaroli D, *et al.* FLASH is required for histone transcription and S-phase progression. *Proceedings of the National Academy of Sciences of the United States of America* **103**, 14808-14812 (2006).
12. Kaya-Okur HS, *et al.* CUT&Tag for efficient epigenomic profiling of small samples and single cells. *Nature communications* **10**, 1930 (2019).
13. Nizami Z, Deryusheva S, Gall JG. The Cajal body and histone locus body. *Cold Spring Harbor perspectives in biology* **2**, a000653 (2010).
14. Machyna M, Heyn P, Neugebauer KM. Cajal bodies: where form meets function. *Wiley interdisciplinary reviews RNA* **4**, 17-34 (2013).
15. Ghule PN, *et al.* Cell cycle dependent phosphorylation and subnuclear organization of the histone gene regulator p220(NPAT) in human embryonic stem cells. *Journal of cellular physiology* **213**, 9-17 (2007).
16. Morse R, Shaw DJ, Todd AG, Young PJ. Targeting of SMN to Cajal bodies is mediated by self-association. *Human molecular genetics* **16**, 2349-2358 (2007).
17. Mahmoudi S, *et al.* WRAP53 is essential for Cajal body formation and for targeting the survival of motor neuron complex to Cajal bodies. *PLoS biology* **8**, e1000521 (2010).
18. Stanek D. Cajal bodies and snRNPs - friends with benefits. *RNA biology* **14**, 671-679 (2017).
19. Roithova A, *et al.* The Sm-core mediates the retention of partially-assembled spliceosomal snRNPs in Cajal bodies until their full maturation. *Nucleic acids research* **46**, 3774-3790 (2018).
20. Battle DJ, Lau CK, Wan L, Deng H, Lotti F, Dreyfuss G. The Gemin5 protein of the SMN complex identifies snRNAs. *Molecular cell* **23**, 273-279 (2006).
21. Battle DJ, *et al.* The SMN complex: an assembly machine for RNPs. *Cold Spring Harbor symposia on quantitative biology* **71**, 313-320 (2006).

22. Tycowski KT, Shu MD, Kukoyi A, Steitz JA. A conserved WD40 protein binds the Cajal body localization signal of scaRNP particles. *Molecular cell* **34**, 47-57 (2009).
23. Richard P, Darzacq X, Bertrand E, Jady BE, Verheggen C, Kiss T. A common sequence motif determines the Cajal body-specific localization of box H/ACA scaRNAs. *The EMBO journal* **22**, 4283-4293 (2003).
24. Bar DZ, Atkatsk K, Tavarez U, Erdos MR, Gruenbaum Y, Collins FS. Biotinylation by antibody recognition-a method for proximity labeling. *Nat Methods* **15**, 127-133 (2018).
25. Wang Q, *et al.* Cajal bodies are linked to genome conformation. *Nature communications* **7**, 10966 (2016).

REVIEWERS' COMMENTS

Reviewer #3 (Remarks to the Author):

The authors have addressed many of the concerns raised (1) in the last review and added substantial new data. More important the discussion of the results is now more balanced. I still think the Cajal body part is written too strongly (there are cells that don't have visible Cajal bodies that grow fine). The evidence coilin crosslinks to histone genes (at least in cells that have Cajal bodies that colocalize with histone genes, as noted in ref. 42). However it certainly is possible that coilin does interact with histone transcripts even if a Cajal body is not visible. The authors provide some rationale for why their results differ from others, including their own previous results.

Comments:

- 1, Page 4, l.3, They need to define what they mean by TES's better. I think they mean transcript end site, formed after processing, and not transcription end sites (which is where termination occurs). It is critical that they clearly make this distinction, since it is unusual nomenclature, and as written "transcription end sites", strongly implies the site of transcription termination.
2. page 7, line 16; need to specify here whether these are snoRNA genes present in introns or snoRNAs transcribed as individual genes (e.g. U3 snoRNA or U8 snoRNA).
3. page 8, l. 14. There is no compelling evidence in the paper (or the literature) the CBs provide 3' processing factors to the HLBs while histone gene transcription is taking place. U7 snRNP is present in HLBs and acts catalytically in the HLB. Delete this conclusion from the introduction.
4. Page 9, l. 15. Calling the TES proximal region (which is before processing occurs), TER is horrible nomenclature and MUST be changed. Everyone will think that means "terminator". Why not call it TES?
5. Pg. 10, l/13, 14. Change "it is thought our results do not contradict reports" to "we think our results do not contradict previous reports". When the people who did those experiments read this paper they will think it contradicts their results, and it is necessary for the authors to argue why it doesn't, which they do in line 17,18) by indicating they only get this result if they permeabilized cells, and not if they homogenized them which is done in all the other protocols, including one of their previous papers.
6. Fig. 5J needs better axes and description in the legend. Is this a pol II CHIP-Seq?. I think it is. They say this is a log scale and only show 2 values (0 and 50). Need to put some intermediate values on the scale (I presume 50 does not mean 250). On the figure HIST2H2BC is a pseudo gene and should be deleted. This is an inverted repeat in the genome which explains why the data looks identical on each side. They need to mention this or just show one side. There is a pseudoH2B gene on the other side which is not annotated on the browser. The log scale clearly will overemphasize the read-through, as the box plot in panel k shows the very small absolute difference.
- 7.. In Fig 8j I would indicate the position of the stemloop on the RDH genes. It is a defined sequence, while the TES is their general region and in Fig. 8j almost certainly contains the HDE, which is not how they have talked about it before, or show it in Fig. 10.
8. bottom page 23 and 24. , They should clearly state the study in ref. 84 showed that CDK11 localized on histone mRNAs just before the stem-loop (a major point of that paper and a novel finding). They should change their first sentence in that paragraph to "(CDK11 binds to the 3' region of RDH gene transcripts, just before the stemloop [insert this] and plays an important role in the 3' processing on those transcripts.....".
What they did is take the data of ref. 42, where the authors did not comment on where on histone genes the coilin was crosslinked and showed that coilin maps to the same place. The discovery that cdk11 crosslinks to that region was made by Blazek and coworkers in ref. 84.
- 9.. Similarly their paragraph on the bottom of page 26 also does not discuss the previous work appropriately (l. 9-12). The first sentence should say "a recent report showed that Cdk 11 binds to the histone transcripts immediately upstream of the stem loop of RDH gene transcripts and that CDK11 plays role in ser 2 phosphorylation of the pol II CTD and histone mRNA processing (Ref 84)." We found that coilin binds to the same site by analyzing the data in ref. 42)..
8. Note on page 28 they have inserted essentially the same paragraph as on page 26 , but this paragraph is much better written acknowledging the contributions of others appropriately. I would

delete the paragraph on page 26 and end the result with the reference to the model.

9.. The last sentence of the paper on page 32, it would better to say the "the controlled induction of synthesis of histone mRNAs". A major part of the induction of histone mRNA synthesis is the synthesis of SLBP [1] which occurs at the same time as NPAT is phosphorylated, and without which histone mRNA cannot be processed.. Many studies have shown that histone gene transcription is increased then, but only 3-5 fold [2, 3].

10. In Fig. 10, they need to indicate when processing occurs and then termination which must be well after the HDE (on the scale of the inset in Fig. 10) since the HDE has to be transcribed and exit the polymerase before it can be processed. Maybe indicate after the stop sign, slow transcription occurs past the HDE and then processing occurs, followed by termination somewhere further downstream
Minor comments:

1. Introduction l. 11,12, page 3. WDR33 is also a component of CPSF as is symplekin, and should be added here. They should also mention CstF here. The two multimeric factors originally described were CPSF and CstF. In the Feb. 2022 issue of *Genes and Development* there are two papers that reconstitute cleavage and polyadenylation, and define the two subcomplexes of CPSF.

2. Pg. 4 lines 12-17. Only a small subset of the RDH histone genes make polyadenylated mRNAs in terminally differentiated cells, or when processing factors are depleted.

3. Pg. 5, l. 15 HLBs are the site of histone gene transcription and 3' processing [4].

4, Pg. 6 l.6,7. I think they mean that "depletion of NPAT or FLASH result in loss of coilin from the HLBS at histone genes (which disappeared), but the other CBs were still present.

Harris, M.E., Böhni, R., Schneiderman, M.H., Ramamurthy, L., Schümperli, D., and Marzluff, W.F. (1991). Regulation of histone mRNA in the unperturbed cell cycle: Evidence suggesting control at two posttranscriptional steps. *Mol Cell Biol* 11, 2416-2424.

Heintz, N., Sive, H.L., and Roeder, R.G. (1983). Regulation of human histone gene expression: kinetics of accumulation and changes in the rate of synthesis and in the half-lives of individual histone mRNAs during the HeLa cell cycle. *Mol Cell Biol* 3, 539-550.

Kemp, J.P., Jr., Yang, X.C., Dominski, Z., Marzluff, W.F., and Duronio, R.J. (2021). Superresolution light microscopy of the *Drosophila* histone locus body reveals a core-shell organization associated with expression of replication-dependent histone genes. *Mol Biol Cell* 32, 942-955.

Whitfield, M.L., Zheng, L.-X., Baldwin, A., Ohta, T., Hurt, M.M., and Marzluff, W.F. (2000). Stem-loop binding protein, the protein that binds the 3' end of histone mRNA, is cell cycle regulated by both translational and posttranslational mechanisms. *Mol Cell Biol* 20, 4188-4198.

POINT-BY-POINT RESPONSES TO THE REVIEWERS' COMMENTS

Please find our responses to the reviewers' comments listed below. The reviewers' comments are shown in a bold black font, and our responses to the comments are shown in green. Text quoted from the revised manuscript is shown in brown.

The authors have addressed many of the concerns raised (1) in the last review and added substantial new data. More important the discussion of the results is now more balanced. I still think the Cajal body part is written too strongly (there are cells that don't have visible Cajal bodies that grow fine). The evidence coilin crosslinks to histone genes (at least in cells that have Cajal bodies that colocalize with histone genes, as noted in ref. 42). However it certainly is possible that coilin does interact with histone transcripts even if a Cajal body is not visible. The authors provide some rationale for why their results differ from others, including their own previous results.

We are pleased that our changes were mostly approved by the reviewer.

Comments:

1, Page 4, l.3, They need to define what they mean by TES's better. I think they mean transcript end site, formed after processing, and not transcription end sites (which is where termination occurs). It is critical that they clearly make this distinction, since it is unusual nomenclature, and as written "transcription end sites", strongly implies the site of transcription termination.

We thank the reviewer for raising this issue. As the reviewer pointed out, TES means 'transcript end site', not 'transcription end site'. In accordance with the reviewer's suggestion, we changed this in the revised manuscript.

2. page 7, line 16; need to specify here whether these are snoRNA genes present in introns or snoRNAs transcribed as individual genes (e.g. U3 snoRNA or U8 snoRNA).

We thank the reviewer for raising this issue. These snoRNA genes include both cistronic and intronic snoRNA genes. Therefore, we specified this point in the revised manuscript (page 7, line 14).

3. page 8, l. 14. There is no compelling evidence in the paper (or the literature) the CBs

provide 3' processing factors to the HLBs while histone gene transcription is taking place. U7 snRNP is present in HLBs and acts catalytically in the HLB. Delete this conclusion from the introduction.

We thank the reviewer for raising this issue. In accordance with the reviewer's suggestion, we deleted the conclusion describing that CBs provide 3'-end processing factors to the HLBs during histone gene transcription in the introduction of the revised manuscript (page 8, lines 10–11).

4. Page 9, l. 15. Calling the TES proximal region (which is before processing occurs), TER is horrible nomenclature and MUST be changed. Everyone will think that means “terminator”. Why not call it TES?

We thank the reviewer for raising this issue. In accordance with the reviewer's suggestion, we changed “TER” to “TESr” in the revised manuscript (page 9, line 15).

5. Pg. 10, l/13, 14. Change “it is thought our results do not contradict reports” to “we think our results do not contradict previous reports”. When the people who did those experiments read this paper they will think it contradicts their results, and it is necessary for the authors to argue why it doesn't, which they do in line 17,18) by indicating they only get this result if they permeabilized cells, and not if they homogenized them which is done in all the other protocols, including one of their previous papers.

We thank the reviewer for raising this issue and agree with the reviewer's suggestion. In accordance with the reviewer's suggestion, we changed “it is thought our results do not contradict reports” to “we think that our results do not contradict the previous reports” in the revised manuscript (page 10, lines 13–14).

6. Fig. 5J needs better axes and description in the legend. Is this a pol II CHIP-Seq?. I think it is. They say this is a log scale and only show 2 values (0 and 50). Need to put some intermediate values on the scale (I presume 50 does not mean 250). On the figure HIST2H2BC is a pseudo gene and should be deleted. This is an inverted repeat in the genome which explains why the data looks identical on each side. They need to mention this or just show one side. There is a pseudoH2B gene on the other side which is not annotated on the browser. The log scale clearly will overemphasize the read-through, as the box plot in panel k shows the very small absolute difference.

We thank the reviewer for raising this issue. As the reviewer pointed out, Fig. 5J gives an exaggerated impression. We changed Fig. 5j to revised Fig. 5j with log scale 0–20 and a narrower range of the RDH genomic region than in the original figure. In addition, we deleted the HIST2H2BC pseudogene in revised Fig. 5J.

7. In Fig 8j I would indicate the position of the stemloop on the RDH genes. It is a defined sequence, while the TES is their general region and in Fig. 8j almost certainly contains the HDE, which is not how they have talked about it before, or show it in Fig. 10.

We thank the reviewer for raising this point. In accordance with the reviewer's suggestion, we indicated the stem-loop and HDE in revised Fig. 8j.

8. bottom page 23 and 24. , They should clearly state the study in ref. 84 showed that CDK11 localized on histone mRNAs just before the stem-loop (a major point of that paper and a novel finding). They should change their first sentence in that paragraph to “(CDK11 binds to the 3’ region of RDH gene transcripts, just before the stemloop [insert this] and plays an important role in the 3’ processing on those transcripts.....”. What they did is take the data of ref. 42, where the authors did not comment on where on histone genes the coilin was crosslinked and showed that coilin maps to the same place. The discovery that cdk11 crosslinks to that region was made by Blazek and coworkers in ref. 84.

We thank the reviewer for raising this issue. In accordance with the reviewer's suggestion, we changed the paragraph. Now we state: ‘A recent study showed that cyclin-dependent kinase 11 (CDK11) binds to the 3’ region of RDH gene transcripts, just before the stem-loop, and plays an important role in the 3’-end processing of those transcripts through Ser2 phosphorylation of the Pol II CTD⁸⁶. Since we found that CDK11 colocalized at CBs (Fig. 8), we took advantage of the individual-nucleotide-resolution UV crosslinking and immunoprecipitation (iCLIP)–seq data of Coilin and CDK11 and compared them at the RDH genes^{45, 86}. This analysis revealed that both Coilin and CDK11 bind to the region immediately upstream of the stem-loop of RDH gene transcripts (Fig. 8j). This raises the possibility that CBs associate with the nascent RDH gene transcripts synthesized by Pol II, which pauses within the stem-loop region of RDH genes.’ (page 23, lines 21–24, and page 24, lines 1–5)

9 & 10. Similarly their paragraph on the bottom of page 26 also does not discuss the

previous work appropriately (l. 9-12). The first sentence should say “a recent report showed that Cdk 11 binds to the histone transcripts immediately upstream of the stem loop of RDH gene transcripts and that CDK11 plays role in ser 2 phosphorylation of the pol II CTD and histone mRNA processing (Ref 84).” We found that coilin binds to the same site by analyzing the data in ref. 42). Note on page 28 they have inserted essentially the same paragraph as on page 26, but this paragraph is much better written acknowledging the contributions of others appropriately. I would delete the paragraph on page 26 and end the result with the reference to the model.

We thank the reviewer for raising this issue. In accordance with the reviewer’s suggestion, we deleted the paragraph from line 7 to line 15 of page 26 of the original manuscript.

11. The last sentence of the paper on page 32, it would better to say the “the controlled induction of synthesis of histone mRNAs”. A major part of the induction of histone mRNA synthesis is the synthesis of SLBP [1] which occurs at the same time as NPAT is phosphorylated, and without which histone mRNA cannot be processed.. Many studies have shown that histone gene transcription is increased then, but only 3-5 fold [2, 3].

We thank the reviewer for raising this issue. In accordance with the reviewer’s suggestion, we changed ‘the controlled induction of the transcription of RDH genes’ to ‘the controlled induction of synthesis of RDH mRNAs’ and cited the papers that the reviewer has highlighted (page 32, lines 1–2).

12. In Fig. 10, they need to indicate when processing occurs and then termination which must be well after the HDE (on the scale of the inset in Fig. 10) since the HDE has to be transcribed and exit the polymerase before it can be processed. Maybe indicate after the stop sign, slow transcription occurs past the HDE and then processing occurs, followed by termination somewhere further downstream.

We thank the reviewer for raising this issue. In accordance with the reviewer’s suggestion, we indicated that slow elongation occurs past the HDE, and then 3'-end processing occurs, followed by termination, in revised Fig. 10.

Minor comments:

1. Introduction l. 11,12, page 3. WDR33 is also a component of CPSF as is symplekin, and should be added here. They should also mention CstF here. The two multimeric factors

originally described were CPSF and CstF. In the Feb. 2022 issue of Genes and Development there are two papers that reconstitute cleavage and polyadenylation, and define the two subcomplexes of CPSF.

We thank the reviewer for raising this issue. In accordance with the reviewer's suggestion, we mentioned WDR33 and CstF in the introduction and cited the paper that the reviewer has highlighted. Now we state: 'Such processing of genes producing polyadenylated mRNA is achieved by multimeric cleavage and polyadenylation-specific factors (CPSFs), consisting of CPSF1 (CPSF160), CPSF2 (CPSF100), CPSF3 (CPSF73), CPSF4 (CPSF30), hFip1 and WDR33, and multimeric cleavage stimulation factors (CstFs) which produce poly(A) tail transcripts^{6, 7, 8}' (page 3, lines 10–13).

2. Pg. 4 lines 12-17. Only a small subset of the RDH histone genes make polyadenylated mRNAs in terminally differentiated cells, or when processing factors are depleted.

In accordance with the reviewer's suggestion, we changed the text that the reviewer pointed out. Now we state: 'In addition, knockdown of 3'-end processing factors or terminally differentiation of cells cause read-through past normal sites of termination to conserved polyadenylation signals (PASs) just downstream of HDEs, resulting in the synthesis of polyadenylated RDH mRNAs of a small subset of the RDH genes^{21, 22}' (page 4, lines 11–15).

3. Pg. 5, l. 15 HLBs are the site of histone gene transcription and 3' processing [4].

In accordance with the reviewer's suggestion, we changed the text that the reviewer pointed out and cited the paper that the reviewer has highlighted. Now we state: 'HLBs are the sites for RDH genes transcription and 3'-end processing of RDH gene transcripts^{42, 43}' (page 5, lines 15–16).

4, Pg. 6 l.6,7. I think they mean that “depletion of NPAT or FLASH result in loss of coilin from the HLBS at histone genes (which disappeared), but the other CBs were still present.

We thank the reviewer for raising this issue. In accordance with the reviewer's suggestion, we changed 'localization of Coilin was affected by the depletion of HLB components NPAT or FLASH' to 'depletion of HLB components NPAT or FLASH result in loss of Coilin from the HLBs at RDH genes' in the revised manuscript (page 6, lines 6–7).

Reference list

1. Harris, M.E., Böhni, R., Schneiderman, M.H., Ramamurthy, L., Schümperli, D., and Marzluff, W.F. (1991). Regulation of histone mRNA in the unperturbed cell cycle: Evidence suggesting control at two posttranscriptional steps. *Mol Cell Biol* 11, 2416-2424.
2. Heintz, N., Sive, H.L., and Roeder, R.G. (1983). Regulation of human histone gene expression: kinetics of accumulation and changes in the rate of synthesis and in the half-lives of individual histone mRNAs during the HeLa cell cycle. *Mol Cell Biol* 3, 539-550.
3. Kemp, J.P., Jr., Yang, X.C., Dominski, Z., Marzluff, W.F., and Duronio, R.J. (2021). Superresolution light microscopy of the *Drosophila* histone locus body reveals a core-shell organization associated with expression of replication-dependent histone genes. *Mol Biol Cell* 32, 942-955.
4. Whitfield, M.L., Zheng, L.-X., Baldwin, A., Ohta, T., Hurt, M.M., and Marzluff, W.F. (2000). Stem-loop binding protein, the protein that binds the 3' end of histone mRNA, is cell cycle regulated by both translational and posttranslational mechanisms. *Mol Cell Biol* 20, 4188-4198.

We appreciate the reviewer for suggesting the citation of these important papers. We cited all of the papers listed above in the revised manuscript.